# Global Guarantees for Blind Demodulation with Generative Priors

**Paul Hand**
Dept. of Mathematics and College of Computer Science and Information
Northeastern University, MA
p.hand@northeastern.edu

**Babhru Joshi**
Dept. of Mathematics
University of British Columbia, BC
b.joshi@math.ubc.ca

## Abstract

We study a deep learning inspired formulation for the blind demodulation problem, which is the task of recovering two unknown vectors from their entrywise multiplication. We consider the case where the unknown vectors are in the range of known deep generative models, $\mathcal{G}^{(1)} : \mathbb{R}^n \to \mathbb{R}^\ell$ and $\mathcal{G}^{(2)} : \mathbb{R}^p \to \mathbb{R}^\ell$. In the case when the networks corresponding to the generative models are expansive, the weight matrices are random and the dimension of the unknown vectors satisfy $\ell = \Omega(n^2 + p^2)$, up to log factors, we show that the empirical risk objective has a favorable landscape for optimization. That is, the objective function has a descent direction at every point outside of a small neighborhood around four hyperbolic curves. We also characterize the local maximizers of the empirical risk objective and, hence, show that there does not exist any other stationary points outside of these neighborhood around four hyperbolic curves and the set of local maximizers. We also implement a gradient descent scheme inspired by the geometry of the landscape of the objective function. In order to converge to a global minimizer, this gradient descent scheme exploits the fact that exactly one of the hyperbolic curve corresponds to the global minimizer, and thus points near this hyperbolic curve have a lower objective value than points close to the other spurious hyperbolic curves. We show that this gradient descent scheme can effectively remove distortions synthetically introduced to the MNIST dataset.

## 1 Introduction

We study the problem of recovering two unknown vectors $\boldsymbol{x}_0 \in \mathbb{R}^\ell$ and $\boldsymbol{w}_0 \in \mathbb{R}^\ell$ from observations $\boldsymbol{y}_0 \in \mathbb{R}^\ell$ of the form

$$\boldsymbol{y}_0 = \boldsymbol{w}_0 \odot \boldsymbol{x}_0, \tag{1}$$

where $\odot$ is entrywise multiplication. This bilinear inverse problem (BIP) is known as the blind demodulation problem. BIPs, in general, have been extensively studied and include problems such as blind deconvolution/demodulation [Ahmed et al., 2014, Stockham et al., 1975, Kundur and Hatzinakos, 1996, Aghasi et al., 2016, 2019], phase retrieval [Fienup, 1982, Candès and Li, 2012, Candès et al., 2013], dictionary learning [Tosic and Frossard, 2011], matrix factorization [Hoyer, 2004, Lee and Seung, 2001], and self-calibration [Ling and Strohmer, 2015]. A significant challenge of BIP is the ambiguity of solutions. These ambiguities are challenging because they cause the set of solutions to be non-convex.

A common ambiguity, also shared by the BIP in (1), is the scaling ambiguity. That is any member of the set $\{c\boldsymbol{w}_0, \frac{1}{c}\boldsymbol{x}_0\}$ for $c \neq 0$ solves (1). In addition to the scaling ambiguity, this BIP is difficult to solve because the solutions are non-unique, even when excluding the scaling ambiguity. For example, $(\boldsymbol{w}_0, \boldsymbol{x}_0)$ and $(\mathbf{1}, \boldsymbol{w}_0 \odot \boldsymbol{x}_0)$ both satisfy (1). This structural ambiguity can be solved by assuming a prior model of the unknown vectors. In past works relating to blind deconvolution and blind demodulation [Ahmed et al., 2014, Aghasi et al., 2019], this structural ambiguity issue was addressed by assuming a subspace prior, i.e. the unknown signals belong to known subspaces. Additionally, in many applications, the signals are compressible or sparse with respect to a basis like a wavelet basis or the Discrete Cosine Transform basis, which can address this structural ambiguity issue.

In contrast to subspace and sparsity priors, we address the structural ambiguity issue by assuming the signals $\boldsymbol{w}_0$ and $\boldsymbol{x}_0$ belong to the range of known generative models $\mathcal{G}^{(1)} : \mathbb{R}^n \to \mathbb{R}^\ell$ and $\mathcal{G}^{(2)} : \mathbb{R}^p \to \mathbb{R}^\ell$, respectively. That is, we assume that $\boldsymbol{w}_0 = \mathcal{G}^{(1)}(\boldsymbol{h}_0)$ for some $\boldsymbol{h}_0 \in \mathbb{R}^n$ and $\boldsymbol{x}_0 = \mathcal{G}^{(2)}(\boldsymbol{m}_0)$ for some $\boldsymbol{m}_0 \in \mathbb{R}^p$. So, to recover the unknown vectors $\boldsymbol{w}_0$ and $\boldsymbol{x}_0$, we first recover the latent code variables $\boldsymbol{h}_0$ and $\boldsymbol{m}_0$ and then apply $\mathcal{G}^{(1)}$ and $\mathcal{G}^{(2)}$ on $\boldsymbol{h}_0$ and $\boldsymbol{m}_0$, respectively. Thus, the blind demodulation problem under generative prior we study is:

find $\boldsymbol{h} \in \mathbb{R}^n$ and $\boldsymbol{m} \in \mathbb{R}^p$, up to the scaling ambiguity, such that $\boldsymbol{y}_0 = \mathcal{G}^{(1)}(\boldsymbol{h}) \odot \mathcal{G}^{(2)}(\boldsymbol{m})$.

In recent years, advances in generative modeling of images [Karras et al., 2017] has significantly increased the scope of using a generative model as a prior in inverse problems. Generative models are now used in speech synthesis [van den Oord et al., 2016], image in-painting [Iizuka et al., 2017], image-to-image translation [Zhu et al., 2017], superresolution [Sønderby et al., 2017], compressed sensing [Bora et al., 2017, Lohit et al., 2018], blind deconvolution [Asim et al., 2018], blind pty-chography [Shamshad et al., 2018], and in many more fields. Most of these papers empirically show that using generative model as a prior to solve inverse problems outperform classical methods. For example, in compressed sensing, optimization over the latent code space to recover images from its compressive measurements have been empirically shown to succeed with 10x fewer measurements than classical sparsity based methods [Bora et al., 2017]. Similarly, the authors of Asim et al. [2018] empirically show that using generative priors in image debluring inverse problem provide a very effective regularization that produce sharp deblurred images from very blurry images.

In the present paper, we use generative priors to solve the blind demodulation problem (1). The generative model we consider is the an expansive, fully connected, feed forward neural network with Rectified Linear Unit (ReLU) activation functions and no bias terms. Our main contribution is we show that the empirical risk objective function, for a sufficiently expansive random generative model, has a landscape favorable for gradient based methods to converge to a global minimizer. Our result implies that if the dimension of the unknown signals satisfy $\ell = \Omega(n^2 + p^2)$, up to log factors, then the landscape is favorable. In comparison, classical sparsity based methods for similar BIPs like sparse blind demodulation [Lee et al., 2017] and sparse phase retrieval [Li and Voroninski, 2013] showed that exact recovery of the unknown signals is possible if the number of measurements scale quadratically, up to a log factor, w.r.t. the sparsity level of the signals. While we show a similar scaling of the number of measurements w.r.t. the latent code dimension, the latent code dimension can be smaller than the sparsity level for the same signal, and thus recovering the signal using generative prior would require less number of measurements.

## 1.1 Main results

We study the problem of recovering two unknown signals $\boldsymbol{w}_0$ and $\boldsymbol{x}_0$ in $\mathbb{R}^\ell$ from observations $\boldsymbol{y}_0 = \boldsymbol{w}_0 \odot \boldsymbol{x}_0$, where $\odot$ denotes entrywise product. We assume, as a prior, that the vectors $\boldsymbol{w}_0$ and $\boldsymbol{x}_0$ belong to the range of $d$-layer and $s$-layer neural networks $\mathcal{G}^{(1)} : \mathbb{R}^n \to \mathbb{R}^\ell$ and $\mathcal{G}^{(2)} : \mathbb{R}^p \to \mathbb{R}^\ell$, respectively. The task of recovering $\boldsymbol{w}_0$ and $\boldsymbol{x}_0$ is reduced to finding the latent codes $\boldsymbol{h}_0 \in \mathbb{R}^n$ and $\boldsymbol{m}_0 \in \mathbb{R}^p$ such that $\mathcal{G}^{(1)}(\boldsymbol{h}_0) = \boldsymbol{w}_0$ and $\mathcal{G}^{(2)}(\boldsymbol{m}_0) = \boldsymbol{x}_0$. More precisely, we consider the generative networks modeled by $\mathcal{G}^{(1)}(\boldsymbol{h}) = \mathrm{relu}(\boldsymbol{W}_d^{(1)} \dots \mathrm{relu}(\boldsymbol{W}_2^{(1)}\mathrm{relu}(\boldsymbol{W}_1^{(1)}\boldsymbol{h}))\dots)$ and $\mathcal{G}^{(2)}(\boldsymbol{m}) = \mathrm{relu}(\boldsymbol{W}_s^{(2)} \dots \mathrm{relu}(\boldsymbol{W}_2^{(2)}\mathrm{relu}(\boldsymbol{W}_1^{(2)}\boldsymbol{m}))\dots)$, where $\mathrm{relu}(\boldsymbol{x}) = \max(\boldsymbol{x}, \mathbf{0})$ applies entrywise, $\boldsymbol{W}_i^{(1)} \in \mathbb{R}^{n_i \times n_{i-1}}$ for $i = 1, \dots, d$ with $n = n_0 < n_1 < \dots < n_d = \ell$, and $\boldsymbol{W}_i^{(2)} \in \mathbb{R}^{p_i \times p_{i-1}}$ for $i = 1, \dots, s$ with $p = p_0 < p_1 < \dots < p_s = \ell$. The blind demodulation problem we consider is:

Let: $\boldsymbol{y}_0 \in \mathbb{R}^\ell, \boldsymbol{h}_0 \in \mathbb{R}^n, \boldsymbol{m}_0 \in \mathbb{R}^p$ such that $\boldsymbol{y}_0 = \mathcal{G}^{(1)}(\boldsymbol{h}_0) \odot \mathcal{G}^{(2)}(\boldsymbol{m}_0)$,

Given: $\mathcal{G}^{(1)}, \mathcal{G}^{(2)}$ and measurements $\boldsymbol{y}_0$,

Find: $\boldsymbol{h}_0$ and $\boldsymbol{m}_0$, up to the scaling ambiguity.

In order to recover $\boldsymbol{h}_0$ and $\boldsymbol{m}_0$, up to the scaling ambiguity, we consider the following empirical risk minimization program:

$$\underset{\boldsymbol{h}\in\mathbb{R}^n, \boldsymbol{m}\in\mathbb{R}^p}{\text{minimize}} \; f(\boldsymbol{h}, \boldsymbol{m}) := \frac{1}{2} \left\| \mathcal{G}^{(1)}(\boldsymbol{h}_0) \odot \mathcal{G}^{(2)}(\boldsymbol{m}_0) - \mathcal{G}^{(1)}(\boldsymbol{h}) \odot \mathcal{G}^{(2)}(\boldsymbol{m}) \right\|_2^2. \qquad (2)$$

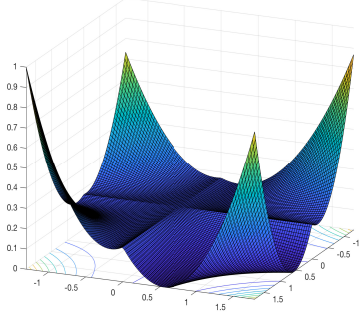

(a) Landscape of the empirical risk function.

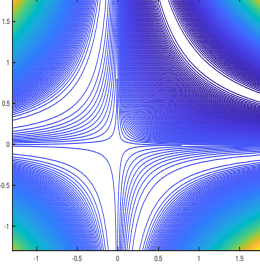

(b) Note the four hyperbolic branches visible.

Figure 1: Plots showing the landscape of the objective function with $\boldsymbol{h}_0 = 1$ and $\boldsymbol{m}_0 = 1$.

Figures 1a and 1b show the landscape of the objective function in the case when $\boldsymbol{h}_0 = \boldsymbol{m}_0 = 1$, $s = d = 2$, the networks are expansive, and the weight matrices $\boldsymbol{W}_i^{(1)}$ and $\boldsymbol{W}_i^{(2)}$ contain i.i.d. Gaussian entries. Clearly, the objective function in (2) is non-convex and, as a result, there does not exist a prior guarantee that gradient based methods will converge to a global minima. Additionally, the objective function does not contain any regularizer which are generally be used to resolve the scaling ambiguity, and thus every point in $\left\{ (c\boldsymbol{h}_0, \frac{1}{c}\boldsymbol{m}_0)|c > 0 \right\}$ is a global optima of (2). Nonetheless, we show that under certain conditions on the networks, the minimizers of (2) are in the neighborhood of four hyperbolic curves, one of which is the hyperbolic curve containing the global minimizers.

In order to define these hyperbolic neighborhoods, let

$$\mathcal{A}_{\epsilon,(\tilde{\boldsymbol{h}},\tilde{\boldsymbol{m}})} = \left\{ (\boldsymbol{h}, \boldsymbol{m}) \in \mathbb{R}^{n \times p} \,\middle|\, \exists\, c > 0 \text{ s.t. } \left\| (\boldsymbol{h}, \boldsymbol{m}) - \left( c\tilde{\boldsymbol{h}}, \frac{1}{c}\tilde{\boldsymbol{m}} \right) \right\|_2 \leq \epsilon \left\| \left( c\tilde{\boldsymbol{h}}, \frac{1}{c}\tilde{\boldsymbol{m}} \right) \right\|_2 \right\}, \qquad (3)$$

where $(\tilde{\boldsymbol{h}}, \tilde{\boldsymbol{m}}) \in \mathbb{R}^{n \times p}$ is fixed. This set is an $\epsilon$-neighborhood of the hyperbolic set $\left\{ (c\tilde{\boldsymbol{h}}, \frac{1}{c}\tilde{\boldsymbol{m}}) | c > 0 \right\}$. We show that the minimizers of (2) are contained in the four hyperbolic sets given by $\mathcal{A}_{\epsilon,(\boldsymbol{h}_0, \boldsymbol{m}_0)}$, $\mathcal{A}_{\epsilon,(-\rho_d^{(1)}\boldsymbol{h}_0, \boldsymbol{m}_0)}$, $\mathcal{A}_{\epsilon,(\boldsymbol{h}_0, -\rho_s^{(2)}\boldsymbol{m}_0)}$, and $\mathcal{A}_{\epsilon,(-\rho_d^{(1)}\boldsymbol{h}_0, -\rho_s^{(2)}\boldsymbol{m}_0)}$. Here, $\epsilon$ depends on the expansivity and number of layers in the networks, and both $\rho_d^{(1)}$ and $\rho_s^{(2)}$ are positive constants close to 1. We also show that the points in the set $\{(\boldsymbol{h}, \boldsymbol{0}) | \boldsymbol{h} \in \mathbb{R}^n\} \cup \{(\boldsymbol{0}, \boldsymbol{m}) | \boldsymbol{m} \in \mathbb{R}^p\}$ are local maximizers. This result holds for networks with the following assumptions:

A1. The weight matrices are random.

A2. The weight matrices of inner layers satisfy $n_i \geq c n_{i-1} \log n_{i-1}$ for $i = 1, \ldots, d-1$ and $p_i \geq c p_{i-1} \log p_{i-1}$ for $i = 1, \ldots, s-1$.

A3. The weight matrices of the last layer for each generator satisfy $\ell \geq c((n_{d-1} \log n_{d-1})^2 + (p_{s-1} \log p_{s-1})^2)$.

In the above assumptions, $c$ is a constant that depends polynomially on the expansivity parameter of $\mathcal{G}^{(1)}$ and $\mathcal{G}^{(2)}$. Figures 1a and 1b show the landscape of the objective function and corroborate our findings. In the paper we provide two deterministic conditions that are sufficient to characterize the landscape of the objective function, and show that Gaussian matrices satisfy these conditions. In essence, we only require approximate Gaussian matrices. We also note that the state-of-the-art literature for provable convergence of the training of neural networks for regression and classification admit proofs only in the case that the final trained weights are close to their random initialization.

Thus, our neural network assumptions are consistent with the best known cases for which networks can be provably trained.

**Theorem 1** (Informal). *Let*

$$\mathcal{A} = \mathcal{A}_{\epsilon,(\boldsymbol{h}_0,\boldsymbol{m}_0)} \cup \mathcal{A}_{\epsilon,(-\rho_d^{(1)}\boldsymbol{h}_0,\boldsymbol{m}_0)} \cup \mathcal{A}_{\epsilon,(\boldsymbol{h}_0,-\rho_s^{(2)}\boldsymbol{m}_0)} \cup \mathcal{A}_{\epsilon,(-\rho_d^{(1)}\boldsymbol{h}_0,-\rho_s^{(2)}\boldsymbol{m}_0)},$$

*where $\epsilon > 0$ depends on the expansivity of our networks and $\rho_d^{(1)}, \rho_s^{(2)} \to 1$ as $d, s \to \infty$, respectively. Suppose the networks are sufficiently expansive such that the number of neurons in the inner layers and the last layers satisfy assumptions A2 and A3, respectively. Then there exist a descent direction, given by one of the one-sided partial derivative of the objective function in (2), for every $(\boldsymbol{h}, \boldsymbol{m}) \notin \mathcal{A} \cup \{(\boldsymbol{h}, \boldsymbol{0})|\boldsymbol{h} \in \mathbb{R}^n\} \cup \{(\boldsymbol{0}, \boldsymbol{m})|\boldsymbol{m} \in \mathbb{R}^p\}$ with high probability. In addition, elements of the set $\{(\boldsymbol{h}, \boldsymbol{0})|\boldsymbol{h} \in \mathbb{R}^n\} \cup \{(\boldsymbol{0}, \boldsymbol{m})|\boldsymbol{m} \in \mathbb{R}^p\}$ are local maximizers.*

Our main result states that the objective function in (2) does not have any spurious minimizers outside of the four hyperbolic neighborhoods. Thus, a gradient descent algorithm will converge to a point inside the four neighborhoods, one of which contains the global minimizers of (2). However, it may not guarantee convergence to a global minimizer and it may not resolve the inherent scaling ambiguity present in the problem. So, in order to converge to a global minimizer, we implement a gradient descent scheme that exploits the landscape of the objective function. That is, we exploit the fact that points near the hyperbolic curve corresponding to the global minimizer have a lower objective value than points that are close to the remaining three spurious hyperbolic curves. Second, in order to resolve the scaling ambiguity, we promote solutions that have equal $\ell_2$ norm by normalizing the estimates in each iteration of the gradient descent scheme (See Section 2). In principle, a convergence result to a global minimizer by gradient descent is possible, and would require showing a convexity-like property around the hyperbola. We leave this for possible future work.

Theorem 1 also provides a global guarantee of the landscape of the objective function in (2) if the dimension of the unknown signals scale quadratically w.r.t. to the dimension of the latent codes, i.e. $\ell = \Omega(n^2 + p^2)$, up to log factors. Our result, which we get by enforcing generative priors may enjoy better sample complexity than classical priors like sparsity because: i) existing recovery guarantee of unstructured signals require number of measurements that scale quadratically with the sparsity level, and ii) a signal can have a latent code dimension with respect to a GAN that is smaller than its sparsity level with respect to a wavelet basis. For example, consider a set of images that correspond to a single train going down a single track. This set of images form a one dimensional sub-manifold of the manifold of natural images. If properly parameterized by a generative model, then it would have a latent dimensionality of approximately 1, whereas the number of wavelet coefficients needed to describe any of those images is much greater. The work in Bora et al. [2017] shows that compressed sensing can be done with 5-10x fewer measurements than sparsity models. This provides evidence for the more economical representation of generative models than of sparsity models. Additionally, it is more natural to view the natural signal manifold as a low-dimensional manifold, as opposed to being the combinatorially-many union of low dimensional spaces. Performance gains are provided by the fact that the natural signal manifold can be directly exploited, whereas the union of subspaces can only be indirectly exploited via convex relaxations. Thus, our result may be less limiting in terms of sample complexity.

## 1.2 Prior work on problems related to blind demodulation

A common approach of solving the BIP in (1) is to assume a subspace or sparsity prior on the unknown vectors. In these cases the unknown vectors $\boldsymbol{w}_0$ and $\boldsymbol{x}_0$ are assumed to be in the range of known matrices $\boldsymbol{B} \in \mathbb{R}^{\ell \times n}$ and $\boldsymbol{C} \in \mathbb{R}^{\ell \times p}$, respectively. In Ahmed et al. [2014], the authors assumed a subspace prior and cast the BIP as a linear low rank matrix recovery problem. They introduced a semidefinite program based on nuclear norm minimization to recover the unknown matrix. For the case where the rows of $\boldsymbol{B}$ and $\boldsymbol{C}$ are Fourier and Gaussian vectors, respectively, they provide a recovery guarantee that depend on the number of measurements as $\ell = \Omega(n + p)$, up to log factors. However, because this method operates in the space of matrices, it is computationally prohibitively expensive. Another limitation of the lifted approach is that recovering a low rank and sparse matrix efficiently from linear observation of the matrix has been challenging. Recently, Lee et al. [2017] provided a recovery guarantee with near optimal sample complexity for the low rank and sparse matrix recovery problem using an alternating minimization method for a class of signals that satisfy a peakiness condition. However, for general signals the same work established a recovery result for the case where the number of measurements scale quadratically with the sparsity level.

In order to address the computational cost of working in the lifted case, a recent theme has been to introduce convex and non-convex programs that work in the natural parameter space. For example, in Bahmani and Romberg [2016], Goldstein and Studer [2016], the authors introduced PhaseMax, which is a convex program for phase retrieval that is based on finding a simple convex relaxation via the convex hull of the feasibility set. The authors showed that PhaseMax enjoys rigorous recovery guarantee if a good anchor is available. This formulation was extended to the sparse case in Hand and Voroninski [2016], where the authors considered SparsePhaseMax and provided a recovery guarantee with optimal sample complexity. The idea of formulating a convex program using a simple convex relaxation via the convex hull of the feasibility set was used in the blind demodulation problem as well [Aghasi et al., 2019, 2018]. In particular, Aghasi et al. [2018] introduced a convex program in the natural parameter space for the sparse blind demodulation problem in the case where the sign of the unknown signals are known. Like in Lee et al. [2017], the authors in Aghasi et al. [2019] provide a recovery guarantee with optimal sample complexity for a class of signals. However, the result does not extend to signals with no constraints. Other approaches that operate in the natural parameter space are methods based on Wirtinger Flow. For example, in Candès et al. [2015], Wang et al. [2016], Li et al. [2016], the authors use Wirtinger Flow and its variants to solve the phase retrieval and the blind deconvolution problem. These methods are non-convex and require a good initialization to converge to a global solution. However, they are simple to solve and enjoys rigorous recovery guarantees.

## 1.3    Other related work

In this paper, we consider the blind demodulation problem with the unknown signals assumed to be in the range of known generative models. Our work is motivated by experimental results in deep compressed sensing and deep blind deconvolution presented in Bora et al. [2017], Asim et al. [2018] and theoretical work in deep compressed sensing presented in Hand and Voroninski [2017]. In Bora et al. [2017], the authors consider the compressed sensing problem where, instead of a sparsity prior, a generative prior is considered. They used an empirical risk optimization program over the latent code space to recover images and empirical showed that their method succeeds with 10x fewer measurements than previous sparsity based methods. Following the empirical successes of deep compressed sensing, the authors in Hand and Voroninski [2017] provided a theoretical understanding for these successes by characterizing the landscape of the empirical risk objective function. In the random case with the layers of the generative model sufficiently expansive, they showed that every point outside of a small neighborhood around the true solution and a negative multiple of the true solution has a descent direction with high probability. Another instance where generative model currently outperforms sparsity based methods is in sparse phase retrieval Hand et al. [2018]. In sparse phase retrieval, current algorithms that enjoy a provable recovery guarantee of an unknown $n$-dimensional $k$-sparse signal require at least $O(k^2 \log n)$ measurements; whereas, when assuming the unknown signal is an output of a known $d$-layer generator $\mathcal{G} : \mathbb{R}^k \to \mathbb{R}^n$, the authors in Hand et al. [2018] showed that, under favorable conditions on the generator and with at least $O(kd^2 \log n)$ measurements, the empirical risk objective enjoys a favorable landscape.

Similarly, in Asim et al. [2018], the authors consider the blind deconvolution problem where a generative prior over the unknown signal is considered. They empirically showed that using generative priors in the image deblurring inverse problem provide a very effective regularization that produce sharp deblurred images from very blurry images. The algorithm used to recovery these deblurred images is an alternating minimization approach which solves the empirical risk minimization with $\ell_2$ regularization on the unknown signals. The $\ell_2$ regularization promotes solution with least $\ell_2$ norm and resolves the scaling ambiguity present in the blind deconvolution problem. We consider a related problem, namely the blind demodulation problem with a generative prior on the unknown signals, and show that under certain conditions on the generators the empirical risk objective has a favorable landscape.

## 1.4    Notations

Vectors and matrices are written with boldface, while scalars and entries of vectors are written in plain font. We write $\mathbf{1}$ as the vector of all ones with dimensionality appropriate for the context. Let $\mathbb{S}^{n-1}$ be the unit sphere in $\mathbb{R}^n$. We write $\boldsymbol{I}_n$ as the $n \times n$ identity matrix. For $\boldsymbol{x} \in \mathbb{R}^K$ and $\boldsymbol{y} \in \mathbb{R}^N$, $(\boldsymbol{x}, \boldsymbol{y})$ is the corresponding vector in $\mathbb{R}^K \times \mathbb{R}^N$. Let $\mathrm{relu}(\boldsymbol{x}) = \max(\boldsymbol{x}, \mathbf{0})$ apply entrywise for $\boldsymbol{x} \in \mathbb{R}^n$. Let $\mathrm{diag}(\boldsymbol{W}\boldsymbol{x} > 0)$ be the diagonal matrix that is 1 in the $(i, i)$ entry if $(\boldsymbol{W}\boldsymbol{x})_i > 0$ and 0 otherwise. Let $\boldsymbol{A} \preceq \boldsymbol{B}$ mean that $\boldsymbol{B} - \boldsymbol{A}$ is a positive semidefinite matrix. We will write $\gamma = O(\delta)$ to mean that

there exists a positive constant $C$ such that $\gamma \leq C\delta$, where $\gamma$ is understood to be positive. Similarly we will write $c = \Omega(\delta)$ to mean that there exists a positive constant $C$ such that $c \geq C\delta$. When we say that a constant depends polynomially on $\epsilon^{-1}$, that means that it is at most $C\epsilon^{-k}$ for some positive $C$ and positive integer $k$. For notational convenience, we will write $\boldsymbol{a} = \boldsymbol{b} + O_1(\epsilon)$ if $\|\boldsymbol{a} - \boldsymbol{b}\| \leq \epsilon$, where the norm is understood to be absolute value for scalars, the $\ell_2$ norm for vectors, and the spectral norm for matrices.

## 2 Algorithm

In this section, we propose a gradient descent scheme that solves (2). The gradient descent scheme exploits the global geometry present in the landscape of the objective function in (2) and avoids regions containing spurious minimizers. The gradient descent scheme is based on two observations. The first observation is that the minimizers of (2) are close to four hyperbolic curves given by $\{(c\boldsymbol{h}_0, \frac{1}{c}\boldsymbol{m}_0)|c > 0\}$, $\{(-c\rho_d^{(1)}\boldsymbol{h}_0, \frac{1}{c}\boldsymbol{m}_0)|c > 0\}$, $\{(c\boldsymbol{h}_0, -\frac{\rho_s^{(2)}}{c}\boldsymbol{m}_0)|c > 0\}$, and $\{(-c\rho_d^{(1)}\boldsymbol{h}_0, -\frac{\rho_s^{(2)}}{c}\boldsymbol{m}_0)|c > 0\}$, where $\rho_d^{(1)}$ and $\rho_s^{(2)}$ are close to 1. The second observation is that $f(c\boldsymbol{h}_0, \frac{1}{c}\boldsymbol{m}_0)$ is less than $f(-c\boldsymbol{h}_0, \frac{1}{c}\boldsymbol{m}_0)$, $f(c\boldsymbol{h}_0, -\frac{1}{c}\boldsymbol{m}_0)$, and $f(-c\boldsymbol{h}_0, -\frac{1}{c}\boldsymbol{m}_0)$ for any $c > 0$. This is because the curve $\{(c\boldsymbol{h}_0, \frac{1}{c}\boldsymbol{m}_0)|c > 0\}$ corresponds to the global minimizer of (2).

We now introduce some quantities which are useful in stating the gradient descent algorithm. For any $\boldsymbol{h} \in \mathbb{R}^n$ and $\boldsymbol{W} \in \mathbb{R}^{l \times n}$, define $\boldsymbol{W}_{+,\boldsymbol{h}} = \text{diag}(\boldsymbol{W}\boldsymbol{h} > 0)\boldsymbol{W}$. That is, $\boldsymbol{W}_{+,\boldsymbol{h}}$ zeros out the rows of $\boldsymbol{W}$ that do not have a positive inner product with $\boldsymbol{h}$ and keeps the remaining rows. We will extend the definition of $\boldsymbol{W}_{+,\boldsymbol{h}}$ to each layer of weights $\boldsymbol{W}_i^{(1)}$ in our neural network. For $\boldsymbol{W}_i^{(1)} \in \mathbb{R}^{n_1 \times n}$ and $\boldsymbol{h} \in \mathbb{R}^n$, define $\boldsymbol{W}_{1,+,\boldsymbol{h}}^{(1)} := (\boldsymbol{W}_1^{(1)})_{+,\boldsymbol{h}} = \text{diag}(\boldsymbol{W}_1^{(1)}\boldsymbol{h} > 0)\boldsymbol{W}_1^{(1)}$. For each layer i > 1, define

$$\boldsymbol{W}_{i,+,\boldsymbol{h}}^{(1)} = \text{diag}(\boldsymbol{W}_i^{(1)}\boldsymbol{W}_{i-1,+,\boldsymbol{h}}^{(1)} \cdots \boldsymbol{W}_{2,+,\boldsymbol{h}}^{(1)}\boldsymbol{W}_{1,+,\boldsymbol{h}}^{(1)}\boldsymbol{h} > 0)\boldsymbol{W}_i^{(1)}.$$

Lastly, define $\boldsymbol{\Lambda}_{d,+,\boldsymbol{h}}^{(1)} := \prod_{i=d}^{1} \boldsymbol{W}_{i,+,\boldsymbol{h}}^{(1)}$. Using the above notation, $\mathcal{G}^{(1)}(\boldsymbol{h})$ can be compactly written as $\boldsymbol{\Lambda}_{d,+,\boldsymbol{h}}^{(1)}\boldsymbol{h}$. Similarly, we may write $\mathcal{G}^{(2)}(\boldsymbol{m})$ compactly as $\boldsymbol{\Lambda}_{s,+,\boldsymbol{h}}^{(2)}\boldsymbol{m}$.

The gradient descent scheme is an alternating descent direction algorithm. We first pick an initial iterate $(\boldsymbol{h}_1, \boldsymbol{m}_1)$ such that $\boldsymbol{h}_1 \neq \boldsymbol{0}$ and $\boldsymbol{m}_1 \neq \boldsymbol{0}$. At each iteration $i = 1, 2, \ldots$, we first compare the objective value at $(\boldsymbol{h}_1, \boldsymbol{m}_1)$, $(-\boldsymbol{h}_1, \boldsymbol{m}_1)$, $(\boldsymbol{h}_1, -\boldsymbol{m}_1)$, and $(-\boldsymbol{h}_1, -\boldsymbol{m}_1)$ and reset $(\boldsymbol{h}_1, \boldsymbol{m}_1)$ to be the point with least objective value. Second we descend along a direction. We compute the descent direction $\tilde{\boldsymbol{g}}_{1,(\boldsymbol{h},\boldsymbol{m})}$, given by the partial derivative of $f$ in (2) w.r.t. $\boldsymbol{h}$,

$$\tilde{\boldsymbol{g}}_{1,(\boldsymbol{h},\boldsymbol{m})} := \boldsymbol{\Lambda}_{d,+,\boldsymbol{h}}^{(1)}{}^{\mathsf{T}}\left(\text{diag}(\boldsymbol{\Lambda}_{s,+,\boldsymbol{m}}^{(2)}\boldsymbol{m})^2\boldsymbol{\Lambda}_{d,+,\boldsymbol{h}}^{(1)}\boldsymbol{h} - \text{diag}(\boldsymbol{\Lambda}_{s,+,\boldsymbol{m}}^{(2)}\boldsymbol{m} \odot \boldsymbol{\Lambda}_{s,+,\boldsymbol{m}_0}^{(2)}\boldsymbol{m}_0)\boldsymbol{\Lambda}_{d,+,\boldsymbol{h}_0}^{(1)}\boldsymbol{h}_0\right)$$

and take a step along this direction. Next, we compute the descent direction $\tilde{\boldsymbol{g}}_{2,(\boldsymbol{h},\boldsymbol{m})}$, given by the partial derivative of $f$ in w.r.t. $\boldsymbol{m}$,

$$\tilde{\boldsymbol{g}}_{2,(\boldsymbol{h},\boldsymbol{m})} := \boldsymbol{\Lambda}_{s,+,\boldsymbol{m}}^{(2)}{}^{\mathsf{T}}\left(\text{diag}(\boldsymbol{\Lambda}_{d,+,\boldsymbol{h}}^{(1)}\boldsymbol{h})^2\boldsymbol{\Lambda}_{s,+,\boldsymbol{m}}^{(2)}\boldsymbol{m} - \text{diag}(\boldsymbol{\Lambda}_{d,+,\boldsymbol{h}}^{(1)}\boldsymbol{h} \odot \boldsymbol{\Lambda}_{d,+,\boldsymbol{h}_0}^{(1)}\boldsymbol{h}_0)\boldsymbol{\Lambda}_{s,+,\boldsymbol{m}_0}^{(2)}\boldsymbol{m}_0\right).$$

and again take a step along this direction. Lastly, we normalize the iterate so that at each iteration $i$ $\|\boldsymbol{h}_i\|_2 = \|\boldsymbol{m}_i\|_2$. We repeat this process until convergence. Algorithm 1 outlines this process.

---

**Algorithm 1** Alternating descent algorithm for (2)

Input: Weight matrices, $\boldsymbol{W}_i^{(1)}$ and $\boldsymbol{W}_i^{(2)}$, observation $\boldsymbol{y}_0$ and step size $\eta > 0$.
Output: An estimate of a global minimizer of (2)
1: Choose an arbitrary point $(\boldsymbol{h}_1, \boldsymbol{m}_1)$ such that $\boldsymbol{h}_1 \neq \boldsymbol{0}$ and $\boldsymbol{m}_1 \neq \boldsymbol{0}$
2: **for** $i = 1, 2, \ldots$ **do**:
3:      $(\boldsymbol{h}_i, \boldsymbol{m}_i) \leftarrow \arg\min(f(\boldsymbol{h}_i, \boldsymbol{m}_i), f(-\boldsymbol{h}_i, \boldsymbol{m}_i), f(\boldsymbol{h}_i, -\boldsymbol{m}_i), f(-\boldsymbol{h}_i, -\boldsymbol{m}_i))$
4:      $\boldsymbol{h}_{i+1} \leftarrow \boldsymbol{h}_i - \eta\tilde{\boldsymbol{g}}_{1,(\boldsymbol{h}_i,\boldsymbol{m}_i)}, \quad \boldsymbol{m}_{i+1} \leftarrow \boldsymbol{m}_i - \eta\tilde{\boldsymbol{g}}_{2,(\boldsymbol{h}_{i+1},\boldsymbol{m}_i)}$
5:      $c \leftarrow \sqrt{\|\boldsymbol{h}_{i+1}\|_2/\|\boldsymbol{m}_{i+1}\|_2}, \quad \boldsymbol{h}_{i+1} \leftarrow \boldsymbol{h}_{i+1}/c, \quad \boldsymbol{m}_{i+1} \leftarrow \boldsymbol{m}_{i+1} \cdot c$
6: **end for**

---

# 3 Proof Outline

We now present our main results which states that the objective function has a descent direction at every point outside of four hyperbolic regions. In order to state these directions, we first note that the partial derivatives of $f$ at a differentiable point $(\boldsymbol{h}, \boldsymbol{m})$ are

$$\nabla_{\boldsymbol{h}} f(\boldsymbol{h}, \boldsymbol{m}) = \tilde{\boldsymbol{g}}_{1,(\boldsymbol{h}, \boldsymbol{m})} \text{ and } \nabla_{\boldsymbol{m}} f(\boldsymbol{h}, \boldsymbol{m}) = \tilde{\boldsymbol{g}}_{2,(\boldsymbol{h}, \boldsymbol{m})}.$$

The function $f$ is not differentiable everywhere because of the behavior of the RELU activation function in the neural network. However, since $\mathcal{G}^{(1)}$ and $\mathcal{G}^{(2)}$ are piecewise linear, $f$ is differentiable at $(\boldsymbol{h}, \boldsymbol{m}) + \delta \boldsymbol{w}$ for all $(\boldsymbol{h}, \boldsymbol{m})$ and $\boldsymbol{w}$ and sufficiently small $\delta$. The directions we consider are $g_{1,(\boldsymbol{h}, \boldsymbol{m})} \in \mathbb{R}^{n+p}$ and $g_{2,(\boldsymbol{h}, \boldsymbol{m})} \in \mathbb{R}^{n+p}$, where

$$\boldsymbol{g}_{1,(\boldsymbol{h}, \boldsymbol{m})} = \begin{bmatrix} \lim_{\delta \to 0^+} \nabla_{\boldsymbol{h}} f((\boldsymbol{h}, \boldsymbol{m}) + \delta \boldsymbol{w}) \\ \boldsymbol{0} \end{bmatrix}, \boldsymbol{g}_{2,(\boldsymbol{h}, \boldsymbol{m})} = \begin{bmatrix} \boldsymbol{0} \\ \lim_{\delta \to 0^+} \nabla_{\boldsymbol{m}} f((\boldsymbol{h}, \boldsymbol{m}) + \delta \boldsymbol{w}) \end{bmatrix}, \text{ and} \tag{4}$$

$\boldsymbol{w}$ is fixed. Let $D_{\boldsymbol{g}} f(\boldsymbol{h}, \boldsymbol{m})$ be the unnormalized one-sided directional derivative of $f(\boldsymbol{h}, \boldsymbol{m})$ in the direction of $\boldsymbol{g}$: $D_{\boldsymbol{g}} f(\boldsymbol{h}, \boldsymbol{m}) = \lim_{t \to 0^+} \frac{f((\boldsymbol{h}, \boldsymbol{m}) + t\boldsymbol{g}) - f(\boldsymbol{h}, \boldsymbol{m})}{t}$.

**Theorem 2.** *Fix $\epsilon > 0$ such that $K_1(d^7 s^2 + d^2 s^7) \epsilon^{1/4} < 1$, $d \geq 2$, and $s \geq 2$. Assume the networks satisfy assumptions A2 and A3. Assume $\boldsymbol{W}_i^{(1)} \sim \mathcal{N}(\boldsymbol{0}, \frac{1}{n_i} \boldsymbol{I}_{n_{i-1}})$ for $i = 1, \ldots, d-1$ and $i$th row of $\boldsymbol{W}_d^{(1)}$ satisfies $(\boldsymbol{w}_d^{(1)})_i^\mathsf{T} = \boldsymbol{w}^\mathsf{T} \cdot \boldsymbol{1}_{\|\boldsymbol{w}\|_2 \leq 3\sqrt{n_{d-1}/\ell}}$ with $\boldsymbol{w} \sim \mathcal{N}(\boldsymbol{0}, \frac{1}{\ell} \boldsymbol{I}_{n_{d-1}})$. Similarly, assume $\boldsymbol{W}_i^{(2)} \sim \mathcal{N}(\boldsymbol{0}, \frac{1}{p_i} \boldsymbol{I}_{p_{i-1}})$ for $i = 1, \ldots, s-1$ and $i$th row of $\boldsymbol{W}_s^{(2)}$ satisfies $(\boldsymbol{w}_s^{(2)})_i^\mathsf{T} = \boldsymbol{w}^\mathsf{T} \cdot \boldsymbol{1}_{\|\boldsymbol{w}\|_2 \leq 3\sqrt{p_{s-1}/\ell}}$ with $\boldsymbol{w} \sim \mathcal{N}(\boldsymbol{0}, \frac{1}{\ell} \boldsymbol{I}_{p_{s-1}})$. Let $\mathcal{K} = \{(\boldsymbol{h}, \boldsymbol{0}) \in \mathbb{R}^{n \times p} \, | \boldsymbol{h} \in \mathbb{R}^n \} \cup \{(\boldsymbol{0}, \boldsymbol{m}) \in \mathbb{R}^{n \times p} \, | \boldsymbol{m} \in \mathbb{R}^p \}$ and $\mathcal{A} = \mathcal{A}_{K_2 d^3 s^3 \epsilon^{\frac{1}{4}}, (\boldsymbol{h}_0, \boldsymbol{m}_0)} \cup \mathcal{A}_{K_2 d^8 s^3 \epsilon^{\frac{1}{4}}, \left(-\rho_d^{(1)} \boldsymbol{h}_0, \boldsymbol{m}_0\right)} \cup \mathcal{A}_{K_2 d^3 s^8 \epsilon^{\frac{1}{4}}, \left(\rho_s^{(2)} \boldsymbol{h}_0, -\boldsymbol{m}_0\right)} \cup \mathcal{A}_{K_2 d^8 s^8 \epsilon^{\frac{1}{4}}, \left(-\rho_d^{(1)} \rho_s^{(2)} \boldsymbol{h}_0, -\boldsymbol{m}_0\right)}$. Then on an event of probability at least $1 - \sum_{i=1}^d \tilde{c} n_i e^{-\gamma n_{i-1}} - \sum_{i=1}^s \tilde{c} p_i e^{-\gamma p_{i-1}} - \tilde{c} e^{-\gamma \ell / (n_{d-1} \log n_{d-1} + p_{s-1} \log p_{s-1})}$ we have the following: for $(\boldsymbol{h}_0, \boldsymbol{m}_0) \neq (\boldsymbol{0}, \boldsymbol{0})$, and*

$$(\boldsymbol{h}, \boldsymbol{m}) \notin \mathcal{A} \cup \mathcal{K}$$

*the one-sided directional derivative of $f$ in the direction of $\boldsymbol{g} = \boldsymbol{g}_{1,(\boldsymbol{h}, \boldsymbol{m})}$ or $\boldsymbol{g} = \boldsymbol{g}_{2,(\boldsymbol{h}, \boldsymbol{m})}$, defined in (4), satisfy $D_{-\boldsymbol{g}} f(\boldsymbol{h}, \boldsymbol{m}) < 0$. Additionally, elements of the set $\mathcal{K}$ are local maximizers. Here, $\rho_d^{(k)}$ are positive numbers that converge to 1 as $d \to \infty$, $c$ and $\gamma^{-1}$ are constants that depend polynomially on $\epsilon^{-1}$ and $\tilde{c}$, $K_1$, and $K_2$ are absolute constants.*

We prove Theorem 2 by showing that neural networks with random weights satisfy two deterministic conditions. These conditions are the Weight Distributed Condition (WDC) and the joint Weight Distributed Condition (joint-WDC). The WDC is a slight generalization of the WDC introduced in Hand and Voroninski [2017]. We say a matrix $\boldsymbol{W} \in \mathbb{R}^{\ell \times n}$ satisfies the WDC with constants $\epsilon > 0$ and $0 < \alpha \leq 1$ if for all nonzero $\boldsymbol{x}, \boldsymbol{y} \in \mathbb{R}^k$,

$$\left\| \sum_{i=1}^\ell \boldsymbol{1}_{\boldsymbol{w}_i \cdot \boldsymbol{x} > 0} \boldsymbol{1}_{\boldsymbol{w}_i \cdot \boldsymbol{y} > 0} \cdot \boldsymbol{w}_i \boldsymbol{w}_i^\mathsf{T} - \alpha \boldsymbol{Q}_{\boldsymbol{x}, \boldsymbol{y}} \right\| \leq \epsilon, \text{ with } \boldsymbol{Q}_{\boldsymbol{x}, \boldsymbol{y}} = \frac{\pi - \theta_0}{2\pi} \boldsymbol{I}_n + \frac{\sin \theta_0}{2\pi} \boldsymbol{M}_{\hat{\boldsymbol{x}} \leftrightarrow \hat{\boldsymbol{y}}}, \tag{5}$$

where $\boldsymbol{w}_i \in \mathbb{R}^n$ is the $i$th row of $\boldsymbol{W}$; $\boldsymbol{M}_{\hat{\boldsymbol{x}} \leftrightarrow \hat{\boldsymbol{y}}} \in \mathbb{R}^{n \times n}$ is the matrix such that $\hat{\boldsymbol{x}} \to \hat{\boldsymbol{y}}, \hat{\boldsymbol{y}} \to \hat{\boldsymbol{x}}$, and $\boldsymbol{z} \to \boldsymbol{0}$ for all $\boldsymbol{z} \in \text{span}(\{\boldsymbol{x}, \boldsymbol{y}\})^\perp$; $\hat{\boldsymbol{x}} = \boldsymbol{x}/\|\boldsymbol{x}\|_2$ and $\hat{\boldsymbol{y}} = \boldsymbol{y}/\|\boldsymbol{y}\|_2$; $\theta_0 = \angle(\boldsymbol{x}, \boldsymbol{y})$; and $\boldsymbol{1}_S$ is the indicator function on $S$. If $\boldsymbol{w}_i \sim \mathcal{N}(\boldsymbol{0}, \frac{1}{\ell} \boldsymbol{I}_n)$ for all $i$, then an elementary calculation shows that $\mathbb{E} \left[ \sum_{i=1}^\ell \boldsymbol{1}_{\boldsymbol{w}_i \cdot \boldsymbol{x} > 0} \boldsymbol{1}_{\boldsymbol{w}_i \cdot \boldsymbol{y} > 0} \cdot \boldsymbol{w}_i \boldsymbol{w}_i^\mathsf{T} \right] = \boldsymbol{Q}_{\boldsymbol{x}, \boldsymbol{y}}$ and if $\boldsymbol{x} = \boldsymbol{y}$ then $\boldsymbol{Q}_{\boldsymbol{x}, \boldsymbol{y}}$ is an isometry up to a factor of $1/2$. Also, note that if $\boldsymbol{W}$ satisfies WDC with constants $\epsilon$ and $\alpha$, then $\frac{1}{\sqrt{\alpha}} \boldsymbol{W}$ satisfies WDC with constants $\epsilon/\alpha$ and 1.

We now state the joint Weight Distributed Condition. We say that $\boldsymbol{B} \in \mathbb{R}^{\ell \times n}$ and $\boldsymbol{C} \in \mathbb{R}^{\ell \times p}$ satisfy joint-WDC with constants $\epsilon > 0$ and $0 < \alpha \leq 1$ if for all nonzero $\boldsymbol{h}, \boldsymbol{x} \in \mathbb{R}^n$ and nonzero $\boldsymbol{m}, \boldsymbol{y} \in \mathbb{R}^p$,

$$\left\| \boldsymbol{B}_{+,\boldsymbol{h}}^\mathsf{T} \text{diag} \left( \boldsymbol{C}_{+,\boldsymbol{m}} \boldsymbol{m} \odot \boldsymbol{C}_{+,\boldsymbol{y}} \boldsymbol{y} \right) \boldsymbol{B}_{+,\boldsymbol{x}} - \frac{\alpha}{\ell} \boldsymbol{m}^\mathsf{T} \boldsymbol{Q}_{\boldsymbol{m}, \boldsymbol{y}} \boldsymbol{y} \cdot \boldsymbol{Q}_{\boldsymbol{h}, \boldsymbol{x}} \right\| \leq \frac{\epsilon}{\ell} \|\boldsymbol{m}\|_2 \|\boldsymbol{y}\|_2, \text{ and} \tag{6}$$

$$\left\| \boldsymbol{C}_{+,\boldsymbol{m}}^{\mathsf{T}}\mathrm{diag}\left(\boldsymbol{B}_{+,\boldsymbol{h}}\boldsymbol{h}\odot\boldsymbol{B}_{+,\boldsymbol{x}}\boldsymbol{x}\right)\boldsymbol{C}_{+,\boldsymbol{y}}-\frac{\alpha}{\ell}\boldsymbol{h}^{\mathsf{T}}\boldsymbol{Q}_{\boldsymbol{h},\boldsymbol{x}}\boldsymbol{x}\cdot\boldsymbol{Q}_{\boldsymbol{m},\boldsymbol{y}}\right\| \leq \frac{\epsilon}{\ell}\|\boldsymbol{h}\|_2\|\boldsymbol{x}\|_2 \tag{7}$$

We analyze networks $\mathcal{G}^{(1)}$ and $\mathcal{G}^{(2)}$ where the weight matrices corresponding to the inner layers satisfy the WDC with constants $\epsilon > 0$ and 1 and for the two matrices corresponding to the outer layers, we assume that one of them satisfies WDC with constants $\epsilon$ and $0 < \alpha_1 \leq 1$ and the other satifies WDC with constants $\epsilon$ and $0 < \alpha_2 \leq 1$. We also assume that the two outer layer matrices satisfy joint-WDC with constants $\epsilon > 0$ and $\alpha = \alpha_1 \cdot \alpha_2$. We now state the main deterministic result:

**Theorem 3.** *Fix $\epsilon > 0$, $0 < \alpha_1 \leq 1$ and $0 < \alpha_2 \leq 1$ such that $K_1(d^7 s^2 + d^2 s^7)\epsilon^{1/4}/(\alpha_1\alpha_2) < 1$, $d \geq 2$, and $s \geq 2$. Let $\mathcal{K} = \{(\boldsymbol{h},\boldsymbol{0}) \in \mathbb{R}^{n\times p} \,|\, \boldsymbol{h} \in \mathbb{R}^n\} \cup \{(\boldsymbol{0},\boldsymbol{m}) \in \mathbb{R}^{n\times p} \,|\, \boldsymbol{m} \in \mathbb{R}^p\}$. Suppose that $\boldsymbol{W}_i^{(1)} \in \mathbb{R}^{n_i \times n_{i-1}}$ for $i = 1,\dots,d-1$ and $\boldsymbol{W}_i^{(2)} \in \mathbb{R}^{p_i \times p_{i-1}}$ for $i = 1,\dots,s-1$ satisfy the WDC with constant $\epsilon$ and 1. Suppose $\boldsymbol{W}_d^{(1)} \in \mathbb{R}^{\ell \times n_{d-1}}$ satisfy WDC with constants $\epsilon$ and $\alpha_1$, and $\boldsymbol{W}_s^{(2)} \in \mathbb{R}^{\ell \times p_{s-1}}$ satisfy WDC with constants $\epsilon$ and $\alpha_2$. Also, suppose $\left(\boldsymbol{W}_d^{(1)}, \boldsymbol{W}_s^{(2)}\right)$ satisfy joint-WDC with constants $\epsilon$, $\alpha = \alpha_1 \cdot \alpha_2$. Let $\mathcal{K} = \{(\boldsymbol{h},\boldsymbol{0}) \in \mathbb{R}^{n\times p} \,|\, \boldsymbol{h} \in \mathbb{R}^n\} \cup \{(\boldsymbol{0},\boldsymbol{m}) \in \mathbb{R}^{n\times p} \,|\, \boldsymbol{m} \in \mathbb{R}^p\}$ and $\mathcal{A} = \mathcal{A}_{K_2 d^3 s^3 \epsilon^{\frac{1}{4}}\alpha^{-1},(\boldsymbol{h}_0,\boldsymbol{m}_0)} \cup \mathcal{A}_{K_2 d^8 s^3 \epsilon^{\frac{1}{4}}\alpha^{-1},\left(-\rho_d^{(1)}\boldsymbol{h}_0,\boldsymbol{m}_0\right)} \cup \mathcal{A}_{K_2 d^3 s^8 \epsilon^{\frac{1}{4}}\alpha^{-1},\left(\rho_s^{(2)}\boldsymbol{h}_0,-\boldsymbol{m}_0\right)} \cup \mathcal{A}_{K_2 d^8 s^8 \epsilon^{\frac{1}{4}}\alpha^{-1},\left(-\rho_d^{(1)}\rho_s^{(2)}\boldsymbol{h}_0,-\boldsymbol{m}_0\right)}$. Then, for $(\boldsymbol{h}_0,\boldsymbol{m}_0) \neq (\boldsymbol{0},\boldsymbol{0})$, and*

$$(\boldsymbol{h},\boldsymbol{m}) \notin \mathcal{A} \cup \mathcal{K}$$

*the one-sided directional derivative of $f$ in the direction of $\boldsymbol{g} = \boldsymbol{g}_{1,(\boldsymbol{h},\boldsymbol{m})}$ or $\boldsymbol{g} = \boldsymbol{g}_{2,(\boldsymbol{h},\boldsymbol{m})}$ satisfy $D_{-\boldsymbol{g}}f(\boldsymbol{h},\boldsymbol{m}) < 0$. Additionally, elements of the set $\mathcal{K}$ are local maximizers. Here, $\rho_d^{(k)}$ are positive numbers that converge to 1 as $d \to \infty$, and $K_1$, and $K_2$ are absolute constants.*

We prove the theorems by showing that the descent directions $\boldsymbol{g}_{1,(\boldsymbol{h},\boldsymbol{m})}$ and $\boldsymbol{g}_{2,(\boldsymbol{h},\boldsymbol{m})}$ concentrate around its expectation and then characterize the set of points where the corresponding expectations are simultaneously zero. The outline of the proof is:

- The WDC and joint-WDC imply that the one-sided partial directional derivatives of $f$ concentrate uniformly for all non-zero $\boldsymbol{h}, \boldsymbol{h}_0 \in \mathbb{R}^n$ and $\boldsymbol{m}, \boldsymbol{m}_0 \in \mathbb{R}^p$ around continuous vectors $\boldsymbol{t}_{(\boldsymbol{h},\boldsymbol{m}),(\boldsymbol{h}_0,\boldsymbol{m}_0)}^{(1)}$ and $\boldsymbol{t}_{(\boldsymbol{h},\boldsymbol{m}),(\boldsymbol{h}_0,\boldsymbol{m}_0)}^{(2)}$, respectively, defined in equations (10) and (11) in the Appendix.

- Direct analysis show that $\boldsymbol{t}_{(\boldsymbol{h},\boldsymbol{m}),(\boldsymbol{h}_0,\boldsymbol{m}_0)}^{(1)}$ and $\boldsymbol{t}_{(\boldsymbol{h},\boldsymbol{m}),(\boldsymbol{h}_0,\boldsymbol{m}_0)}^{(2)}$ are simultaneously approximately zero around the four hyperbolic sets $\mathcal{A}_{\epsilon,(\boldsymbol{h}_0,\boldsymbol{m}_0)}$, $\mathcal{A}_{\epsilon,(-\rho_d^{(1)}\boldsymbol{h}_0,\boldsymbol{m}_0)}$, $\mathcal{A}_{\epsilon,(\boldsymbol{h}_0,-\rho_s^{(2)}\boldsymbol{m}_0)}$, and $\mathcal{A}_{\epsilon,(-\rho_d^{(1)}\boldsymbol{h}_0,-\rho_s^{(2)}\boldsymbol{m}_0)}$, where $\epsilon$ depends on the expansivity and number of layers in the networks, and both $\rho_d^{(1)}$ and $\rho_s^{(2)}$ are positive constants close to 1 and depends on the number of layers in the two neural networks as well.

- Using sphere covering arguments, Gaussian and truncated Gaussian matrices with appropriate dimensions satisfy the WDC and joint-WDC conditions.

The full proof of Theorem 3 is provided in the Appendix.

## 4 Numerical Experiment

We now empirically show that Algorithm 1 can remove distortions present in the dataset. We consider the image recovery task of removing distortions that were synthetically introduced to the MNIST dataset. The distortion dataset contain 8100 images of size $28 \times 28$ where the distortions are generated using a 2D Gaussian function, $g(x,y) = e^{-\frac{(x-c)^2+(y-c)^2}{\sigma}}$, where $c$ is the center and $\sigma$ controls its tail behavior. For each of the 8100 image, we fix $c$ and $\sigma$, which vary uniformly in the intervals $[-3,3]$ and $[20,35]$, respectively, and $x$ and $y$ are in the interval $[-5,5]$. Prior to training the generators, the images in the MNIST dataset and the distortion dataset were resized to $64 \times 64$ images. We used DCGAN [Radford et al., 2016] with a learning rate of 0.0002 and latent code dimension of 50 to train a generator, $\mathcal{G}^{(2)}$, for the distortion images. Similarly, we used the DCGAN with learning rate of 0.0002 and latent code dimension of 100 to train a generator, $\mathcal{G}^{(1)}$, for the MNIST images. Finally,

a distorted image $y_0$ is generated via the pixelwise multiplication of an image $w_0$ from the MNIST dataset and an image $x_0$ from the distortion dataset, i.e. $y_0 = w_0 \odot x_0$.

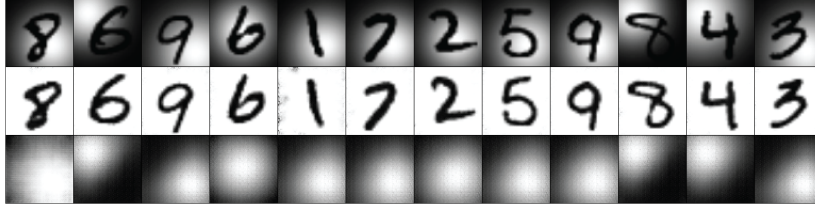

Figure 2: The figure shows the result removing distortion in an image by solving (2) using Algorithm 1. The top row corresponds to the input distorted image. The second and third row corresponds to the images recovered using empirical risk minimization.

Figure 2 shows the result of using Algorithm 1 to remove distortion from $y_0$. In the implementation of Algorithm 1, $\tilde{g}_{1,(h_i,m_i)}$ and $\tilde{g}_{1,(h_i,m_i)}$ corresponds to the partial derivatives of $f$ with the generators as $\mathcal{G}^{(1)}$ and $\mathcal{G}^{(2)}$. We used the Stochastic Gradient Descent algorithm with the step size set to 1 and momentum set to 0.9. For each image in the first row of Figure 2, the corresponding images in the second and third rows are the output of Algorithm 1 after 500 iterations.

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
