[Supplementary Material]

# 5 Appendix

Let $\angle(\boldsymbol{h}, \boldsymbol{h}_0) = \bar{\theta}_0^{(1)}$ and $\angle(\boldsymbol{m}, \boldsymbol{m}_0) = \bar{\theta}_0^{(2)}$ for non-zero $\boldsymbol{h}, \boldsymbol{h}_0 \in \mathbb{R}^n$ and $\boldsymbol{m}, \boldsymbol{m}_0 \in \mathbb{R}^p$. In order to understand how the operators $\boldsymbol{h} \to \boldsymbol{W}_{+,\boldsymbol{h}}^{(1)} \boldsymbol{h}$ and $\boldsymbol{m} \to \boldsymbol{W}_{+,\boldsymbol{m}}^{(2)} \boldsymbol{m}$ distort angles, we define

$$g(\theta) = \cos^{-1}\left(\frac{(\pi - \theta)\cos\theta + \sin\theta}{\pi}\right). \tag{8}$$

Also, for a fixed $\boldsymbol{p}, \boldsymbol{q} \in \mathbb{R}^n$, define

$$\tilde{t}_{\boldsymbol{p},\boldsymbol{q}}^{(k)} := \frac{1}{2^{a^{(k)}}}\left[\left(\prod_{i=0}^{a^{(k)}-1}\frac{\pi - \bar{\theta}_i^{(k)}}{\pi}\right)\boldsymbol{q} + \sum_{i=0}^{a^{(k)}-1}\frac{\sin\bar{\theta}_i^{(k)}}{\pi}\left(\prod_{j=i+1}^{a^{(k)}-1}\frac{\pi - \bar{\theta}_j^{(k)}}{\pi}\right)\frac{\|\boldsymbol{q}\|_2}{\|\boldsymbol{p}\|_2}\boldsymbol{p}\right], \tag{9}$$

where $\bar{\theta}_i^{(k)} = g(\bar{\theta}_{i-1}^{(k)})$ for $g$ given by (8), $\bar{\theta}_0^{(k)} = \angle(\boldsymbol{p}, \boldsymbol{q})$, $a^{(1)} = d$, and $a^{(2)} = s$.

## 5.1 Proof of Deterministic Theorem

**Theorem 4** (Also Theorem 3). *Fix $\epsilon > 0$, $0 < \alpha_1 \leq 1$ and $0 < \alpha_2 \leq 1$ such that $K_1(d^7 s^2 + d^2 s^7)\epsilon^{1/4}/(\alpha_1\alpha_2) < 1$, $d \geq 2$, and $s \geq 2$. Let $\mathcal{K} = \{(\boldsymbol{h}, \boldsymbol{0}) \in \mathbb{R}^{n \times p} \,|\, \boldsymbol{h} \in \mathbb{R}^n\} \cup \{(\boldsymbol{0}, \boldsymbol{m}) \in \mathbb{R}^{n \times p} \,|\, \boldsymbol{m} \in \mathbb{R}^p\}$. Suppose that $\boldsymbol{W}_i^{(1)} \in \mathbb{R}^{n_i \times n_{i-1}}$ for $i = 1, \ldots, d-1$ and $\boldsymbol{W}_i^{(2)} \in \mathbb{R}^{p_i \times p_{i-1}}$ for $i = 1, \ldots, s-1$ satisfy the WDC with constant $\epsilon$ and 1. Suppose $\boldsymbol{W}_d^{(1)} \in \mathbb{R}^{\ell \times n_{d-1}}$ satisfy WDC with constants $\epsilon$ and $\alpha_1$, and $\boldsymbol{W}_s^{(2)} \in \mathbb{R}^{\ell \times p_{s-1}}$ satisfy WDC with constants $\epsilon$ and $\alpha_2$. Also, suppose $\left(\boldsymbol{W}_d^{(1)}, \boldsymbol{W}_s^{(2)}\right)$ satisfy joint-WDC with constants $\epsilon$, $\alpha = \alpha_1 \cdot \alpha_2$. Let $\mathcal{K} = \{(\boldsymbol{h}, \boldsymbol{0}) \in \mathbb{R}^{n \times p} \,|\, \boldsymbol{h} \in \mathbb{R}^n\} \cup \{(\boldsymbol{0}, \boldsymbol{m}) \in \mathbb{R}^{n \times p} \,|\, \boldsymbol{m} \in \mathbb{R}^p\}$ and $\mathcal{A} = \mathcal{A}_{K_2 d^3 s^3 \epsilon^{\frac{1}{4}}\alpha^{-1}, (\boldsymbol{h}_0, \boldsymbol{m}_0)} \cup \mathcal{A}_{K_2 d^8 s^3 \epsilon^{\frac{1}{4}}\alpha^{-1}, \left(-\rho_d^{(1)}\boldsymbol{h}_0, \boldsymbol{m}_0\right)} \cup \mathcal{A}_{K_2 d^3 s^8 \epsilon^{\frac{1}{4}}\alpha^{-1}, \left(\rho_s^{(2)}\boldsymbol{h}_0, -\boldsymbol{m}_0\right)} \cup \mathcal{A}_{K_2 d^8 s^8 \epsilon^{\frac{1}{4}}\alpha^{-1}, \left(-\rho_d^{(1)}\rho_s^{(2)}\boldsymbol{h}_0, -\boldsymbol{m}_0\right)}$. Then, for $(\boldsymbol{h}_0, \boldsymbol{m}_0) \neq (\boldsymbol{0}, \boldsymbol{0})$, and*

$$(\boldsymbol{h}, \boldsymbol{m}) \notin \mathcal{A} \cup \mathcal{K}$$

*the one-sided directional derivative of $f$ in the direction of $\boldsymbol{g} = \boldsymbol{g}_{1,(\boldsymbol{h},\boldsymbol{m})}$ or $\boldsymbol{g} = \boldsymbol{g}_{2,(\boldsymbol{h},\boldsymbol{m})}$ satisfy $D_{-\boldsymbol{g}}f(\boldsymbol{h}, \boldsymbol{m}) < 0$. Additionally, for all $(\boldsymbol{h}, \boldsymbol{m}) \in \mathcal{K}$ and for all $(\boldsymbol{x}, \boldsymbol{y})$*

$$D_{(\boldsymbol{x},\boldsymbol{y})}f(\boldsymbol{h}, \boldsymbol{m}) \leq 0.$$

*Here, $\rho_d^{(k)}$ are positive numbers that converge to 1 as $d \to \infty$, and $K_1$, and $K_2$ are absolute constants.*

*Proof.* Recall that

$$\boldsymbol{v}_{(\boldsymbol{h},\boldsymbol{m}),(\boldsymbol{h}_0,\boldsymbol{m}_0)}^{(1)} = \begin{cases} \nabla_{\boldsymbol{h}}f(\boldsymbol{h}, \boldsymbol{m}) & G \text{ is differentiable at } (\boldsymbol{h}, \boldsymbol{m}), \\ \lim_{\delta \to 0^+} \nabla_{\boldsymbol{h}}f((\boldsymbol{h}, \boldsymbol{m}) + \delta\boldsymbol{w}) & \text{otherwise} \end{cases}$$

$$\boldsymbol{v}_{(\boldsymbol{h},\boldsymbol{m}),(\boldsymbol{h}_0,\boldsymbol{m}_0)}^{(2)} = \begin{cases} \nabla_{\boldsymbol{m}}f(\boldsymbol{h}, \boldsymbol{m}) & G \text{ is differentiable at } (\boldsymbol{h}, \boldsymbol{m}), \\ \lim_{\delta \to 0^+} \nabla_{\boldsymbol{m}}f((\boldsymbol{h}, \boldsymbol{m}) + \delta\boldsymbol{w}) & \text{otherwise} \end{cases}$$

where $G(\boldsymbol{h}, \boldsymbol{m})$ is differentiable at $(\boldsymbol{h}, \boldsymbol{m}) + \delta\boldsymbol{w}$ for sufficiently small $\delta$. Such a $\delta$ exists because of piecewise linearity of $G(\boldsymbol{h}, \boldsymbol{m})$ and any such $\boldsymbol{w}$ can be arbitrarily selected. Also, recall that for any differentiable point $(\boldsymbol{h}, \boldsymbol{m})$, we have

$$\nabla_{\boldsymbol{h}}f(\boldsymbol{h}, \boldsymbol{m}) = \left(\boldsymbol{\Lambda}_{d,+,\boldsymbol{h}}^{(1)}\right)^{\top} \text{diag}\left(\boldsymbol{\Lambda}_{s,+,\boldsymbol{m}}^{(2)}\boldsymbol{m}\right)^2 \boldsymbol{\Lambda}_{d,+,\boldsymbol{h}}^{(1)}\boldsymbol{h}$$
$$- \left(\boldsymbol{\Lambda}_{d,+,\boldsymbol{h}}^{(1)}\right)^{\top} \text{diag}\left(\boldsymbol{\Lambda}_{s,+,\boldsymbol{m}}^{(2)}\boldsymbol{m} \odot \boldsymbol{\Lambda}_{s,+,\boldsymbol{m}_0}^{(2)}\boldsymbol{m}_0\right) \boldsymbol{\Lambda}_{d,+,\boldsymbol{h}_0}^{(1)}\boldsymbol{h}_0,$$

$$\nabla_{\boldsymbol{m}}f(\boldsymbol{h}, \boldsymbol{m}) = \left(\boldsymbol{\Lambda}_{s,+,\boldsymbol{m}}^{(2)}\right)^{\top} \text{diag}\left(\boldsymbol{\Lambda}_{d,+,\boldsymbol{h}}^{(1)}\boldsymbol{h}\right)^2 \boldsymbol{\Lambda}_{s,+,\boldsymbol{m}}^{(2)}\boldsymbol{m}$$
$$- \left(\boldsymbol{\Lambda}_{s,+,\boldsymbol{m}}^{(2)}\right)^{\top} \text{diag}\left(\boldsymbol{\Lambda}_{d,+,\boldsymbol{h}}^{(1)}\boldsymbol{h} \odot \boldsymbol{\Lambda}_{d,+,\boldsymbol{h}_0}^{(1)}\boldsymbol{h}_0\right) \boldsymbol{\Lambda}_{s,+,\boldsymbol{m}_0}^{(2)}\boldsymbol{m}_0.$$

Let

$$\boldsymbol{g}_{1,(\boldsymbol{h},\boldsymbol{m})} = \left[ \begin{array}{c} \boldsymbol{v}^{(1)}_{(\boldsymbol{h},\boldsymbol{m}),(\boldsymbol{h}_0,\boldsymbol{m}_0)} \\ \boldsymbol{0} \end{array} \right] \in \mathbb{R}^{n \times p},$$

$$\boldsymbol{g}_{2,(\boldsymbol{h},\boldsymbol{m})} = \left[ \begin{array}{c} \boldsymbol{0} \\ \boldsymbol{v}^{(2)}_{(\boldsymbol{h},\boldsymbol{m}),(\boldsymbol{h}_0,\boldsymbol{m}_0)} \end{array} \right] \in \mathbb{R}^{n \times p},$$

$$\boldsymbol{t}^{(1)}_{(\boldsymbol{h},\boldsymbol{m}),(\boldsymbol{h}_0,\boldsymbol{m}_0)} = \frac{\alpha}{2^{d+s}\ell}\|\boldsymbol{m}\|_2^2 \boldsymbol{h} - \frac{\alpha}{\ell} \boldsymbol{m}^\intercal \tilde{\boldsymbol{t}}^{(2)}_{\boldsymbol{m},\boldsymbol{m}_0} \tilde{\boldsymbol{t}}^{(1)}_{\boldsymbol{h},\boldsymbol{h}_0}, \tag{10}$$

$$\boldsymbol{t}^{(2)}_{(\boldsymbol{h},\boldsymbol{m}),(\boldsymbol{h}_0,\boldsymbol{m}_0)} = \frac{\alpha}{2^{d+s}\ell}\|\boldsymbol{h}\|_2^2 \boldsymbol{m} - \frac{\alpha}{\ell} \boldsymbol{h}^\intercal \tilde{\boldsymbol{t}}^{(1)}_{\boldsymbol{h},\boldsymbol{h}_0} \tilde{\boldsymbol{t}}^{(2)}_{\boldsymbol{m},\boldsymbol{m}_0}, \tag{11}$$

$$S^{(1)}_{\epsilon,(\boldsymbol{h}_0,\boldsymbol{m}_0)} = \left\{ (\boldsymbol{h},\boldsymbol{m}) \in \mathbb{R}^{n \times p} \backslash \mathcal{K} \left| \frac{\|\boldsymbol{t}^{(1)}_{(\boldsymbol{h},\boldsymbol{m}),(\boldsymbol{h}_0,\boldsymbol{m}_0)}\|_2}{\|\boldsymbol{m}\|_2} \leq \frac{\epsilon \max(\|\boldsymbol{h}\|_2\|\boldsymbol{m}\|_2, \|\boldsymbol{h}_0\|_2\|\boldsymbol{m}_0\|_2)}{2^{d+s}\ell} \right. \right\},$$

$$S^{(2)}_{\epsilon,(\boldsymbol{h}_0,\boldsymbol{m}_0)} = \left\{ (\boldsymbol{h},\boldsymbol{m}) \in \mathbb{R}^{n \times p} \backslash \mathcal{K} \left| \frac{\|\boldsymbol{t}^{(2)}_{(\boldsymbol{h},\boldsymbol{m}),(\boldsymbol{h}_0,\boldsymbol{m}_0)}\|_2}{\|\boldsymbol{h}\|_2} \leq \frac{\epsilon \max(\|\boldsymbol{h}\|_2\|\boldsymbol{m}\|_2, \|\boldsymbol{h}_0\|_2\|\boldsymbol{m}_0\|_2)}{2^{d+s}\ell} \right. \right\},$$

where $\tilde{\boldsymbol{t}}^{(2)}_{\boldsymbol{m},\boldsymbol{m}_0}$ and $\tilde{\boldsymbol{t}}^{(1)}_{\boldsymbol{h},\boldsymbol{h}_0}$ is as defined in (9). For brevity of notation, write $\boldsymbol{v}^{(1)}_{(\boldsymbol{h},\boldsymbol{m})} = \boldsymbol{v}^{(1)}_{(\boldsymbol{h},\boldsymbol{m}),(\boldsymbol{h}_0,\boldsymbol{m}_0)}$, $\boldsymbol{v}^{(2)}_{(\boldsymbol{h},\boldsymbol{m})} = \boldsymbol{v}^{(2)}_{(\boldsymbol{h},\boldsymbol{m}),(\boldsymbol{h}_0,\boldsymbol{m}_0)}$, $\boldsymbol{t}_{(\boldsymbol{h},\boldsymbol{m})} = \boldsymbol{t}_{(\boldsymbol{h},\boldsymbol{m}),(\boldsymbol{h}_0,\boldsymbol{m}_0)}$, $\boldsymbol{t}^{(1)}_{(\boldsymbol{h},\boldsymbol{m})} = \boldsymbol{t}^{(1)}_{(\boldsymbol{h},\boldsymbol{m}),(\boldsymbol{h}_0,\boldsymbol{m}_0)}$ and $\boldsymbol{t}^{(2)}_{(\boldsymbol{h},\boldsymbol{m})} = \boldsymbol{t}^{(2)}_{(\boldsymbol{h},\boldsymbol{m}),(\boldsymbol{h}_0,\boldsymbol{m}_0)}$.

Since $\boldsymbol{W}^{(1)}_i \in \mathbb{R}^{n_i \times n_{i-1}}$ for $i = 1, \ldots, d-1$ and $\boldsymbol{W}^{(2)}_i \in \mathbb{R}^{p_i \times p_{i-1}}$ for $i = 1, \ldots, s-1$ satisfy the WDC with constant $\epsilon$ and $1$, $\boldsymbol{W}^{(1)}_d \in \mathbb{R}^{\ell \times n_{d-1}}$ satisfy WDC with constants $\epsilon$ and $\alpha_1$, $\boldsymbol{W}^{(2)}_s \in \mathbb{R}^{\ell \times p_{s-1}}$ satisfy WDC with constants $\epsilon$ and $\alpha_2$, and $\left( \boldsymbol{W}^{(1)}_d, \boldsymbol{W}^{(2)}_s \right)$ satisfy joint-WDC with constants $\epsilon$, $\alpha = \alpha_1 \cdot \alpha_2$, we have lemma 2 implying for all nonzero $\boldsymbol{h}$, $\boldsymbol{h}_0 \in \mathbb{R}^n$ and nonzero $\boldsymbol{m}$, $\boldsymbol{m}_0 \in \mathbb{R}^p$

$$\|\nabla_{\boldsymbol{h}} f(\boldsymbol{h},\boldsymbol{m}) - \boldsymbol{t}^{(1)}_{(\boldsymbol{h},\boldsymbol{m})}\|_2 \leq K \frac{d^3 s^3 \sqrt{\epsilon}}{2^{d+s}\ell} \max(\|\boldsymbol{h}\|_2\|\boldsymbol{m}\|_2, \|\boldsymbol{h}_0\|_2\|\boldsymbol{m}_0\|_2)\|\boldsymbol{m}\|_2, \tag{12}$$

$$\|\nabla_{\boldsymbol{m}} f(\boldsymbol{h},\boldsymbol{m}) - \boldsymbol{t}^{(2)}_{(\boldsymbol{h},\boldsymbol{m})}\|_2 \leq K \frac{d^3 s^2 \sqrt{\epsilon}}{2^{d+s}\ell} \max(\|\boldsymbol{h}\|_2\|\boldsymbol{m}\|_2, \|\boldsymbol{h}_0\|_2\|\boldsymbol{m}_0\|_2)\|\boldsymbol{h}\|_2. \tag{13}$$

Thus, we have, for all nonzero $\boldsymbol{h}$, $\boldsymbol{h}_0 \in \mathbb{R}^n$ and nonzero $\boldsymbol{m}$, $\boldsymbol{m}_0 \in \mathbb{R}^p$,

$$\|\boldsymbol{v}^{(1)}_{(\boldsymbol{h},\boldsymbol{m})} - \boldsymbol{t}^{(1)}_{(\boldsymbol{h},\boldsymbol{m})}\|_2 = \lim_{\delta \to 0^+} \|\nabla_{\boldsymbol{h}} f\left((\boldsymbol{h},\boldsymbol{x}) + \delta \boldsymbol{w}\right) - \boldsymbol{t}^{(1)}_{(\boldsymbol{h},\boldsymbol{m})+\delta \boldsymbol{w}}\|_2$$

$$\leq K \frac{d^3 s^3 \sqrt{\epsilon}}{2^{d+s}\ell} \max(\|\boldsymbol{h}\|_2\|\boldsymbol{m}\|_2, \|\boldsymbol{h}_0\|_2\|\boldsymbol{m}_0\|_2)\|\boldsymbol{m}\|_2, \text{ and}$$

$$\|\boldsymbol{v}^{(2)}_{(\boldsymbol{h},\boldsymbol{m})} - \boldsymbol{t}^{(2)}_{(\boldsymbol{h},\boldsymbol{m})}\|_2 = \lim_{\delta \to 0^+} \|\nabla_{\boldsymbol{m}} f\left((\boldsymbol{h},\boldsymbol{x}) + \delta \boldsymbol{w}\right) - \boldsymbol{t}^{(2)}_{(\boldsymbol{h},\boldsymbol{m})+\delta \boldsymbol{w}}\|_2$$

$$\leq K \frac{d^3 s^3 \sqrt{\epsilon}}{2^{d+s}\ell} \max(\|\boldsymbol{h}\|_2\|\boldsymbol{m}\|_2, \|\boldsymbol{h}_0\|_2\|\boldsymbol{m}_0\|_2)\|\boldsymbol{h}\|_2,$$

where the inequalities follow from (12) and (13).

Note that the one-sided directional derivative of $f$ in the direction of $(\boldsymbol{x},\boldsymbol{y}) \neq \boldsymbol{0}$ at $(\boldsymbol{h},\boldsymbol{y})$ is $D_{(\boldsymbol{x},\boldsymbol{y})} f(\boldsymbol{h},\boldsymbol{x}) = \lim_{t \to 0^+} \frac{1}{t} \left( f((\boldsymbol{h},\boldsymbol{x}) + t(\boldsymbol{x},\boldsymbol{y})) - f(\boldsymbol{h},\boldsymbol{x}) \right)$. Due to the continuity and piecewise linearity of the function

$$\mathcal{G}(\boldsymbol{h},\boldsymbol{m}) = \boldsymbol{\Lambda}^{(1)}_{d,+,\boldsymbol{h}} \boldsymbol{h} \odot \boldsymbol{\Lambda}^{(2)}_{s,+,\boldsymbol{m}} \boldsymbol{m},$$

we have that for any $(\boldsymbol{h},\boldsymbol{m}) \neq (\boldsymbol{0},\boldsymbol{0})$ and $(\boldsymbol{x},\boldsymbol{y}) \neq \boldsymbol{0}$ that there exists a sequence $\{(\boldsymbol{h}_n,\boldsymbol{m}_n)\} \to (\boldsymbol{h},\boldsymbol{m})$ such that $f$ is differentiable at each $(\boldsymbol{h}_n,\boldsymbol{m}_n)$ and $D_{(\boldsymbol{x},\boldsymbol{y})} f(\boldsymbol{h},\boldsymbol{m}) = \lim_{n \to \infty} \nabla f(\boldsymbol{h}_n,\boldsymbol{m}_n) \cdot (\boldsymbol{x},\boldsymbol{y})$. Thus, as $\nabla f(\boldsymbol{h}_n,\boldsymbol{m}_n) = \left[ \begin{array}{c} \boldsymbol{v}^{(1)}_{(\boldsymbol{h}_n,\boldsymbol{m}_n)} \\ \boldsymbol{v}^{(2)}_{(\boldsymbol{h}_n,\boldsymbol{m}_n)} \end{array} \right]$,

$$D_{-\boldsymbol{g}_{1,(\boldsymbol{h},\boldsymbol{m})}} f(\boldsymbol{h},\boldsymbol{m}) = \lim_{n \to \infty} \nabla f(\boldsymbol{h}_n,\boldsymbol{m}_n) \cdot \frac{-\boldsymbol{g}_{1,(\boldsymbol{h},\boldsymbol{m})}}{\|\boldsymbol{g}_{1,(\boldsymbol{h},\boldsymbol{m})}\|_2} = \frac{-1}{\|\boldsymbol{g}_{1,(\boldsymbol{h},\boldsymbol{m})}\|_2} \lim_{n \to \infty} \boldsymbol{v}^{(1)}_{(\boldsymbol{h}_n,\boldsymbol{m}_n)} \cdot \boldsymbol{v}^{(1)}_{(\boldsymbol{h},\boldsymbol{m})},$$

$$D_{-\boldsymbol{g}_{2,(\boldsymbol{h},\boldsymbol{m})}} f(\boldsymbol{h},\boldsymbol{m}) = \lim_{n\to\infty} \nabla f(\boldsymbol{h}_n,\boldsymbol{m}_n) \cdot \frac{-\boldsymbol{g}_{2,(\boldsymbol{h},\boldsymbol{m})}}{\|\boldsymbol{g}_{2,(\boldsymbol{h},\boldsymbol{m})}\|_2} = \frac{-1}{\|\boldsymbol{g}_{2,(\boldsymbol{h},\boldsymbol{m})}\|_2} \lim_{n\to\infty} \boldsymbol{v}^{(2)}_{(\boldsymbol{h}_n,\boldsymbol{m}_n)} \cdot \boldsymbol{v}^{(2)}_{(\boldsymbol{h},\boldsymbol{m})}.$$

Now, we write

$$\boldsymbol{v}^{(1)}_{(\boldsymbol{h}_n,\boldsymbol{m}_n)} \cdot \boldsymbol{v}^{(1)}_{(\boldsymbol{h},\boldsymbol{m})}$$
$$=\boldsymbol{t}^{(1)}_{(\boldsymbol{h}_n,\boldsymbol{m}_n)} \cdot \boldsymbol{t}^{(1)}_{(\boldsymbol{h},\boldsymbol{m})} + (\boldsymbol{v}^{(1)}_{(\boldsymbol{h}_n,\boldsymbol{m}_n)} - \boldsymbol{t}^{(1)}_{(\boldsymbol{h}_n,\boldsymbol{m}_n)}) \cdot \boldsymbol{t}^{(1)}_{(\boldsymbol{h},\boldsymbol{m})} + \boldsymbol{t}^{(1)}_{(\boldsymbol{h}_n,\boldsymbol{m}_n)} \cdot (\boldsymbol{v}^{(1)}_{(\boldsymbol{h},\boldsymbol{m})} - \boldsymbol{t}^{(1)}_{(\boldsymbol{h},\boldsymbol{m})})$$
$$+ (\boldsymbol{v}^{(1)}_{(\boldsymbol{h}_n,\boldsymbol{m}_n)} - \boldsymbol{t}^{(1)}_{(\boldsymbol{h}_n,\boldsymbol{m}_n)}) \cdot (\boldsymbol{v}^{(1)}_{(\boldsymbol{h},\boldsymbol{m})} - \boldsymbol{t}^{(1)}_{(\boldsymbol{h},\boldsymbol{m})})$$
$$\geq \boldsymbol{t}^{(1)}_{(\boldsymbol{h}_n,\boldsymbol{m}_n)} \cdot \boldsymbol{t}^{(1)}_{(\boldsymbol{h},\boldsymbol{m})} - \|\boldsymbol{v}^{(1)}_{(\boldsymbol{h}_n,\boldsymbol{m}_n)} - \boldsymbol{t}^{(1)}_{(\boldsymbol{h}_n,\boldsymbol{m}_n)}\|_2 \|\boldsymbol{t}^{(1)}_{(\boldsymbol{h},\boldsymbol{m})}\|_2 - \|\boldsymbol{t}^{(1)}_{(\boldsymbol{h}_n,\boldsymbol{m}_n)}\|_2 \|\boldsymbol{v}^{(1)}_{(\boldsymbol{h},\boldsymbol{m})} - \boldsymbol{t}^{(1)}_{(\boldsymbol{h},\boldsymbol{m})}\|_2$$
$$\|\boldsymbol{v}^{(1)}_{(\boldsymbol{h}_n,\boldsymbol{m}_n)} - \boldsymbol{t}^{(1)}_{(\boldsymbol{h}_n,\boldsymbol{m}_n)}\|_2 \|\boldsymbol{v}^{(1)}_{(\boldsymbol{h},\boldsymbol{m})} - \boldsymbol{t}^{(1)}_{(\boldsymbol{h},\boldsymbol{m})}\|_2$$
$$\geq \boldsymbol{t}^{(1)}_{(\boldsymbol{h}_n,\boldsymbol{m}_n)} \cdot \boldsymbol{t}^{(1)}_{(\boldsymbol{h},\boldsymbol{m})} - K \frac{d^3 s^3 \sqrt{\epsilon}}{2^{d+s}\ell} \max(\|\boldsymbol{h}_n\|_2 \|\boldsymbol{m}_n\|_2, \|\boldsymbol{h}_0\|_2 \|\boldsymbol{m}_0\|_2) \|\boldsymbol{m}_n\|_2 \|\boldsymbol{t}^{(1)}_{(\boldsymbol{h},\boldsymbol{m})}\|_2$$
$$- K \frac{d^3 s^3 \sqrt{\epsilon}}{2^{d+s}\ell} \max(\|\boldsymbol{h}\|_2 \|\boldsymbol{m}\|_2, \|\boldsymbol{h}_0\|_2 \|\boldsymbol{m}_0\|_2) \|\boldsymbol{m}\|_2 \|\boldsymbol{t}^{(1)}_{(\boldsymbol{h}_n,\boldsymbol{m}_n)}\|_2$$
$$- \left( K \frac{d^3 s^3 \sqrt{\epsilon}}{2^{d+s}\ell} \right)^2 \max(\|\boldsymbol{h}_n\|_2 \|\boldsymbol{m}_n\|_2, \|\boldsymbol{h}_0\|_2 \|\boldsymbol{m}_0\|_2)$$
$$\max(\|\boldsymbol{h}\|_2 \|\boldsymbol{m}\|_2, \|\boldsymbol{h}_0\|_2 \|\boldsymbol{m}_0\|_2) \|\boldsymbol{m}_n\|_2 \|\boldsymbol{m}\|_2.$$

As $\boldsymbol{t}^{(1)}_{(\boldsymbol{h},\boldsymbol{m})}$ is continuous in $(\boldsymbol{h},\boldsymbol{m})$ for all $(\boldsymbol{h},\boldsymbol{m}) \notin \mathcal{K}$, we have for all $(\boldsymbol{h},\boldsymbol{m}) \notin \mathcal{S}^{(1)}_{4Kd^3 s^3 \sqrt{\epsilon},(\boldsymbol{h}_0,\boldsymbol{m}_0)} \cup \mathcal{K}$,

$$\lim_{n\to\infty} \boldsymbol{v}^{(1)}_{(\boldsymbol{h}_n,\boldsymbol{m}_n)} \cdot \boldsymbol{v}^{(1)}_{(\boldsymbol{h},\boldsymbol{m})}$$
$$\geq \|\boldsymbol{t}^{(1)}_{(\boldsymbol{h},\boldsymbol{m})}\|_2^2 - 2K \frac{d^3 s^3 \sqrt{\epsilon}}{2^{d+s}\ell} \max(\|\boldsymbol{h}\|_2 \|\boldsymbol{m}\|_2, \|\boldsymbol{h}_0\|_2 \|\boldsymbol{m}_0\|_2) \|\boldsymbol{m}\|_2 \|\boldsymbol{t}^{(1)}_{(\boldsymbol{h},\boldsymbol{m})}\|_2$$
$$- \left( K \frac{d^3 s^3 \sqrt{\epsilon}}{2^{d+s}\ell} \max(\|\boldsymbol{h}\|_2 \|\boldsymbol{m}\|_2, \|\boldsymbol{h}_0\|_2 \|\boldsymbol{m}_0\|_2) \|\boldsymbol{m}\|_2 \right)^2$$
$$\geq \frac{\|\boldsymbol{t}^{(1)}_{(\boldsymbol{h},\boldsymbol{m})}\|_2}{2} \left[ \|\boldsymbol{t}^{(1)}_{(\boldsymbol{h},\boldsymbol{m})}\|_2 - 4K \frac{d^3 s^3 \sqrt{\epsilon}}{2^{d+s}\ell} \max(\|\boldsymbol{h}\|_2 \|\boldsymbol{m}\|_2, \|\boldsymbol{h}_0\|_2 \|\boldsymbol{m}_0\|_2) \|\boldsymbol{m}\|_2 \right]_+ +$$
$$\frac{1}{2} \left[ \|\boldsymbol{t}^{(1)}_{(\boldsymbol{h},\boldsymbol{m})}\|_2^2 - 2 \left( K \frac{d^3 s^3 \sqrt{\epsilon}}{2^{d+s}\ell} \max(\|\boldsymbol{h}\|_2 \|\boldsymbol{m}\|_2, \|\boldsymbol{h}_0\|_2 \|\boldsymbol{m}_0\|_2) \|\boldsymbol{m}\|_2 \right)^2 \right]$$
$$> 0. \tag{14}$$

Similarly, we have for all $(\boldsymbol{h},\boldsymbol{m}) \notin \mathcal{S}^{(2)}_{4Kd^3 s^3 \sqrt{\epsilon},(\boldsymbol{h}_0,\boldsymbol{m}_0)} \cup \mathcal{K}$,

$$\lim_{n\to\infty} \boldsymbol{v}^{(2)}_{(\boldsymbol{h}_n,\boldsymbol{m}_n)} \cdot \boldsymbol{v}^{(2)}_{(\boldsymbol{h},\boldsymbol{m})} > 0. \tag{15}$$

So, for all $(\boldsymbol{h},\boldsymbol{m}) \notin \left( \mathcal{S}^{(1)}_{4Kd^3 s^3 \sqrt{\epsilon},(\boldsymbol{h}_0,\boldsymbol{m}_0)} \cap \mathcal{S}^{(2)}_{4Kd^3 s^3 \sqrt{\epsilon},(\boldsymbol{h}_0,\boldsymbol{m}_0)} \right) \cup \mathcal{K}$, at least (14) or (15) holds. If (14) holds, then we have $D_{-\boldsymbol{g}_{1,(\boldsymbol{h},\boldsymbol{m})}} f(\boldsymbol{h},\boldsymbol{m}) < 0$ and if (15) holds, then we have $D_{-\boldsymbol{g}_{2,(\boldsymbol{h},\boldsymbol{m})}} f(\boldsymbol{h},\boldsymbol{m}) < 0$. Let $\mathcal{S} = \mathcal{S}^{(1)}_{4Kd^3 s^3 \sqrt{\epsilon},(\boldsymbol{h}_0,\boldsymbol{m}_0)} \cap \mathcal{S}^{(2)}_{4Kd^3 s^3 \sqrt{\epsilon},(\boldsymbol{h}_0,\boldsymbol{m}_0)}$. Apply Lemma 3 and $38(d^5 + s^5)\sqrt{4Kd^3 s^3 \sqrt{\epsilon}}/\alpha < 1$ to get

$$\mathcal{S} \subseteq \mathcal{A}_{\tilde{K}\frac{d^3 s^3 \epsilon^{1/4}}{\alpha},(\boldsymbol{h}_0,\boldsymbol{m}_0)} \cup \mathcal{A}_{\tilde{K}\frac{d^8 s^3 \epsilon^{1/4}}{\alpha},\left(-\rho_d^{(1)}\boldsymbol{h}_0,\boldsymbol{m}_0\right)} \cup \mathcal{A}_{\tilde{K}\frac{d^3 s^8 \epsilon^{1/4}}{\alpha},\left(\rho_s^{(2)}\boldsymbol{h}_0,-\boldsymbol{m}_0\right)}$$
$$\cup \mathcal{A}_{\tilde{K}\frac{d^8 s^8 \epsilon^{1/4}}{\alpha},\left(-\rho_d^{(1)}\rho_s^{(2)}\boldsymbol{h}_0,-\boldsymbol{m}_0\right)},$$

for some absolute constant $\tilde{K}$.

It remains to prove that elements of the set $\mathcal{K}$ are local maximizers. We now show that elements of the set $\{(\boldsymbol{0},\boldsymbol{m}) \in \mathbb{R}^{n\times p} | \boldsymbol{m} \in \mathbb{R}^p\}$ are local maximizers. Fix a direction $(\boldsymbol{x},\boldsymbol{y}) \in \mathbb{R}^{n\times p}$ and let

$$\tilde{\boldsymbol{h}}_0 = \boldsymbol{\Lambda}^{(1)}_{d-1,+,\boldsymbol{h}_0} \boldsymbol{h}_0, \quad \tilde{\boldsymbol{x}} = \boldsymbol{\Lambda}^{(1)}_{d-1,+,\boldsymbol{x}} \boldsymbol{x}, \quad \tilde{\boldsymbol{m}} = \boldsymbol{\Lambda}^{(2)}_{s-1,+,\boldsymbol{m}} \boldsymbol{m}, \quad \tilde{\boldsymbol{m}}_0 = \boldsymbol{\Lambda}^{(2)}_{s-1,+,\boldsymbol{m}_0} \boldsymbol{m}_0,$$

$\bar{\theta}_i^{(k)} = g(\bar{\theta}_{i-1}^{(k)})$ for $g$ given in (8), $\bar{\theta}_0^{(1)} = \angle(\boldsymbol{x}, \boldsymbol{h}_0)$ and $\bar{\theta}_0^{(2)} = \angle(\boldsymbol{m}, \boldsymbol{m}_0)$. We compute

$$
\begin{aligned}
&- D_{(\boldsymbol{x},\boldsymbol{y})} f(\boldsymbol{h}, \boldsymbol{m}) \cdot \|(\boldsymbol{x}, \boldsymbol{y})\|_2 \\
&= \lim_{t \to 0^+} \frac{f((\boldsymbol{h}, \boldsymbol{m}) + t(\boldsymbol{x}, \boldsymbol{y})) - f(\boldsymbol{h}, \boldsymbol{m})}{t} \\
&= \left\langle \text{diag}\left(\boldsymbol{\Lambda}_{s,+,\boldsymbol{m}}^{(2)} \boldsymbol{m}\right) \boldsymbol{\Lambda}_{d,+,\boldsymbol{x}}^{(1)} \boldsymbol{x}, \text{diag}\left(\boldsymbol{\Lambda}_{s,+,\boldsymbol{m}_0}^{(2)} \boldsymbol{m}_0\right) \boldsymbol{\Lambda}_{d,+,\boldsymbol{h}_0}^{(1)} \boldsymbol{h}_0 \right\rangle \\
&= \left\langle \tilde{\boldsymbol{x}}, \left(\boldsymbol{W}_{d-1,+,\boldsymbol{m}}^{(2)}\right)^{\mathsf{T}} \text{diag}\left(\boldsymbol{W}_{d-1,+,\boldsymbol{x}}^{(1)} \tilde{\boldsymbol{m}} \odot \boldsymbol{W}_{d-1,+,\boldsymbol{x}_0}^{(1)} \tilde{\boldsymbol{m}}_0\right) \boldsymbol{W}_{d-1,+,\boldsymbol{h}_0}^{(1)} \tilde{\boldsymbol{h}}_0 \right\rangle \\
&= \left\langle \tilde{\boldsymbol{x}}, \left( \left(\boldsymbol{W}_{d-1,+,\boldsymbol{m}}^{(2)}\right)^{\mathsf{T}} \text{diag}\left(\boldsymbol{W}_{d-1,+,\boldsymbol{x}}^{(1)} \tilde{\boldsymbol{m}} \odot \boldsymbol{W}_{d-1,+,\boldsymbol{x}_0}^{(1)} \tilde{\boldsymbol{m}}_0\right) \boldsymbol{W}_{d-1,+,\boldsymbol{h}_0}^{(1)} \right. \right. \\
&\qquad \left. \left. - \frac{\alpha}{n} \boldsymbol{Q}_{\tilde{\boldsymbol{x}},\tilde{\boldsymbol{h}}_0} \tilde{\boldsymbol{m}}^{\mathsf{T}} \boldsymbol{Q}_{\tilde{\boldsymbol{m}},\tilde{\boldsymbol{m}}_0} \tilde{\boldsymbol{m}}_0 \right) \tilde{\boldsymbol{h}}_0 \right\rangle + \frac{\alpha}{n} \tilde{\boldsymbol{m}}^{\mathsf{T}} \boldsymbol{Q}_{\tilde{\boldsymbol{m}},\tilde{\boldsymbol{m}}_0} \tilde{\boldsymbol{m}}_0 \cdot \tilde{\boldsymbol{x}}^{\mathsf{T}} \boldsymbol{Q}_{\tilde{\boldsymbol{x}},\tilde{\boldsymbol{h}}_0} \tilde{\boldsymbol{h}}_0 \\
&\geq -\left\| \left(\boldsymbol{W}_{d-1,+,\boldsymbol{m}}^{(2)}\right)^{\mathsf{T}} \text{diag}\left(\boldsymbol{W}_{d-1,+,\boldsymbol{x}}^{(1)} \tilde{\boldsymbol{m}} \odot \boldsymbol{W}_{d-1,+,\boldsymbol{x}_0}^{(1)} \tilde{\boldsymbol{m}}_0\right) \boldsymbol{W}_{d-1,+,\boldsymbol{h}_0}^{(1)} \right. \\
&\qquad \left. - \frac{\alpha}{n} \boldsymbol{Q}_{\tilde{\boldsymbol{x}},\tilde{\boldsymbol{h}}_0} \tilde{\boldsymbol{m}}^{\mathsf{T}} \boldsymbol{Q}_{\tilde{\boldsymbol{m}},\tilde{\boldsymbol{m}}_0} \tilde{\boldsymbol{m}}_0 \right\| \|\tilde{\boldsymbol{x}}\|_2 \|\tilde{\boldsymbol{h}}_0\|_2 + \frac{\alpha}{4n} \left( \frac{(\pi - \bar{\theta}_{d-1}^{(1)}) \cos \bar{\theta}_{d-1}^{(1)} + \sin \bar{\theta}_{d-1}^{(1)}}{\pi} \right) \\
&\qquad \left( \frac{(\pi - \bar{\theta}_{d-1}^{(2)}) \cos \bar{\theta}_{d-1}^{(2)} + \sin \bar{\theta}_{d-1}^{(2)}}{\pi} \right) \|\tilde{\boldsymbol{x}}\| \|\tilde{\boldsymbol{h}}_0\| \|\tilde{\boldsymbol{m}}\| \|\tilde{\boldsymbol{m}}_0\| \\
&\geq -\frac{4\epsilon}{n} \|\tilde{\boldsymbol{m}}\|_2 \|\tilde{\boldsymbol{m}}_0\|_2 \|\tilde{\boldsymbol{x}}\|_2 \|\tilde{\boldsymbol{h}}_0\|_2 + \frac{\alpha}{4n} \|\tilde{\boldsymbol{x}}\| \|\tilde{\boldsymbol{h}}_0\| \|\tilde{\boldsymbol{m}}\| \|\tilde{\boldsymbol{m}}_0\| \cos \bar{\theta}_d^{(1)} \cos \bar{\theta}_d^{(2)} \\
&\geq \left( -\frac{4\epsilon}{n} + \frac{\alpha}{4\pi^2 n} \right) \|\tilde{\boldsymbol{x}}\| \|\tilde{\boldsymbol{h}}_0\| \|\tilde{\boldsymbol{m}}\| \|\tilde{\boldsymbol{m}}_0\|.
\end{aligned}
$$

By Lemma 1, we have $\|\tilde{\boldsymbol{x}}\| \geq \frac{1}{4\pi} \frac{1}{2^{d-1}} \|\boldsymbol{x}\|$ and $\|\tilde{\boldsymbol{m}}\| \geq \frac{1}{4\pi} \frac{1}{2^{s-1}} \|\boldsymbol{m}\|$. So, if $4\pi^2 \epsilon / \alpha < 1$, then $D_{(\boldsymbol{x},\boldsymbol{y})} f(\boldsymbol{h}, \boldsymbol{m}) \cdot \|(\boldsymbol{x}, \boldsymbol{y})\|_2 < 0$ for all $(\boldsymbol{x}, \boldsymbol{y}) \in \mathbb{R}^{n \times p}$ with $\boldsymbol{x} \neq \boldsymbol{0}$ and $D_{(\boldsymbol{x},\boldsymbol{y})} f(\boldsymbol{h}, \boldsymbol{m}) \cdot \|(\boldsymbol{x}, \boldsymbol{y})\|_2 = 0$ for all $(\boldsymbol{x}, \boldsymbol{y}) \in \mathbb{R}^{n \times p}$ with $\boldsymbol{x} = \boldsymbol{0}$. Thus, elements of the set $\{(\boldsymbol{0}, \boldsymbol{m}) \in \mathbb{R}^{n \times p} | \boldsymbol{m} \in \mathbb{R}^p\}$ are local maximizers. Similarly, elements of the set $\{(\boldsymbol{h}, \boldsymbol{0}) \in \mathbb{R}^{n \times p} | \boldsymbol{h} \in \mathbb{R}^n\}$ are local maximizers. This concludes the proof of Theorem 4. $\qquad \square$

## 5.2 Concentration of terms in $\tilde{g}_{1,(\boldsymbol{h},\boldsymbol{m})}$ and $\tilde{g}_{2,(\boldsymbol{h},\boldsymbol{m})}$

**Lemma 1.** *Fix* $0 < \epsilon < d^{-4}/(16\pi)^2$ *and* $d \geq 2$. *Suppose that* $\boldsymbol{W}_i \in \mathbb{R}^{n_i \times n_{i-1}}$ *satisfies the WDC with constant* $\epsilon$ *and* $1$ *for* $i = 1, \ldots, d$. *Define*

$$
\tilde{t}_{\boldsymbol{p},\boldsymbol{q}} = \frac{1}{2^d} \left[ \left( \prod_{i=0}^{d-1} \frac{\pi - \bar{\theta}_i}{\pi} \right) \boldsymbol{q} + \sum_{i=0}^{d-1} \frac{\sin \bar{\theta}_i}{\pi} \left( \prod_{j=i+1}^{d-1} \frac{\pi - \bar{\theta}_j}{\pi} \right) \frac{\|\boldsymbol{q}\|_2}{\|\boldsymbol{p}\|_2} \boldsymbol{p} \right],
$$

*where* $\bar{\theta}_i = g(\bar{\theta}_{i-1})$ *for* $g$ *given by* (8) *and* $\bar{\theta}_0 = \angle(\boldsymbol{p}, \boldsymbol{q})$. *For all* $\boldsymbol{p} \neq 0$ *and* $\boldsymbol{q} \neq 0$,

$$
\left\| \left( \prod_{i=d}^{1} \boldsymbol{W}_{i,+,\boldsymbol{p}} \right)^{\mathsf{T}} \left( \prod_{i=d}^{1} \boldsymbol{W}_{i,+,\boldsymbol{q}} \right) \boldsymbol{q} - \tilde{t}_{\boldsymbol{p},\boldsymbol{q}} \right\|_2 \leq 24 \frac{d^3 \sqrt{\epsilon}}{2^d} \|\boldsymbol{q}\|_2, \tag{16}
$$

$$
\left\langle \left( \prod_{i=d}^{1} \boldsymbol{W}_{i,+,\boldsymbol{p}} \right) \boldsymbol{p}, \left( \prod_{i=d}^{1} \boldsymbol{W}_{i,+,\boldsymbol{q}} \right) \boldsymbol{q} \right\rangle \geq \frac{1}{4\pi} \frac{1}{2^d} \|\boldsymbol{p}\|_2 \|\boldsymbol{q}\|_2 \tag{17}
$$

We refer the readers to Hand and Voroninski [2017] for proof of Lemma 1. We now state a related Lemma.

**Lemma 2.** *Fix* $0 < \epsilon < 1/((d^4 + s^4)16\pi)^2$, $d \geq 2$ *and* $s \geq 2$. *Suppose that* $\boldsymbol{W}_i^{(1)} \in \mathbb{R}^{n_i \times n_{i-1}}$ *for* $i = 1, \ldots, d-1$ *and* $\boldsymbol{W}_i^{(2)} \in \mathbb{R}^{p_i \times p_{i-1}}$ *for* $i = 1, \ldots, s-1$ *satisfy the WDC with constant* $\epsilon$ *and* $1$. *Suppose* $\boldsymbol{W}_d^{(1)} \in \mathbb{R}^{\ell \times n_{d-1}}$ *satisfy WDC with constants* $\epsilon$ *and* $\alpha_1$, *and* $\boldsymbol{W}_s^{(2)} \in \mathbb{R}^{\ell \times p_{s-1}}$ *satisfy WDC with constants* $\epsilon$ *and* $\alpha_2$. *Also, suppose* $\left(\boldsymbol{W}_d^{(1)}, \boldsymbol{W}_s^{(2)}\right)$ *satisfy pair-WDC with constants* $\epsilon$,

$\alpha = \alpha_1 \cdot \alpha_2$. *Define*

$$\tilde{t}_{\boldsymbol{p},\boldsymbol{q}}^{(k)} = \frac{1}{2^{a^{(k)}}} \left[ \left( \prod_{i=0}^{a^{(k)}-1} \frac{\pi - \bar{\theta}_i^{(k)}}{\pi} \right) \boldsymbol{q} + \sum_{i=0}^{a^{(k)}-1} \frac{\sin \bar{\theta}_i^{(k)}}{\pi} \left( \prod_{j=i+1}^{a^{(k)}-1} \frac{\pi - \bar{\theta}_j^{(k)}}{\pi} \right) \frac{\|\boldsymbol{q}\|_2}{\|\boldsymbol{p}\|_2} \boldsymbol{p} \right],$$

*where* $\bar{\theta}_i^{(k)} = g(\bar{\theta}_{i-1}^{(k)})$ *for g given by* (8), $\bar{\theta}_0^{(k)} = \angle(\boldsymbol{p}, \boldsymbol{q})$, $a^{(1)} = d$, *and* $a^{(2)} = s$. *For all* $\boldsymbol{h} \neq 0$, $\boldsymbol{x} \neq 0$, $\boldsymbol{m} \neq 0$ *and* $\boldsymbol{y} \neq 0$,

$$\left\| \left( \boldsymbol{\Lambda}_{d,+,\boldsymbol{h}}^{(1)} \right)^{\mathsf{T}} diag \left( \boldsymbol{\Lambda}_{s,+,\boldsymbol{m}}^{(2)} \boldsymbol{m} \odot \boldsymbol{\Lambda}_{s,+,\boldsymbol{y}}^{(2)} \boldsymbol{y} \right) \boldsymbol{\Lambda}_{d,+,\boldsymbol{x}}^{(1)} \boldsymbol{x} - \frac{\alpha}{n} \left( \boldsymbol{m}^{\mathsf{T}} \tilde{\boldsymbol{t}}_{\boldsymbol{m},\boldsymbol{y}}^{(2)} \right) \tilde{\boldsymbol{t}}_{\boldsymbol{h},\boldsymbol{x}}^{(1)} \right\|_2$$
$$\leq \frac{208 d^3 s^3 \sqrt{\epsilon}}{2^{d+s} \ell} \|\boldsymbol{x}\|_2 \|\boldsymbol{m}\|_2 \|\boldsymbol{y}\|_2, \tag{18}$$

$$\left\| \left( \boldsymbol{\Lambda}_{s,+,\boldsymbol{m}}^{(2)} \right)^{\mathsf{T}} diag \left( \boldsymbol{\Lambda}_{d,+,\boldsymbol{h}}^{(1)} \boldsymbol{h} \odot \boldsymbol{\Lambda}_{d,+,\boldsymbol{x}}^{(1)} \boldsymbol{x} \right) \boldsymbol{\Lambda}_{s,+,\boldsymbol{y}}^{(2)} \boldsymbol{y} - \frac{\alpha}{n} \left( \boldsymbol{h}^{\mathsf{T}} \tilde{\boldsymbol{t}}_{\boldsymbol{h},\boldsymbol{x}}^{(1)} \right) \tilde{\boldsymbol{t}}_{\boldsymbol{m},\boldsymbol{y}}^{(2)} \right\|_2$$
$$\leq \frac{208 d^3 s^3 \sqrt{\epsilon}}{2^{d+s} \ell} \|\boldsymbol{y}\|_2 \|\boldsymbol{h}\|_2 \|\boldsymbol{x}\|_2. \tag{19}$$

*Proof.* We will prove (18). Proof of (19) is identical to proof of (18). Define $\boldsymbol{h}_0 = \boldsymbol{h}$, $\boldsymbol{x}_0 = \boldsymbol{x}$, $\boldsymbol{m}_0 = \boldsymbol{m}$, $\boldsymbol{y}_0 = \boldsymbol{y}$,

$$\boldsymbol{h}_d := \left( \prod_{i=d}^{1} \boldsymbol{W}_{i,+,\boldsymbol{h}}^{(1)} \right) \boldsymbol{h} = \left( \boldsymbol{W}_{d,+,\boldsymbol{h}}^{(1)} \boldsymbol{W}_{d-1,+,\boldsymbol{h}}^{(1)} \cdots \boldsymbol{W}_{1,+,\boldsymbol{h}}^{(1)} \right) \boldsymbol{h}$$
$$= \boldsymbol{W}_{d,+,\boldsymbol{h}}^{(1)} \boldsymbol{h}_{d-1}$$
$$= (\boldsymbol{W}_d^{(1)})_{+,\boldsymbol{h}_{d-1}} \boldsymbol{h}_{d-1},$$

and analogously $\boldsymbol{x}_d = \left( \prod_{i=d}^{1} \boldsymbol{W}_{i,+,\boldsymbol{x}}^{(1)} \right) \boldsymbol{x}$, $\boldsymbol{m}_s = \left( \prod_{i=s}^{1} \boldsymbol{W}_{i,+,\boldsymbol{m}}^{(2)} \right) \boldsymbol{m}$, and $\boldsymbol{y}_s = \left( \prod_{i=s}^{1} \boldsymbol{W}_{i,+,\boldsymbol{y}}^{(2)} \right) \boldsymbol{y}$. By the WDC, we have for all $\boldsymbol{h} \neq 0$, $\boldsymbol{m} \neq 0$,

$$\left\| \left( \boldsymbol{W}_i^{(1)} \right)_{+,\boldsymbol{h}}^{\mathsf{T}} \left( \boldsymbol{W}_i^{(1)} \right)_{+,\boldsymbol{h}} - \frac{1}{2} \boldsymbol{I}_{n_{i-1}} \right\| \leq \epsilon \text{ for all } i = 1, \dots, d-1, \text{ and} \tag{20}$$

$$\left\| \left( \boldsymbol{W}_i^{(2)} \right)_{+,\boldsymbol{m}}^{\mathsf{T}} \left( \boldsymbol{W}_i^{(2)} \right)_{+,\boldsymbol{m}} - \frac{1}{2} \boldsymbol{I}_{p_{i-1}} \right\| \leq \epsilon \text{ for all } i = 1, \dots, s-1. \tag{21}$$

In particular, $\left\| \left( \boldsymbol{W}_{i,+,\boldsymbol{h}}^{(1)} \right)^{\mathsf{T}} \boldsymbol{W}_{i,+,\boldsymbol{h}}^{(1)} - \frac{1}{2} \boldsymbol{I}_{n_{i-1}} \right\| \leq \epsilon$ and $\left\| \left( \boldsymbol{W}_{i,+,\boldsymbol{m}}^{(2)} \right)^{\mathsf{T}} \boldsymbol{W}_{i,+,\boldsymbol{m}}^{(2)} - \frac{1}{2} \boldsymbol{I}_{p_{i-1}} \right\| \leq \epsilon$. and consequently,

$$\frac{1}{2} - \epsilon \leq \left\| \boldsymbol{W}_{i,+,\boldsymbol{h}}^{(1)} \right\|^2 \leq \frac{1}{2} + \epsilon$$
$$\frac{1}{2} - \epsilon \leq \left\| \boldsymbol{W}_{i,+,\boldsymbol{m}}^{(2)} \right\|^2 \leq \frac{1}{2} + \epsilon.$$

Hence,

$$\left\| \prod_{i=d-1}^{1} \boldsymbol{W}_{i,+,\boldsymbol{h}}^{(1)} \right\| \left\| \prod_{i=d-1}^{1} \boldsymbol{W}_{i,+,\boldsymbol{x}}^{(1)} \right\| \leq \frac{1}{2^{d-1}} (1 + 2\epsilon)^{d-1} = \frac{1}{2^{d-1}} e^{(d-1)\log(1+2\epsilon)} \leq \frac{1 + 4\epsilon(d-1)}{2^{d-1}}, \tag{22}$$

where we used that $\log(1 + z) \leq z$, $e^z \leq 1 + 2z$ for $z < 1$, and $2(d-1)\epsilon \leq 1$. Similarly,

$$\left\| \prod_{i=s-1}^{1} \boldsymbol{W}_{i,+,\boldsymbol{m}}^{(2)} \right\| \left\| \prod_{i=s-1}^{1} \boldsymbol{W}_{i,+,\boldsymbol{y}}^{(2)} \right\| \leq \frac{1 + 4\epsilon(s-1)}{2^{s-1}}. \tag{23}$$

Let
$$\tilde{h} = \Lambda^{(1)}_{d-1,+,h}h, \quad \tilde{x} = \Lambda^{(1)}_{d-1,+,x}x, \quad \tilde{m} = \Lambda^{(2)}_{s-1,+,m}m, \quad \tilde{y} = \Lambda^{(2)}_{s-1,+,y}y,$$
and consider
$$\left\| \left(\Lambda^{(1)}_{d,+,h}\right)^{\mathsf{T}} \operatorname{diag}\left(\Lambda^{(2)}_{s,+,m}m \odot \Lambda^{(2)}_{s,+,y}y\right)\Lambda^{(1)}_{d,+,x}x - \frac{\alpha}{\ell}\left(m^{\mathsf{T}}\tilde{t}^{(2)}_{m,y}\right)\tilde{t}^{(1)}_{h,x} \right\|_2$$

$$\leq \left\| \left(\Lambda^{(1)}_{d-1,+,h}\right)^{\mathsf{T}}\left(\left(W^{(1)}_{d,+,h}\right)^{\mathsf{T}}\operatorname{diag}\left(W^{(2)}_{d,+,m}\tilde{m}\odot W^{(2)}_{d,+,y}\tilde{y}\right)W^{(1)}_{d,+,x} - \frac{\alpha}{\ell}Q_{\tilde{h},\tilde{x}} \right. \right.$$
$$\left. \tilde{m}^{\mathsf{T}}Q_{\tilde{m},\tilde{y}}\tilde{y}\right)\Lambda^{(1)}_{d-1,+,x}x \Big\|_2 + \left\| \frac{\alpha}{\ell}\left(\Lambda^{(1)}_{d-1,+,h}\right)^{\mathsf{T}}Q_{\tilde{h},\tilde{x}}\tilde{m}^{\mathsf{T}}Q_{\tilde{m},\tilde{y}}\tilde{y}\Lambda^{(1)}_{d-1,+,x}x \right.$$
$$\left. - \frac{\alpha}{\ell}\left(m^{\mathsf{T}}\tilde{t}^{(2)}_{m,y}\right)\tilde{t}^{(1)}_{h,x} \right\|_2$$

$$\leq \left\| \left(\Lambda^{(1)}_{d-1,+,h}\right)^{\mathsf{T}}\left(\left(W^{(1)}_{d,+,h}\right)^{\mathsf{T}}\operatorname{diag}\left(W^{(2)}_{d,+,m}\tilde{m}\odot W^{(2)}_{d,+,y}\tilde{y}\right)W^{(1)}_{d,+,x} - \frac{\alpha}{\ell}Q_{\tilde{h},\tilde{x}} \right. \right.$$
$$\left. \tilde{m}^{\mathsf{T}}Q_{\tilde{m},\tilde{y}}\tilde{y}\right)\Lambda^{(1)}_{d-1,+,x}x \Big\|_2$$
$$+ \frac{\alpha}{\ell}\left\| \tilde{m}^{\mathsf{T}}Q_{\tilde{m},\tilde{y}}\tilde{y}\left(\Lambda^{(1)}_{d-1,+,h}\right)^{\mathsf{T}}Q_{\tilde{h},\tilde{x}}\Lambda^{(1)}_{d-1,+,x}x - m^{\mathsf{T}}\tilde{t}^{(2)}_{m,y}\left(\Lambda^{(1)}_{d-1,+,h}\right)^{\mathsf{T}}Q_{\tilde{h},\tilde{x}}\Lambda^{(1)}_{d-1,+,x}x \right\|_2$$
$$+ \frac{\alpha}{\ell}\left\| m^{\mathsf{T}}\tilde{t}^{(2)}_{m,y}\left(\Lambda^{(1)}_{d-1,+,h}\right)^{\mathsf{T}}Q_{\tilde{h},\tilde{x}}\Lambda^{(1)}_{d-1,+,x}x - m^{\mathsf{T}}\tilde{t}^{(2)}_{m,y}\tilde{t}^{(1)}_{h,x} \right\|_2 \tag{24}$$

where both the first and second inequality holds because of triangle inequality. We bound the terms in the inequality above separately. First consider

$$\left\| \left(\Lambda^{(1)}_{d-1,+,h}\right)^{\mathsf{T}}\left(\left(W^{(1)}_{d,+,h}\right)^{\mathsf{T}}\operatorname{diag}\left(W^{(2)}_{d,+,m}\tilde{m}\odot W^{(2)}_{d,+,y}\tilde{y}\right)W^{(1)}_{d,+,x} - \frac{\alpha}{\ell}Q_{\tilde{h},\tilde{x}} \right. \right.$$
$$\left. \tilde{m}^{\mathsf{T}}Q_{\tilde{m},\tilde{y}}\tilde{y}\right)\Lambda^{(1)}_{d-1,+,x}x \Big\|_2$$
$$\leq \left\| \left(W^{(1)}_{d,+,h}\right)^{\mathsf{T}}\operatorname{diag}\left(W^{(2)}_{d,+,m}\tilde{m}\odot W^{(2)}_{d,+,y}\tilde{y}\right)W^{(1)}_{d,+,x} - \frac{\alpha}{\ell}Q_{\tilde{h},\tilde{x}}\tilde{m}^{\mathsf{T}}Q_{\tilde{m},\tilde{y}}\tilde{y} \right\|$$
$$\left\| \Lambda^{(1)}_{d-1,+,h} \right\| \left\| \Lambda^{(1)}_{d-1,+,x} \right\| \|x\|_2$$
$$\leq \left(\frac{1+4\epsilon(d-1)}{2^{d-1}}\right)\frac{4\epsilon}{\ell}\|x\|_2\|\tilde{m}\|_2\|\tilde{y}\|_2$$
$$= \frac{(1+4\epsilon(d-1))}{2^d}\frac{8\epsilon}{\ell}\|x\|_2\|\Lambda^{(2)}_{s-1,+,m}m\|_2\|\Lambda^{(2)}_{s-1,+,y}y\|_2$$
$$\leq \frac{(1+4\epsilon(d-1))}{2^d}\frac{(1+4\epsilon(s-1))}{2^s}\frac{16\epsilon}{\ell}\|x\|_2\|m\|_2\|y\|_2$$
$$\leq \frac{64\epsilon}{2^{d+s}\ell}\|x\|_2\|m\|_2\|y\|_2. \tag{25}$$

where the first inequality holds because spectral norm is a sub-multiplicative norm. The second inequality holds because of (22) and joint-WDC. The last inequality holds if $4\epsilon(d-1) < 1$ and $4\epsilon(s-1) < 1$.

Second, consider

$$\left\| \tilde{m}^{\mathsf{T}}Q_{\tilde{m},\tilde{y}}\tilde{y}\left(\Lambda^{(1)}_{d-1,+,h}\right)^{\mathsf{T}}Q_{\tilde{h},\tilde{x}}\Lambda^{(1)}_{d-1,+,x}x - m^{\mathsf{T}}\tilde{t}^{(2)}_{m,y}\left(\Lambda^{(1)}_{d-1,+,h}\right)^{\mathsf{T}}Q_{\tilde{h},\tilde{x}}\Lambda^{(1)}_{d-1,+,x}x \right\|_2$$
$$= \left\| \left(\tilde{m}^{\mathsf{T}}Q_{\tilde{m},\tilde{y}}\tilde{y} - m^{\mathsf{T}}\tilde{t}^{(2)}_{m,y}\right)\left(\Lambda^{(1)}_{d-1,+,h}\right)^{\mathsf{T}}Q_{\tilde{h},\tilde{x}}\Lambda^{(1)}_{d-1,+,x}x \right\|_2$$
$$\leq \frac{1+4\epsilon(d-1)}{2^{d-1}}\|Q_{\tilde{h},\tilde{x}}\|\left\| \left(\Lambda^{(2)}_{s-1,+,m}\right)^{\mathsf{T}}Q_{\tilde{m},\tilde{y}}\Lambda^{(2)}_{s-1,+,y}y - \tilde{t}^{(2)}_{m,y} \right\|_2\|x\|_2\|m\|_2$$
$$= \frac{1+4\epsilon(d-1)}{2^d}\left\| \left(\Lambda^{(2)}_{s-1,+,m}\right)^{\mathsf{T}}\left(Q_{\tilde{m},\tilde{y}} - \frac{1}{\alpha_2}\left(W^{(2)}_{d,+,m}\right)^{\mathsf{T}}W^{(2)}_{d,+,y}\right)\Lambda^{(2)}_{s-1,+,y}y \right.$$
$$\left. + \frac{1}{\alpha_2}\left(\Lambda^{(2)}_{s,+,m}\right)^{\mathsf{T}}\Lambda^{(2)}_{s,+,y}y - \tilde{t}^{(2)}_{m,y} \right\|_2\|x\|_2\|m\|_2$$
$$\leq \frac{1+4\epsilon(d-1)}{2^d}\left(\left\| \Lambda^{(2)}_{s-1,+,m} \right\|\left\| \Lambda^{(2)}_{s-1,+,y} \right\|\left\| \left(Q_{\tilde{m},\tilde{y}} - \frac{1}{\alpha_2}\left(W^{(2)}_{d,+,m}\right)^{\mathsf{T}}W^{(2)}_{d,+,y}\right) \right\|\|y\|_2 \right.$$

$$+ \left\| \frac{1}{\alpha_2} \left( \boldsymbol{\Lambda}_{s,+,\boldsymbol{m}}^{(2)} \right)^\mathsf{T} \boldsymbol{\Lambda}_{s,+,\boldsymbol{y}}^{(2)} \boldsymbol{y} - \tilde{t}_{\boldsymbol{m},\boldsymbol{y}}^{(2)} \right\|_2 \right) \|\boldsymbol{x}\|_2 \|\boldsymbol{m}\|_2$$

$$\leq \frac{1 + 4\epsilon(d-1)}{2^{d+s}} \left( 6(1 + 4\epsilon(s-1))\epsilon/\alpha_2 + 24 s^3 \sqrt{\epsilon/\alpha_2} \right) \|\boldsymbol{x}\|_2 \|\boldsymbol{m}\|_2 \|\boldsymbol{y}\|_2$$

$$\leq \frac{2}{2^{d+s}} \left( 12\epsilon/\alpha_2 + 24 s^3 \sqrt{\epsilon/\alpha_2} \right) \|\boldsymbol{x}\|_2 \|\boldsymbol{m}\|_2 \|\boldsymbol{y}\|_2$$

$$\leq \frac{72 s^3 \sqrt{\epsilon}}{2^{2d} \alpha_2} \|\boldsymbol{x}\|_2 \|\boldsymbol{m}\|_2 \|\boldsymbol{y}\|_2. \tag{26}$$

where the first inequality holds because of (22). The second inequality holds because of triangle inequality. The third inequality holds because of (23), $\frac{1}{\sqrt{\alpha_2}} \boldsymbol{W}_d^{(2)}$ satisfy WDC with constant $\epsilon/\alpha_2$ and 1, and Lemma 1.

Third, consider

$$\left\| \boldsymbol{m}^\mathsf{T} \tilde{t}_{\boldsymbol{m},\boldsymbol{y}}^{(2)} \left( \boldsymbol{\Lambda}_{d-1,+,\boldsymbol{h}}^{(1)} \right)^\mathsf{T} \boldsymbol{Q}_{\tilde{\boldsymbol{h}},\tilde{\boldsymbol{x}}} \boldsymbol{\Lambda}_{d-1,+,\boldsymbol{x}}^{(1)} \boldsymbol{x} - \boldsymbol{m}^\mathsf{T} \tilde{t}_{\boldsymbol{m},\boldsymbol{y}}^{(2)} \tilde{t}_{\boldsymbol{h},\boldsymbol{x}}^{(1)} \right\|_2$$

$$= |\boldsymbol{m}^\mathsf{T} \tilde{t}_{\boldsymbol{m},\boldsymbol{y}}^{(2)}| \left\| \left( \boldsymbol{\Lambda}_{d-1,+,\boldsymbol{h}}^{(1)} \right)^\mathsf{T} \boldsymbol{Q}_{\tilde{\boldsymbol{h}},\tilde{\boldsymbol{x}}} \boldsymbol{\Lambda}_{d-1,+,\boldsymbol{x}}^{(1)} \boldsymbol{x} - \tilde{t}_{\boldsymbol{h},\boldsymbol{x}}^{(1)} \right\|_2$$

$$\leq \|\tilde{t}_{\boldsymbol{m},\boldsymbol{y}}^{(2)}\|_2 \left\| \left( \boldsymbol{\Lambda}_{d-1,+,\boldsymbol{h}}^{(1)} \right)^\mathsf{T} \left( \boldsymbol{Q}_{\tilde{\boldsymbol{h}},\tilde{\boldsymbol{x}}} - \frac{1}{\alpha_1} \left( \boldsymbol{W}_{d,+,\boldsymbol{h}}^{(1)} \right)^\mathsf{T} \boldsymbol{W}_{d,+,\boldsymbol{x}}^{(1)} \right) \boldsymbol{\Lambda}_{d-1,+,\boldsymbol{x}}^{(1)} \boldsymbol{x} \right.$$

$$\left. + \frac{1}{\alpha_1} \left( \boldsymbol{\Lambda}_{d,+,\boldsymbol{h}}^{(1)} \right)^\mathsf{T} \boldsymbol{\Lambda}_{d,+,\boldsymbol{x}}^{(1)} - \tilde{t}_{\boldsymbol{h},\boldsymbol{x}}^{(1)} \right\|_2 \|\boldsymbol{m}\|_2$$

$$\leq \frac{1+s}{2^s} \left( \left\| \boldsymbol{\Lambda}_{d-1,+,\boldsymbol{h}}^{(1)} \right\| \left\| \boldsymbol{\Lambda}_{d-1,+,\boldsymbol{x}}^{(1)} \right\| \left\| \boldsymbol{Q}_{\tilde{\boldsymbol{h}},\tilde{\boldsymbol{x}}} - \frac{1}{\alpha_1} \left( \boldsymbol{W}_{d,+,\boldsymbol{h}}^{(1)} \right)^\mathsf{T} \boldsymbol{W}_{d,+,\boldsymbol{x}}^{(1)} \right\| \|\boldsymbol{x}\|_2 \right.$$

$$\left. + \left\| \frac{1}{\alpha_1} \left( \boldsymbol{\Lambda}_{d,+,\boldsymbol{h}}^{(1)} \right)^\mathsf{T} \boldsymbol{\Lambda}_{d,+,\boldsymbol{x}}^{(1)} \boldsymbol{x} - \tilde{t}_{\boldsymbol{h},\boldsymbol{x}}^{(1)} \right\|_2 \right) \|\boldsymbol{m}\|_2 \|\boldsymbol{y}\|_2$$

$$\leq \frac{2s}{2^{d+s}} \left( 6(1 + 4\epsilon(d-1))\epsilon/\alpha_1 + 24 d^3 \sqrt{\epsilon/\alpha_1} \right) \|\boldsymbol{x}\|_2 \|\boldsymbol{m}\|_2 \|\boldsymbol{y}\|_2$$

$$\leq \frac{72 s d^3 \sqrt{\epsilon}}{2^{d+s} \alpha_1} \|\boldsymbol{x}\|_2 \|\boldsymbol{m}\|_2 \|\boldsymbol{y}\|_2. \tag{27}$$

where the first inequality holds because of Cauchy-Schwartz inequality. The second inequality holds because of triangle inequality along with $\|\tilde{t}_{\boldsymbol{m},\boldsymbol{y}}^{(2)}\|_2 \leq \frac{1+s}{2^s}$. The third inequality holds because of (22), $\frac{1}{\sqrt{\alpha_1}} \boldsymbol{W}_d^{(1)}$ satisfy WDC with constant $\epsilon/\alpha_1$ and 1, and Lemma 1.

Hence, combining (24), (25), (26), and (27), we get

$$\left\| \left( \boldsymbol{\Lambda}_{d,+,\boldsymbol{h}}^{(1)} \right)^\mathsf{T} \text{diag} \left( \boldsymbol{\Lambda}_{s,+,\boldsymbol{m}}^{(2)} \boldsymbol{m} \odot \boldsymbol{\Lambda}_{s,+,\boldsymbol{y}}^{(2)} \boldsymbol{y} \right) \boldsymbol{\Lambda}_{d,+,\boldsymbol{x}}^{(1)} \boldsymbol{x} - \frac{\alpha_1 \alpha_2}{\ell} \left( \boldsymbol{m}^\mathsf{T} \tilde{t}_{\boldsymbol{m},\boldsymbol{y}}^{(2)} \right) \tilde{t}_{\boldsymbol{h},\boldsymbol{x}}^{(1)} \right\|_2$$

$$\leq \left( \frac{64\epsilon}{2^{d+s}\ell} + \frac{72 s^3 \sqrt{\epsilon}}{2^{d+s}\ell} + \frac{72 s d^3 \sqrt{\epsilon}}{2^{2d}\ell} \right) \|\boldsymbol{x}\|_2 \|\boldsymbol{m}\|_2 \|\boldsymbol{y}\|_2$$

$$\leq \frac{208 s^3 d^3 \sqrt{\epsilon}}{2^{d+s}\ell} \|\boldsymbol{x}\|_2 \|\boldsymbol{m}\|_2 \|\boldsymbol{y}\|_2.$$

$\square$

## 5.3 Zeros of $t_{(\boldsymbol{h},\boldsymbol{m}),(\boldsymbol{h}_0,\boldsymbol{m}_0)}$

**Lemma 3.** *Fix* $0 < \epsilon < 1$ *and* $0 < \alpha \leq 1$ *such that* $38(d^5 + s^5)\sqrt{\epsilon}/\alpha < 1$. *Let* $\mathcal{K} = \{(\boldsymbol{h}, \boldsymbol{0}) \in \mathbb{R}^{n \times p} | \boldsymbol{h} \in \mathbb{R}^n \} \cup \{(\boldsymbol{0}, \boldsymbol{m}) \in \mathbb{R}^{n \times p} | \boldsymbol{m} \in \mathbb{R}^p \}$. *Let*

$$S_{\epsilon,(\boldsymbol{h}_0,\boldsymbol{m}_0)}^{(1)} = \left\{ (\boldsymbol{h}, \boldsymbol{x}) \in \mathbb{R}^{n \times p} \backslash \mathcal{K} \left| \frac{\|t_{(\boldsymbol{h},\boldsymbol{m}),(\boldsymbol{h}_0,\boldsymbol{m}_0)}^{(1)}\|_2}{\|\boldsymbol{m}\|_2} \leq \frac{\epsilon \max(\|\boldsymbol{h}\|_2 \|\boldsymbol{m}\|_2, \|\boldsymbol{h}_0\|_2 \|\boldsymbol{m}_0\|_2)}{2^{d+s}\ell} \right. \right\},$$

$$S_{\epsilon,(\boldsymbol{h}_0,\boldsymbol{m}_0)}^{(2)} = \left\{ (\boldsymbol{h}, \boldsymbol{x}) \in \mathbb{R}^{n \times p} \backslash \mathcal{K} \left| \frac{\|t_{(\boldsymbol{h},\boldsymbol{m}),(\boldsymbol{h}_0,\boldsymbol{m}_0)}^{(2)}\|_2}{\|\boldsymbol{h},\|_2} \leq \frac{\epsilon \max(\|\boldsymbol{h}\|_2 \|\boldsymbol{m}\|_2, \|\boldsymbol{h}_0\|_2 \|\boldsymbol{m}_0\|_2)}{2^{d+s}\ell} \right. \right\},$$

*where $d$ and $s$ are an integers greater than 1. Let*

$$\tilde{t}_{m,y}^{(k)} = \frac{1}{2^{a^{(k)}}} \left( \prod_{i=0}^{a^{(k)}-1} \frac{\pi - \bar{\theta}_i^{(k)}}{\pi} y + \sum_{i=0}^{a^{(k)}-1} \frac{\sin \bar{\theta}_i^{(k)}}{\pi} \left( \prod_{j=i+1}^{a^{(k)}-1} \frac{\pi - \bar{\theta}_j^{(k)}}{\pi} \right) \frac{\|y\|_2}{\|m\|_2} m \right), \qquad (28)$$

*where $\bar{\theta}_i^{(k)} = g(\bar{\theta}_{i-1}^{(k)})$ for $g$ given in (8), $\bar{\theta}_0^{(k)} = \angle(m, y)$, $a^{(1)} = d$, and $a^{(2)} = s$. Let*
$t_{(h,m),(h_0,m_0)} = \begin{bmatrix} t_{(h,m),(h_0,m_0)}^{(1)} \\ t_{(h,m),(h_0,m_0)}^{(2)} \end{bmatrix}$ *where*

$$t_{(h,m),(h_0,m_0)}^{(1)} = \frac{\alpha}{2^{d+s}\ell} \|m\|_2^2 h - \frac{\alpha}{\ell} m^\intercal \tilde{t}_{m,m_0}^{(2)} \tilde{t}_{h,h_0}^{(1)}, \qquad (29)$$

$$t_{(h,m),(h_0,m_0)}^{(2)} = \frac{\alpha}{2^{d+s}\ell} \|h\|_2^2 m - \frac{\alpha}{\ell} h^\intercal \tilde{t}_{h,h_0}^{(1)} \tilde{t}_{m,m_0}^{(2)}. \qquad (30)$$

*Define*

$$\rho_{a^{(k)}}^{(k)} := \sum_{i=1}^{a^{(k)}-1} \frac{\sin \check{\theta}_i^{(k)}}{\pi} \left( \prod_{j=i+1}^{a^{(k)}-1} \frac{\pi - \check{\theta}_j^{(k)}}{\pi} \right),$$

*where $\check{\theta}_0^{(k)} = \pi$ and $\check{\theta}_i^{(k)} = g(\check{\theta}_{i-1}^{(k)})$. If $(h, m) \in S_{\epsilon,(h_0,m_0)}^{(1)} \cap S_{\epsilon,(h_0,m_0)}^{(2)}$ then one of the following holds:*

- $\left| \bar{\theta}_0^{(1)} \right| \le 2\sqrt{\epsilon}$, $\left| \bar{\theta}_0^{(2)} \right| \le 2\sqrt{\epsilon}$ and

$$\left| \|h\|_2 \|m\|_2 - \|h_0\|_2 \|m_0\|_2 \right| \le 145 \frac{ds\sqrt{\epsilon}}{\alpha} \|h_0\|_2 \|m_0\|_2,$$

- $\left| \bar{\theta}_0^{(1)} - \pi \right| \le 12\pi^2 d^3 \sqrt{\epsilon}/\alpha$, $\left| \bar{\theta}_0^{(2)} \right| \le 1.5\sqrt{\epsilon}$ and

$$\left| \|h\|_2 \|m\|_2 - \rho_d^{(1)} \|h_0\|_2 \|m_0\|_2 \right| \le 532 \frac{d^6 s \sqrt{\epsilon}}{\alpha} \|h_0\|_2 \|m_0\|_2,$$

- $\left| \bar{\theta}_0^{(1)} \right| \le 2\sqrt{\epsilon}$, $\left| \bar{\theta}_0^{(2)} - \pi \right| \le 12\pi^2 s^3 \sqrt{\epsilon}/\alpha$ and

$$\left| \|h\|_2 \|m\|_2 - \rho_d^{(1)} \|h_0\|_2 \|m_0\|_2 \right| \le 532 \frac{ds^6 \sqrt{\epsilon}}{\alpha} \|h_0\|_2 \|m_0\|_2,$$

- $\left| \bar{\theta}_0^{(1)} - \pi \right| \le 12\pi^2 d^3 \sqrt{\epsilon}\alpha$, $\left| \bar{\theta}_0^{(2)} - \pi \right| \le 12\pi^2 d^3 \sqrt{\epsilon}/\alpha$ and

$$\left| \|h\|_2 \|m\|_2 - \rho_d^{(1)} \rho_s^{(2)} \|h_0\|_2 \|m_0\|_2 \right| \le 3915 \frac{d^6 s^6 \sqrt{\epsilon}}{\alpha} \|h_0\|_2 \|m_0\|_2.$$

*In particular,*

$$S_{\epsilon,(h_0,m_0)}^{(1)} \cap S_{\epsilon,(h_0,m_0)}^{(2)} \subseteq \mathcal{A}_{437\frac{ds\sqrt{\epsilon}}{\alpha},(h_0,m_0)} \cup \mathcal{A}_{2436\pi\frac{d^6 s\sqrt{\epsilon}}{\alpha},\left(-\rho_d^{(1)} h_0, m_0\right)}$$

$$\cup \mathcal{A}_{2436\pi\frac{ds^6\sqrt{\epsilon}}{\alpha},\left(\rho_s^{(2)} h_0, -m_0\right)} \cup \mathcal{A}_{16767\pi^2\frac{d^6 s^6\sqrt{\epsilon}}{\alpha},\left(-\rho_d^{(1)} \rho_s^{(2)} h_0, -m_0\right)},$$

*where $\mathcal{A}_{\epsilon,(h_0,m_0)}$ is defined in (3).*

*Proof.* Without loss of generality, let $h_0 = e_1$, $m_0 = e_1$, $\hat{h} = \cos \bar{\theta}_0^{(1)} + \sin \bar{\theta}_0^{(1)}$ and $\hat{m} = \cos \bar{\theta}_0^{(2)} + \sin \bar{\theta}_0^{(2)}$ for some $\bar{\theta}_i^{(1)}, \bar{\theta}_0^{(2)} \in [0, \pi]$. First we introduce some notation for convenience. Let

$$\xi^{(k)} = \prod_{i=0}^{a^{(k)}-1} \frac{\pi - \bar{\theta}_i^{(k)}}{\pi}, \quad \zeta^{(k)} = \sum_{i=1}^{a^{(k)}-1} \frac{\sin \bar{\theta}_i^{(k)}}{\pi} \prod_{j=i+1}^{a^{(k)}-1} \frac{\pi - \bar{\theta}_j^{(k)}}{\pi},$$

$r^{(1)} = \|h\|_2$, $r^{(2)} = \|m\|_2$, and $M = \max(r^{(1)} r^{(2)}, 1)$. Using these notation, we can rewrite $t_{(h,m),(h_0,m_0)}^{(1)}$ as

$$t_{(h,m),(h_0,m_0)}^{(1)}$$

$$= \frac{\alpha}{2^{d+s}\ell}\left(\|\boldsymbol{m}\|_2\boldsymbol{h} - \left(\xi^{(2)}\cos\bar{\theta}_0^{(2)} + \zeta^{(2)}\right)\left(\xi^{(1)}\frac{\boldsymbol{h}_0}{\|\boldsymbol{h}_0\|_2} + \zeta^{(1)}\frac{\boldsymbol{h}}{\|\boldsymbol{h}\|_2}\right)\|\boldsymbol{h}_0\|_2\|\boldsymbol{m}_0\|_2\right)\|\boldsymbol{m}\|_2$$

$$= \frac{\alpha}{2^{d+s}\ell}\left(\|\boldsymbol{m}\|_2\boldsymbol{h} - \cos\bar{\theta}_s^{(2)}\left(\xi^{(1)}\frac{\boldsymbol{h}_0}{\|\boldsymbol{h}_0\|_2} + \zeta^{(1)}\frac{\boldsymbol{h}}{\|\boldsymbol{h}\|_2}\right)\|\boldsymbol{h}_0\|_2\|\boldsymbol{m}_0\|_2\right)\|\boldsymbol{m}\|_2$$

$$= \frac{\alpha}{2^{d+s}\ell}\left(r^{(1)}r^{(2)}\left(\cos\bar{\theta}_0^{(1)}\boldsymbol{e}_1 + \sin\bar{\theta}_0^{(1)}\boldsymbol{e}_2\right)\right.$$
$$\left. - \cos\bar{\theta}_s^{(2)}\left(\xi^{(1)}\boldsymbol{e}_1 + \zeta^{(1)}\left(\cos\bar{\theta}_0^{(1)}\boldsymbol{e}_1 + \sin\bar{\theta}_0^{(1)}\boldsymbol{e}_2\right)\right)\right)r^{(2)}.$$

By inspecting the components of $\boldsymbol{t}^{(1)}_{(\boldsymbol{h},\boldsymbol{m}),(\boldsymbol{h}_0,\boldsymbol{m}_0)}$, we have that $(\boldsymbol{h},\boldsymbol{m}) \in S^{(1)}_{\epsilon,(\boldsymbol{h}_0,\boldsymbol{m}_0)}$ implies

$$\left|r^{(1)}r^{(2)}\cos\bar{\theta}_0^{(1)} - \cos\bar{\theta}_s^{(2)}\left(\xi^{(1)} + \zeta^{(1)}\cos\bar{\theta}_0^{(1)}\right)\right| \le \frac{\epsilon M}{\alpha} \tag{31}$$

$$\left|r^{(1)}r^{(2)}\sin\bar{\theta}_0^{(1)} - \cos\bar{\theta}_s^{(2)}\zeta^{(1)}\sin\bar{\theta}_0^{(1)}\right| \le \frac{\epsilon M}{\alpha} \tag{32}$$

Similarly, by inspecting the components of $\boldsymbol{t}^{(2)}_{(\boldsymbol{h},\boldsymbol{m}),(\boldsymbol{h}_0,\boldsymbol{m}_0)}$, we have that $(\boldsymbol{h},\boldsymbol{m}) \in S^{(2)}_{\epsilon,(\boldsymbol{h}_0,\boldsymbol{m}_0)}$ implies

$$\left|r^{(1)}r^{(2)}\cos\bar{\theta}_0^{(2)} - \cos\bar{\theta}_d^{(1)}\left(\xi^{(2)} + \zeta^{(2)}\cos\bar{\theta}_0^{(2)}\right)\right| \le \frac{\epsilon M}{\alpha} \tag{33}$$

$$\left|r^{(1)}r^{(2)}\sin\bar{\theta}_0^{(2)} - \cos\bar{\theta}_d^{(1)}\zeta^{(2)}\sin\bar{\theta}_0^{(2)}\right| \le \frac{\epsilon M}{\alpha} \tag{34}$$

Now, we record several properties. We have:

$$\bar{\theta}_i^{(k)} \in [0,\pi/2] \text{ for } i \ge 1 \tag{35}$$

$$\bar{\theta}_i^{(k)} \le \bar{\theta}_{i-1}^{(k)} \text{ for } i \ge 1 \tag{36}$$

$$|\xi^{(k)}| \le 1 \tag{37}$$

$$\check{\theta}_i^{(k)} \le \frac{3\pi}{i+3} \text{ for } i \ge 0 \tag{38}$$

$$\check{\theta}_i^{(k)} \ge \frac{\pi}{i+1} \text{ for } i \ge 0 \tag{39}$$

$$\xi^{(k)} = \prod_{i=1}^{a^{(k)}-1}\frac{\pi - \bar{\theta}_i^{(k)}}{\pi} \ge \frac{\pi - \bar{\theta}_0^{(k)}}{\pi}{a^{(k)}}^{-3} \tag{40}$$

$$\bar{\theta}_0^{(k)} = \pi + O_1(\delta) \Rightarrow \bar{\theta}_i^{(k)} = \check{\theta}_i^{(k)} + O_1(i\delta) \tag{41}$$

$$\bar{\theta}_0^{(k)} = \pi + O_1(\delta) \Rightarrow |\xi^{(k)}| \le \frac{\delta}{\pi} \tag{42}$$

$$\bar{\theta}_0^{(k)} = \pi + O_1(\delta) \Rightarrow \zeta^{(k)} = \rho_d^{(k)} + O_1(3{a^{(k)}}^3\delta) \text{ if } \frac{{a^{(k)}}^2\delta}{\pi} \le 1 \tag{43}$$

$$|\zeta^{(k)}| = |\xi^{(k)}\cos\bar{\theta}_0^{(k)} - \cos\bar{\theta}_{a^{(k)}}^{(k)}| \le 2 \tag{44}$$

$$\cos\bar{\theta}_i^{(k)} \ge \frac{1}{\pi} \text{ for } i \ge 2 \tag{45}$$

For a proof of (37)-(43), we refer the readers to Lemma 8 of Hand and Voroninski [2017]. Also, we note that (45) follows directly from (44).

We first show that if $(\boldsymbol{h},\boldsymbol{m}) \in S^{(1)}_{\epsilon,(\boldsymbol{h}_0,\boldsymbol{m}_0)}$ then $r^{(1)}r^{(2)} \le 6$, and thus $M \le 6$. Suppose $r^{(1)}r^{(2)} > 1$. At least one of the following holds: $|\sin\bar{\theta}_0^{(1)}| \ge 1/\sqrt{2}$ or $|\cos\bar{\theta}_0^{(1)}| \ge 1/\sqrt{2}$. If $|\sin\bar{\theta}_0^{(1)}| \ge 1/\sqrt{2}$ then (32) implies that $\left|r^{(1)}r^{(2)} - \cos\bar{\theta}_s^{(2)}\zeta^{(1)}\right| \le \sqrt{2}\epsilon r^{(1)}r^{(2)}/\alpha$. Using (44), we get $r^{(1)}r^{(2)} \le \frac{2}{1-\sqrt{2}\epsilon/\alpha} \le 4$ if $\epsilon/\alpha < 1/4$. If $|\cos\bar{\theta}_0^{(1)}| \ge 1/\sqrt{2}$, then (31) implies $\left|r^{(1)}r^{(2)} - \cos\bar{\theta}_s^{(2)}\zeta^{(1)}\right| \le \sqrt{2}\left(\epsilon r^{(1)}r^{(2)}/\alpha + |\xi^{(1)}|\right)$. Using (37), (44), and $\epsilon/\alpha < 1/4$, we get

$r^{(1)}r^{(2)} \leq \frac{\sqrt{2}|\xi^{(k)}|+\cos\bar{\theta}_s^{(2)}\zeta^{(1)}}{1-\sqrt{2}\epsilon/\alpha} \leq \frac{2+\sqrt{2}}{1-\sqrt{2}\epsilon/\alpha} \leq 6$. Thus, we have $(\boldsymbol{h},\boldsymbol{m}) \in S_{\epsilon,(\boldsymbol{h}_0,\boldsymbol{m}_0)}^{(1)} \Rightarrow r^{(1)}r^{(2)} \leq 6 \Rightarrow M \leq 6$. Similarly, we have $(\boldsymbol{h},\boldsymbol{m}) \in S_{\epsilon,(\boldsymbol{h}_0,\boldsymbol{m}_0)}^{(2)} \Rightarrow r^{(1)}r^{(2)} \leq 6 \Rightarrow M \leq 6$.

Next we establish that we only need to consider the small angle case and the large angle case (i.e. $\bar{\theta}_0^{(k)} \approx 0$ or $\pi$) if $(\boldsymbol{h},\boldsymbol{m}) \in S_{\epsilon,(\boldsymbol{h}_0,\boldsymbol{m}_0)}^{(1)} \cap S_{\epsilon,(\boldsymbol{h}_0,\boldsymbol{m}_0)}^{(2)}$. Exactly one of the following holds: $\left|r^{(1)}r^{(2)} - \cos\bar{\theta}_s^{(2)}\zeta^{(1)}\right| \geq \sqrt{\epsilon}M/\alpha$ or $\left|r^{(1)}r^{(2)} - \cos\bar{\theta}_s^{(2)}\zeta^{(1)}\right| < \sqrt{\epsilon}M/\alpha$. If $\left|r^{(1)}r^{(2)} - \cos\bar{\theta}_s^{(2)}\zeta^{(1)}\right| \geq \sqrt{\epsilon}M/\alpha$, then by (32), we have $|\sin\bar{\theta}_0^{(1)}| \leq \sqrt{\epsilon}$. Hence $\bar{\theta}_0^{(1)} = O_1(2\sqrt{\epsilon})$ or $\bar{\theta}_0^{(1)} = \pi + O_1(2\sqrt{\epsilon})$, as $\epsilon < 1$. If $\left|r^{(1)}r^{(2)} - \cos\bar{\theta}_s^{(2)}\zeta^{(1)}\right| < \sqrt{\epsilon}M/\alpha$, then by (31) and (45) we have $\left|\xi^{(1)}\right| \leq 2\pi\sqrt{\epsilon}M/\alpha$. Using (40), we get $\bar{\theta}_0^{(1)} = \pi + O_1(2\pi^2 d^3\sqrt{\epsilon}M/\alpha)$. Thus, we only need to consider the small angle case, $\bar{\theta}_0^{(1)} = O_1(2\sqrt{\epsilon})$ and the large angle case $\bar{\theta}_0^{(1)} = \pi + O_1(12\pi^2 d^3\sqrt{\epsilon}/\alpha)$, where we have used $M \leq 6$. Similarly, we only need to consider the small angle case, $\bar{\theta}_0^{(2)} = O_1(2\sqrt{\epsilon})$ and the large angle case $\bar{\theta}_0^{(2)} = \pi + O_1(12\pi^2 s^3\sqrt{\epsilon}/\alpha)$.

**Case 1: $\bar{\theta}_0^{(1)} \approx 0$ and $\bar{\theta}_0^{(2)} \approx 0$** . Assume $\bar{\theta}_0^{(k)} = O_1(2\sqrt{\epsilon})$. As $\bar{\theta}_i^{(k)} \leq \bar{\theta}_0^{(k)} \leq 2\sqrt{\epsilon}$ for all $i$, we have $\xi^{(k)} \geq \left(1 - \frac{2\sqrt{\epsilon}}{\pi}\right)^{a^{(k)}} = 1 + O_1(\frac{4a^{(k)}\sqrt{\epsilon}}{\pi})$ provided $2a^{(k)}\sqrt{\epsilon} \leq 1/2$. By (44), we also have $\zeta^{(k)} = O_1(\frac{a^{(k)}}{\pi}2\sqrt{\epsilon}) = O_1(a^{(k)}\sqrt{\epsilon})$. By (31), we have

$$\left|r^{(1)}r^{(2)}\cos\bar{\theta}_0^{(1)} - \left(\xi^{(2)}\cos\bar{\theta}_0^{(2)} + \zeta^{(2)}\right)\left(\xi^{(1)} + \zeta^{(1)}\cos\bar{\theta}_0^{(1)}\right)\right| \leq \frac{\epsilon M}{\alpha}$$

where we used $\cos\bar{\theta}_{a^{(k)}}^{(k)} = \xi^{(k)}\cos\bar{\theta}_0^{(k)} + \zeta^{(k)}$. As $\cos\bar{\theta}_0^{(k)} = 1 + O_1((\bar{\theta}_0^{(k)})^2/2) = 1 + O_1(2\epsilon)$,

$$\xi^{(2)}\cos\bar{\theta}_0^{(2)} + \zeta^{(2)} = 1 + O_1(8s\epsilon\sqrt{\epsilon} + 4s\sqrt{\epsilon} + 2\epsilon + s\sqrt{\epsilon}) = 1 + O_1(15s\sqrt{\epsilon}),$$
$$\xi^{(1)} + \zeta^{(1)}\cos\bar{\theta}_0^{(1)} = 1 + O_1(4d\sqrt{\epsilon} + 2d\epsilon\sqrt{\epsilon} + d\sqrt{\epsilon}) = 1 + O_1(7d\sqrt{\epsilon}).$$

Thus,

$$r^{(1)}r^{(2)} = 1 + O_1(12\epsilon + 6\epsilon/\alpha + 105ds\epsilon + 7d\sqrt{\epsilon} + 15s\sqrt{\epsilon}) = 1 + O_1(145ds\sqrt{\epsilon}/\alpha). \qquad (46)$$

We now show $(\boldsymbol{h},\boldsymbol{m})$ is close to $\left(c\boldsymbol{h}_0, \frac{1}{c}\boldsymbol{m}_0\right)$, where $c = \frac{\|\boldsymbol{m}_0\|_2}{\|\boldsymbol{m}\|_2}$. Consider

$$\left\|\boldsymbol{h} - \frac{\|\boldsymbol{m}_0\|_2}{\|\boldsymbol{m}\|_2}\boldsymbol{h}_0\right\|_2$$
$$\leq \frac{1}{\|\boldsymbol{m}\|_2}\left(\left|\|\boldsymbol{h}\|_2\|\boldsymbol{m}\|_2 - \|\boldsymbol{m}_0\|_2\|\boldsymbol{h}_0\|_2\right| + (\|\boldsymbol{h}_0\|_2\|\boldsymbol{m}_0\|_2 + |\|\boldsymbol{h}\|_2\|\boldsymbol{m}\|_2 - \|\boldsymbol{m}_0\|_2\|\boldsymbol{h}_0\|_2|)\bar{\theta}_0^{(1)}\right)$$
$$\leq \frac{1}{\|\boldsymbol{m}\|_2}\left(145ds\sqrt{\epsilon}\|\boldsymbol{h}_0\|_2\|\boldsymbol{m}_0\|_2/\alpha + (\|\boldsymbol{h}_0\|_2\|\boldsymbol{m}_0\|_2 + 145ds\sqrt{\epsilon}\|\boldsymbol{h}_0\|_2\|\boldsymbol{m}_0\|_2/\alpha)\, 2\sqrt{\epsilon}\right)$$
$$\leq 437\frac{ds\sqrt{\epsilon}}{\alpha}\frac{\|\boldsymbol{h}_0\|_2\|\boldsymbol{m}_0\|_2}{\|\boldsymbol{m}\|_2}.$$

Similarly,

$$\left\|\boldsymbol{m} - \frac{\|\boldsymbol{m}\|_2}{\|\boldsymbol{m}_0\|_2}\boldsymbol{m}_0\right\|_2 \leq \left(\left|\|\boldsymbol{m}\|_2 - \|\boldsymbol{m}\|_2\right| + (\|\boldsymbol{m}\|_2 + |\|\boldsymbol{m}\|_2 - \|\boldsymbol{m}\|_2|)\bar{\theta}_0^{(2)}\right) \leq 2\sqrt{\epsilon}\|\boldsymbol{m}\|_2.$$

Hence,

$$\left\|(\boldsymbol{h},\boldsymbol{m}) - \left(c\boldsymbol{h}_0, \frac{1}{c}\boldsymbol{m}_0\right)\right\|_2 \leq 437\frac{ds\sqrt{\epsilon}}{\alpha}\left\|\left(c\boldsymbol{h}_0, \frac{1}{c}\boldsymbol{m}_0\right)\right\|_2.$$

**Case 2: $\bar{\theta}_0^{(1)} \approx \pi$ and $\bar{\theta}_0^{(2)} \approx 0$** Assume $\bar{\theta}_0^{(1)} = \pi + O_1(\delta)$ where $\delta = 12\pi^2 d^3\sqrt{\epsilon}/\alpha$. By (42) and (43), we have $\xi^{(1)} = O_1(\delta/\pi)$, and we have $\zeta^{(1)} = \rho_d^{(1)} + O_1(3d^3\delta)$ if $38d^5\sqrt{\epsilon}/\alpha \leq 1$. Also, assume

$\bar{\theta}_0^{(2)} = O_1(2\sqrt{\epsilon})$. As $\bar{\theta}_i^{(2)} \leq \bar{\theta}_0^{(2)} \leq 2\sqrt{\epsilon}$ for all $i$, we have $\xi^{(2)} \geq \left(1 - \frac{2\sqrt{\epsilon}}{\pi}\right)^s = 1 + O_1(\frac{4s\sqrt{\epsilon}}{\pi})$ provided $2s\sqrt{\epsilon} \leq 1/2$. By (44), we also have $\zeta^{(2)} = O_1(\frac{s}{\pi}2\sqrt{\epsilon}) = O_1(s\sqrt{\epsilon})$. By (33), we have

$$\left| r^{(1)}r^{(2)} \cos \bar{\theta}_0^{(2)} - \left(\xi^{(1)} \cos \bar{\theta}_0^{(1)} + \zeta^{(1)}\right)\left(\xi^{(2)} + \zeta^{(2)} \cos \bar{\theta}_0^{(2)}\right) \right| \leq \frac{\epsilon M}{\alpha}$$

where we used $\cos \bar{\theta}_{a^{(k)}}^{(k)} = \xi^{(k)} \cos \bar{\theta}_0^{(k)} + \zeta^{(k)}$. As $\cos \bar{\theta}_0^{(1)} = -1 + O((\bar{\theta}_0^{(1)} - \pi)^2/2) = -1 + O_1(\delta^2/2)$ provided $\delta < 1$ and $\cos \bar{\theta}_0^{(2)} = 1 + O_1((\bar{\theta}_0^{(2)})^2/2) = 1 + O_1(2\epsilon)$,

$$\xi^{(1)} \cos \bar{\theta}_0^{(1)} + \zeta^{(1)} = \rho_d^{(1)} + O_1\left(\frac{\delta^3}{2\pi} + \frac{\delta}{\pi} + 3d^3\delta\right) = \rho_d^{(1)} + O_1(4\delta d^3),$$

$$\xi^{(2)} + \zeta^{(2)} \cos \bar{\theta}_0^{(2)} = 1 + O_1(4s\sqrt{\epsilon} + 2s\epsilon\sqrt{\epsilon} + s\sqrt{\epsilon}) = 1 + O_1(7s\sqrt{\epsilon}).$$

Thus,

$$\begin{aligned} r^{(1)}r^{(2)} &= \rho_d^{(1)} + O_1(12\epsilon + 6\epsilon/\alpha + 4\delta d^3 + 7s\sqrt{\epsilon} + 28d^3 s\delta\sqrt{\epsilon}) \\ &= \rho_d^{(1)} + O_1(30d\sqrt{\epsilon}/\alpha + 4\delta d^3 + 28d^3 s\sqrt{\epsilon}) \\ &= \rho_d^{(1)} + O_1(532d^6 s\sqrt{\epsilon}/\alpha). \end{aligned}$$

where, in the second equality, we use $\delta < 1$. We now show $(\boldsymbol{h}, \boldsymbol{m})$ is close to $\left(-c\rho_d^{(1)}\boldsymbol{h}_0, \frac{1}{c}\boldsymbol{m}_0\right)$, where $c = \frac{\|\boldsymbol{m}_0\|_2}{\|\boldsymbol{m}\|_2}$. Consider

$$\left\| \boldsymbol{h} + \frac{\|\boldsymbol{m}_0\|_2}{\|\boldsymbol{m}\|_2}\rho_d^{(1)}\boldsymbol{h}_0 \right\|_2$$

$$\leq \frac{1}{\|\boldsymbol{m}\|_2}\left( \left| \|\boldsymbol{h}\|_2\|\boldsymbol{m}\|_2 - \rho_d^{(1)}\|\boldsymbol{h}_0\|_2\|\boldsymbol{m}_0\|_2 \right| + \left(\rho_d^{(1)}\|\boldsymbol{h}_0\|_2\|\boldsymbol{m}_0\|_2 + \|\boldsymbol{h}\|_2\|\boldsymbol{m}\|_2 \right. \right.$$

$$\left. \left. - \rho_d^{(1)}\|\boldsymbol{h}_0\|_2\|\boldsymbol{m}_0\|_2 \right| \right) \bar{\theta}_0^{(1)} \right)$$

$$\leq \frac{1}{\|\boldsymbol{m}\|_2}\left(532d^6 s\sqrt{\epsilon}\|\boldsymbol{m}_0\|_2\|\boldsymbol{h}_0\|_2/\alpha + \left(2\|\boldsymbol{h}_0\|_2\|\boldsymbol{m}_0\|_2 + 532d^6 s\sqrt{\epsilon}\|\boldsymbol{m}_0\|_2\|\boldsymbol{h}_0\|_2/\alpha\right)119d^3\sqrt{\epsilon}/\alpha\right)$$

$$\leq \frac{1}{\|\boldsymbol{m}\|_2}\left(532d^6 s\sqrt{\epsilon}\|\boldsymbol{m}_0\|_2\|\boldsymbol{h}_0\|_2/\alpha + \left(2\|\boldsymbol{h}_0\|_2\|\boldsymbol{m}_0\|_2 + 14s\|\boldsymbol{m}_0\|_2\|\boldsymbol{h}_0\|_2\right)119d^3\sqrt{\epsilon}/\alpha\right)$$

$$\leq 2436\pi \frac{d^6 s\sqrt{\epsilon}}{\alpha}\rho_d^{(1)}\frac{\|\boldsymbol{h}_0\|_2\|\boldsymbol{m}_0\|_2}{\|\boldsymbol{m}\|_2}.$$

Similarly,

$$\left\| \boldsymbol{m} - \frac{\|\boldsymbol{m}\|_2}{\|\boldsymbol{m}_0\|_2}\boldsymbol{m}_0 \right\|_2 \leq \left(\left| \|\boldsymbol{m}\|_2 - \|\boldsymbol{m}\|_2 \right| + \left(\|\boldsymbol{m}\|_2 + \left| \|\boldsymbol{m}\|_2 - \|\boldsymbol{m}\|_2 \right|\right)\bar{\theta}_0^{(2)}\right) \leq 2\sqrt{\epsilon}\|\boldsymbol{m}\|_2.$$

Hence,

$$\left\| (\boldsymbol{h}, \boldsymbol{m}) - \left(-c\rho_d^{(1)}\boldsymbol{h}_0, \frac{1}{c}\boldsymbol{m}_0\right) \right\|_2 \leq 2436\pi \frac{d^6 s\sqrt{\epsilon}}{\alpha}\left\| \left(-c\rho_d^{(1)}\boldsymbol{h}_0, \frac{1}{c}\boldsymbol{m}_0\right) \right\|_2.$$

**Case 3: $\bar{\theta}_0^{(1)} \approx 0$ and $\bar{\theta}_0^{(2)} \approx \pi$.** The analysis is similar to case 2. Using (31), we get

$$r^{(1)}r^{(2)} = \rho_s^{(2)} + O_1(532ds^6\sqrt{\epsilon}/\alpha).$$

Again, similar to case 2, we can show $(\boldsymbol{h}, \boldsymbol{m})$ is close to $\left(c\rho_s^{(2)}\boldsymbol{h}_0, -\frac{1}{c}\boldsymbol{m}_0\right)$, where $c = \frac{\|\boldsymbol{h}\|_2}{\|\boldsymbol{h}_0\|_2}$. We get,

$$\left\| (\boldsymbol{h}, \boldsymbol{m}) - \left(c\rho_s^{(2)}\boldsymbol{h}_0, -\frac{1}{c}\boldsymbol{m}_0\right) \right\|_2 \leq 2436\pi \frac{ds^6\sqrt{\epsilon}}{\alpha}\left\| \left(c\rho_s^{(2)}\boldsymbol{h}_0, -\frac{1}{c}\boldsymbol{m}_0\right) \right\|_2.$$

**Case 4: $\bar{\theta}_0^{(1)} \approx \pi$ and $\bar{\theta}_0^{(2)} \approx \pi$.** Assume $\bar{\theta}_0^{(k)} = \pi + O_1(\delta^{(k)})$ where $\delta^{(k)} = 12\pi^2 a^{(k)^3}\sqrt{\epsilon}/\alpha$. By (42) and (43), we have $\xi^{(k)} = O_1(\delta^{(k)}/\pi)$, and we have $\zeta^{(k)} = \rho_d^{(k)} + O_1(3a^{(k)^3}\delta^{(k)})$ if $\frac{a^{(k)^2}\delta}{\pi} \leq 1$. By (31), we have

$$\left| r^{(1)} r^{(2)} \cos\bar{\theta}_0^{(1)} - \left(\xi^{(2)}\cos\bar{\theta}_0^{(2)} + \zeta^{(2)}\right)\left(\xi^{(1)} + \zeta^{(1)}\cos\bar{\theta}_0^{(1)}\right)\right| \leq \frac{\epsilon M}{\alpha}$$

where we used $\cos\bar{\theta}_{a^{(k)}}^{(k)} = \xi^{(k)}\cos\bar{\theta}_0^{(k)} + \zeta^{(k)}$. As $\cos\bar{\theta}_0^{(k)} = -1 + O((\bar{\theta}_0^{(k)} - \pi)^2/2) = -1 + O_1((\delta^{(k)})^2/2)$,

$$\xi^{(2)}\cos\bar{\theta}_0^{(2)} + \zeta^{(2)} = \rho_s^{(2)} + O_1\left(\frac{(\delta^{(2)})^3}{2\pi} + \frac{\delta^{(2)}}{\pi} + 3s^3\delta^{(2)}\right) = \rho_s^{(2)} + O_1(4\delta^{(2)}s^3),$$

$$\xi^{(1)} + \zeta^{(1)}\cos\bar{\theta}_0^{(1)} = -\rho_d^{(1)} + O_1\left(\frac{\delta^{(1)}}{\pi} + \frac{3}{2}d^3(\delta^{(1)})^3 + 3\delta^{(1)}d^3\right) = -\rho_d^{(1)} + O_1(5\delta^{(1)}d^3).$$

Thus,

$$\begin{aligned}
r^{(1)}r^{(2)} &= \rho_d^{(1)}\rho_s^{(2)} + O_1(6\epsilon/\alpha + 4(\delta^{(1)})^2 + 4\delta^{(2)}s^3 + 5\delta^{(1)}d^3 + 20\delta^{(1)}\delta^{(2)}d^3s^3)\\
&= \rho_d^{(1)}\rho_s^{(2)} + O_1(6\epsilon/\alpha + 4\delta^{(1)} + 4\delta^{(2)}s^3 + 5\delta^{(1)}d^3 + 20\delta^{(2)}d^3s^3)\\
&= \rho_d^{(1)}\rho_s^{(2)} + O_1(6\epsilon/\alpha + 3909d^6s^6\sqrt{\epsilon}/\alpha)\\
&= \rho_d^{(1)}\rho_s^{(2)} + O_1(3915d^6s^6\sqrt{\epsilon}/\alpha),
\end{aligned}$$

where, in the second equality, we used $\delta^{(1)} \leq \frac{\pi}{d^2} < 1$. We now show $(h, m)$ is close to $\left(-c\rho_d^{(1)}\rho_s^{(2)}h_0, -\frac{1}{c}m_0\right)$, where $c = \frac{\|m_0\|_2}{\|m\|_2}$. Consider

$$\left\| h + \frac{\|m_0\|_2}{\|m\|_2}\rho_d^{(1)}\rho_s^{(2)}h_0 \right\|_2$$

$$\leq \frac{1}{\|m\|_2}\left(\left| \|h\|_2\|m\|_2 - \rho_d^{(1)}\rho_s^{(2)}\|m_0\|_2\|h_0\|_2\right| + \left(\rho_d^{(1)}\rho_s^{(2)}\|h_0\|_2\|m_0\|_2 + \right.\right.$$

$$\left.\left. \left| \|h\|_2\|m\|_2 - \rho_d^{(1)}\rho_s^{(2)}\|m_0\|_2\|h_0\|_2\right|\right)\bar{\theta}_0^{(1)}\right)$$

$$\leq \frac{1}{\|m\|_2}\left(3915d^6s^6\sqrt{\epsilon}\|h_0\|_2\|m_0\|_2/\alpha + \left(4\|h_0\|_2\|m_0\|_2\right.\right.$$

$$\left.\left. + 3915d^6s^6\sqrt{\epsilon}\|h_0\|_2\|m_0\|_2/\alpha\right)119d^3\sqrt{\epsilon}/\alpha\right)$$

$$\leq \frac{1}{\|m\|_2}\left(3915d^6s^6\sqrt{\epsilon}\|h_0\|_2\|m_0\|_2/\alpha + \left(4\|h_0\|_2\|m_0\|_2 + 104ds^6\|h_0\|_2\|m_0\|_2\right)119d^3\sqrt{\epsilon}/\alpha\right)$$

$$\leq 16767\pi^2\frac{d^6s^6\sqrt{\epsilon}}{\alpha}\rho_d^{(1)}\rho_s^{(2)}\frac{\|h_0\|_2\|m_0\|_2}{\|m\|_2}.$$

Similarly,

$$\left\| m + \frac{\|m\|_2}{\|m_0\|_2}m_0 \right\|_2 \leq \left(\left| \|m\|_2 - \|m\|_2\right| + (\|m\|_2 + \|m\|_2 - \|m\|_2)\bar{\theta}_0^{(2)}\right) \leq 119s^3\sqrt{\epsilon}\|m\|_2/\alpha.$$

Hence,

$$\left\| (h, m) - \left(-c\rho_d^{(1)}\rho_s^{(2)}h_0, -\frac{1}{c}m_0\right)\right\|_2 \leq 16767\pi^2\frac{d^6s^6\sqrt{\epsilon}}{\alpha}\left\| \left(c\rho_d^{(1)}\rho_s^{(2)}h_0, \frac{1}{c}m_0\right)\right\|_2.$$

$\square$

## 5.4 Proof of WDC condition

We first state a lemma that shows that the weight $W \in \mathbb{R}^{\ell \times n}$ of a layer of a neural network layer with i.i.d. $\mathcal{N}(0, 1/\ell)$ entries satisfies the WDC with constant $\epsilon$ and 1, and we refer the readers to Hand and Voroninski [2017] for a proof of the lemma.

**Lemma 4.** *Fix $0 < \epsilon < 1$. Let $\boldsymbol{W} \in \mathbb{R}^{\ell \times n}$ have i.i.d. $\mathcal{N}(0, 1/\ell)$ entries. If $\ell > cn \log n$, then with probability at least $1 - 8\ell e^{-\gamma n}$, $\boldsymbol{W}$ satisfies the WDC with constant $\epsilon$ and $1$. Here $c, \gamma^{-1}$ are constants that depend only polynomially on $\epsilon^{-1}$.*

We now state a lemma similar to Lemma 4 which applies to truncated random variable. The proof follows the proof of lemma 4 in Hand and Voroninski [2017].

**Lemma 5.** *Fix $0 < \epsilon < 1$. Let $\boldsymbol{W} \in \mathbb{R}^{\ell \times n}$ where ith row of $\boldsymbol{W}$ satisfy $\boldsymbol{w}_i^{\mathsf{T}} = \boldsymbol{w}^{\mathsf{T}} \cdot \boldsymbol{1}_{\|\boldsymbol{w}\|_2 \leq 3\sqrt{n/\ell}}$ and $\boldsymbol{w} \sim \mathcal{N}(\boldsymbol{0}, \frac{1}{\ell}\boldsymbol{I}_n)$. If $\ell > cn \log n$, then with probability at least $1 - 8ne^{-\gamma n}$, $\boldsymbol{W}$ satisfies the WDC with constant $\epsilon$ and $\alpha$. Here $c, \gamma^{-1}$ are constants that depend only polynomially on $\epsilon^{-1}$ and*

$$\alpha = \frac{\Gamma\left(\frac{n+2}{2}\right) - \Gamma\left(\frac{n+2}{2}, \frac{9n}{2}\right)}{\Gamma\left(\frac{n+2}{2}\right)}, \tag{47}$$

*where $\Gamma$ is the Gamma function.*

The WDC condition with constant $\epsilon$ and $\alpha$ can be written as

$$\|\boldsymbol{W}_{+,\boldsymbol{x}}^{\mathsf{T}} \boldsymbol{W}_{+,\boldsymbol{y}} - \alpha \boldsymbol{Q}_{\boldsymbol{x},\boldsymbol{y}}\| \leq \epsilon$$

for all nonzero $\boldsymbol{x}, \boldsymbol{y} \in \mathbb{R}^n$. We note that

$$\boldsymbol{W}_{+,\boldsymbol{x}}^{\mathsf{T}} \boldsymbol{W}_{+,\boldsymbol{y}} = \sum_{i=1}^{\ell} \boldsymbol{1}_{\boldsymbol{w}_i^{\mathsf{T}} \boldsymbol{x}} \boldsymbol{1}_{\boldsymbol{w}_i^{\mathsf{T}} \boldsymbol{y}} \boldsymbol{w}_i \boldsymbol{w}_i^{\mathsf{T}}$$

and it is not continuous in $\boldsymbol{x}$ and $\boldsymbol{y}$. So, we consider an arbitrarily good continuous approximation of $\boldsymbol{W}_{+,\boldsymbol{x}}^{\mathsf{T}} \boldsymbol{W}_{+,\boldsymbol{y}}$. Let

$$t_{-\epsilon}(z) = \begin{cases} 0 & z \leq -\epsilon, \\ 1 + \frac{z}{\epsilon} & -\epsilon \leq z \leq 0, \\ 1 & z \leq 0, \end{cases} \quad \text{and} \quad t_{\epsilon}(z) = \begin{cases} 0 & z \leq 0, \\ \frac{z}{\epsilon} & 0 \leq z \leq \epsilon, \\ 1 & z \geq \epsilon. \end{cases}$$

and define

$$H_{-\epsilon}(\boldsymbol{x}\boldsymbol{y}) := \sum_{i=1}^{\ell} t_{-\epsilon}(\boldsymbol{w}_i^{\mathsf{T}} \boldsymbol{x}) t_{-\epsilon}(\boldsymbol{w}_i^{\mathsf{T}} \boldsymbol{y}) \boldsymbol{w}_i \boldsymbol{w}_i^{\mathsf{T}},$$

$$H_{\epsilon}(\boldsymbol{x}, \boldsymbol{y}) := \sum_{i=1}^{\ell} t_{\epsilon}(\boldsymbol{w}_i^{\mathsf{T}} \boldsymbol{x}) t_{\epsilon}(\boldsymbol{w}_i^{\mathsf{T}} \boldsymbol{y}) \boldsymbol{w}_i \boldsymbol{w}_i^{\mathsf{T}}.$$

The proof of Lemma 5 follows from the follow two lemmas. We first provide an upper bound on the singular values of $H_{-\epsilon}(\boldsymbol{x}, \boldsymbol{y})$.

**Lemma 6.** *Fix $0 < \epsilon < 1$. Let $\boldsymbol{W} \in \mathbb{R}^{\ell \times n}$ where ith row of $\boldsymbol{W}$ satisfy $\boldsymbol{w}_i^{\mathsf{T}} = \boldsymbol{w}^{\mathsf{T}} \cdot \boldsymbol{1}_{\|\boldsymbol{w}\|_2 \leq 3\sqrt{n}}$ and $\boldsymbol{w} \sim \mathcal{N}(\boldsymbol{0}, \boldsymbol{I}_n)$. If $\ell > cn \log n$, then with probability at least $1 - 4ne^{-\gamma n}$,*

$$\forall (\boldsymbol{x}, \boldsymbol{y}) \neq (\boldsymbol{0}, \boldsymbol{0}), \quad H_{-\epsilon}(\boldsymbol{x}, \boldsymbol{y}) \preceq \alpha \ell \boldsymbol{Q}_{\boldsymbol{x},\boldsymbol{y}} + 3\ell\epsilon I_n.$$

*Here, $c$ and $\gamma^{-1}$ are constants that depend only polynomially on $\epsilon^{-1}$ and $\alpha$ is*

$$\alpha = \frac{\Gamma\left(\frac{n+2}{2}\right) - \Gamma\left(\frac{n+2}{2}, \frac{9n}{2}\right)}{\Gamma\left(\frac{n+2}{2}\right)}, \tag{48}$$

*where $\Gamma$ is the Gamma function.*

*Proof.* First we bound $\mathbb{E}[H_{-\epsilon}(\boldsymbol{x}, \boldsymbol{y})]$ for fixed $\boldsymbol{x}, \boldsymbol{y} \in \mathcal{S}^{n-1}$. Noting that $t_{-\epsilon}(z) \leq \boldsymbol{1}_{z \geq -\epsilon}(z) = \boldsymbol{1}_{z > 0}(z) + \boldsymbol{1}_{-\epsilon \leq z \leq 0}(z)$, we have

$$\mathbb{E}\left[H_{-\epsilon}(\boldsymbol{x}, \boldsymbol{y})\right]$$

$$\preceq \mathbb{E}\left[\sum_{i=1}^{\ell} \boldsymbol{1}_{\boldsymbol{w}_i^{\mathsf{T}} \boldsymbol{x} \geq -\epsilon} \boldsymbol{1}_{\boldsymbol{w}_i^{\mathsf{T}} \boldsymbol{y} \geq -\epsilon} \boldsymbol{w}_i \boldsymbol{w}_i^{\mathsf{T}}\right]$$

$$=\ell\mathbb{E}\left[\mathbf{1}_{\boldsymbol{w}_i^\mathsf{T}\boldsymbol{x}\geq-\epsilon}\mathbf{1}_{\boldsymbol{w}_i^\mathsf{T}\boldsymbol{y}\geq-\epsilon}\boldsymbol{w}_i\boldsymbol{w}_i^\mathsf{T}\right]$$

$$=\ell\mathbb{E}\left[\left(\mathbf{1}_{\boldsymbol{w}_i^\mathsf{T}\boldsymbol{x}\geq0}\mathbf{1}_{\boldsymbol{w}_i^\mathsf{T}\boldsymbol{y}\geq0}\boldsymbol{w}_i\boldsymbol{w}_i^\mathsf{T}\right)\right]+2\ell\mathbb{E}\left[\left(\mathbf{1}_{-\epsilon\leq\boldsymbol{w}_i^\mathsf{T}\boldsymbol{x}\leq0}\boldsymbol{w}_i\boldsymbol{w}_i^\mathsf{T}\right)\right].$$

We first note that $\mathbb{E}\left[\mathbf{1}_{\boldsymbol{w}_i^\mathsf{T}\boldsymbol{x}\geq0}\mathbf{1}_{\boldsymbol{w}_i^\mathsf{T}\boldsymbol{y}\geq0}\boldsymbol{b}_i\boldsymbol{b}_i^\mathsf{T}\right]=\alpha\boldsymbol{Q}_{\boldsymbol{x},\boldsymbol{y}}$ where $\alpha$ satisfies $0.97<\alpha<1$. Also, we have $\mathbb{E}\left[\mathbf{1}_{-\epsilon\leq\boldsymbol{w}_i^\mathsf{T}\boldsymbol{x}\leq0}\boldsymbol{w}_i\boldsymbol{w}_i^\mathsf{T}\right]\preceq\frac{\epsilon\alpha}{2}\boldsymbol{I}_n$. Thus,

$$\mathbb{E}\left[H_{-\epsilon}(\boldsymbol{x},\boldsymbol{y})\right]\preceq\alpha\ell\cdot\boldsymbol{m}^\mathsf{T}\boldsymbol{Q}_{\boldsymbol{x},\boldsymbol{y}}\boldsymbol{y}+\epsilon\alpha\ell\boldsymbol{I}_n$$
$$\preceq\alpha\ell\cdot\boldsymbol{m}^\mathsf{T}\boldsymbol{Q}_{\boldsymbol{x},\boldsymbol{y}}+\epsilon\ell\boldsymbol{I}_n \tag{49}$$

Second, we show concentration of $H_{-\epsilon}(\boldsymbol{x},\boldsymbol{y})$ for fixed $\boldsymbol{x},\boldsymbol{y}\in\mathcal{S}^{n-1}$. Let

$$\boldsymbol{\xi}_i=\sqrt{t_{-\epsilon}(\boldsymbol{w}_i^\mathsf{T}\boldsymbol{x})t_{-\epsilon}(\boldsymbol{w}_i^\mathsf{T}\boldsymbol{y})}\boldsymbol{w}_i.$$

We have

$$H_{-\epsilon}(\boldsymbol{x},\boldsymbol{y})-\mathbb{E}\left[H_{-\epsilon}(\boldsymbol{x},\boldsymbol{y})\right]$$
$$=\sum_{i=1}^{\ell}\left(t_{-\epsilon}(\boldsymbol{w}_i^\mathsf{T}\boldsymbol{x})t_{-\epsilon}(\boldsymbol{w}_i^\mathsf{T}\boldsymbol{y})\boldsymbol{w}_i\boldsymbol{w}_i^\mathsf{T}-\mathbb{E}\left[t_{-\epsilon}(\boldsymbol{w}_i^\mathsf{T}\boldsymbol{x})t_{-\epsilon}(\boldsymbol{w}_i^\mathsf{T}\boldsymbol{y})\boldsymbol{w}_i\boldsymbol{w}_i^\mathsf{T}\right]\right)$$
$$=\sum_{i=1}^{\ell}\left(\boldsymbol{\xi}_i\boldsymbol{\xi}_i^\mathsf{T}-\mathbb{E}\left[\boldsymbol{\xi}_i\boldsymbol{\xi}_i^\mathsf{T}\right]\right).$$

Note that $\boldsymbol{\xi}_i$ is sub-Gaussian for all $i$ and that the sub-Gaussian norm of $\boldsymbol{\xi}_i$ is bounded from above by an absolute constant which we call $K$. By first part of Remark 5.40 in Vershynin [2012], there exists a $c_K$ and $\gamma_K$ such that for all $t\geq0$, with probability at least $1-2e^{-\gamma_K t^2}$,

$$\|H_{-\epsilon}(\boldsymbol{x},\boldsymbol{y})-\mathbb{E}\left[H_{-\epsilon}(\boldsymbol{x},\boldsymbol{y})\right]\|\leq\max(\delta,\delta^2)\ell,\quad\text{where }\delta=c_K\sqrt{\frac{n}{\ell}}+\frac{t}{\sqrt{\ell}}.$$

If $\ell>(2c_K/\epsilon)^2n$, $t=\epsilon\sqrt{\ell}/2$, and $\epsilon<1$, we have

$$\|H_{-\epsilon}(\boldsymbol{x},\boldsymbol{y})-\mathbb{E}\left[H_{-\epsilon}(\boldsymbol{x},\boldsymbol{y})\right]\|\leq\epsilon\ell \tag{50}$$

with probability at least $1-2e^{-\gamma_K\frac{\epsilon^2\ell}{4}}$.

Third, we bound the Lipschitz constant of $H_{-\epsilon}$. For $\tilde{\boldsymbol{x}},\tilde{\boldsymbol{y}}\in\mathbb{R}^n$ we have

$$H_{-\epsilon}(\boldsymbol{x},\boldsymbol{y})-H_{-\epsilon}(\tilde{\boldsymbol{x}},\tilde{\boldsymbol{y}})$$
$$=\sum_{i=1}^{\ell}\left[t_{-\epsilon}(\boldsymbol{w}_i^\mathsf{T}\boldsymbol{x})t_{-\epsilon}(\boldsymbol{w}_i^\mathsf{T}\boldsymbol{y})-t_{-\epsilon}(\boldsymbol{w}_i^\mathsf{T}\tilde{\boldsymbol{x}})t_{-\epsilon}(\boldsymbol{w}_i^\mathsf{T}\tilde{\boldsymbol{y}})\right]\boldsymbol{w}_i\boldsymbol{w}_i^\mathsf{T}$$
$$=\sum_{i=1}^{\ell}\left[t_{-\epsilon}(\boldsymbol{w}_i^\mathsf{T}\boldsymbol{x})\left(t_{-\epsilon}(\boldsymbol{w}_i^\mathsf{T}\boldsymbol{y})-t_{-\epsilon}(\boldsymbol{w}_i^\mathsf{T}\tilde{\boldsymbol{y}})\right)\right.$$
$$\left.+t_{-\epsilon}(\boldsymbol{w}_i^\mathsf{T}\tilde{\boldsymbol{y}})\left(t_{-\epsilon}(\boldsymbol{w}_i^\mathsf{T}\boldsymbol{x})-t_{-\epsilon}(\boldsymbol{w}_i^\mathsf{T}\tilde{\boldsymbol{x}})\right)\right]\boldsymbol{w}_i\boldsymbol{w}_i^\mathsf{T}$$
$$=\boldsymbol{W}^\mathsf{T}\left[\text{diag}\left(t_{-\epsilon}(\boldsymbol{W}\boldsymbol{x})\right)\text{diag}((\boldsymbol{W}\boldsymbol{y})_+-(\boldsymbol{W}\tilde{\boldsymbol{y}})_+)\right.$$
$$\left.+\text{diag}\left(t_{-\epsilon}(\boldsymbol{W}\tilde{\boldsymbol{y}})\right)\text{diag}\left((\boldsymbol{W}\boldsymbol{x})_+-(\boldsymbol{W}\tilde{\boldsymbol{x}})_+\right)\right]\boldsymbol{W}$$

Thus,

$$\|H_{-\epsilon}(\boldsymbol{x},\boldsymbol{y})-H_{-\epsilon}(\tilde{\boldsymbol{x}},\tilde{\boldsymbol{y}})\|$$
$$\leq\|\boldsymbol{W}\|^2\left[\|t_{-\epsilon}(\boldsymbol{W}\boldsymbol{y})-t_{-\epsilon}(\boldsymbol{W}\tilde{\boldsymbol{y}})\|_\infty+\|t_{-\epsilon}(\boldsymbol{W}\boldsymbol{x})-t_{-\epsilon}(\boldsymbol{W}\tilde{\boldsymbol{x}})\|_\infty\right]$$
$$\leq\|\boldsymbol{W}\|^2\left[\max_{i\in[\ell]}|t_{-\epsilon}(\boldsymbol{w}_i^\mathsf{T}\boldsymbol{y})-t_{-\epsilon}(\boldsymbol{w}_i^\mathsf{T}\tilde{\boldsymbol{y}})|+\max_{i\in[\ell]}|t_{-\epsilon}(\boldsymbol{w}_i^\mathsf{T}\boldsymbol{x})-t_{-\epsilon}(\boldsymbol{w}_i^\mathsf{T}\tilde{\boldsymbol{x}})|\right]$$

$$\leq \|\boldsymbol{W}\|^2 \left[ \max_{i\in[\ell]} \frac{1}{\epsilon} |\boldsymbol{w}_i^\mathsf{T}(\boldsymbol{y}-\tilde{\boldsymbol{y}})| + \max_{i\in[\ell]} \frac{1}{\epsilon} |\boldsymbol{w}_i^\mathsf{T}(\boldsymbol{x}-\tilde{\boldsymbol{x}})| \right]$$

$$\leq \|\boldsymbol{W}\|^2 \left[ \frac{1}{\epsilon} \max_{i\in[\ell]} \|\boldsymbol{w}_i\|_2 \|\boldsymbol{y}-\tilde{\boldsymbol{y}}\| + \frac{1}{\epsilon} \max_{i\in[\ell]} \|\boldsymbol{w}_i\|_2 \|\boldsymbol{x}-\tilde{\boldsymbol{x}}\| \right]$$

$$\leq \|\boldsymbol{W}\|^2 \left[ \frac{9}{\epsilon}\sqrt{n} \|\boldsymbol{x}-\tilde{\boldsymbol{x}}\| + \frac{9}{\epsilon}\sqrt{n} \left\|\boldsymbol{h}-\tilde{\boldsymbol{h}}\right\| \right]$$

where the first inequality follows because $|t_{-\epsilon}(z)| \leq 1$ for all $z$, and the third inequality follows because $t_{-\epsilon}(z)$ is $1/\epsilon$-Lipschitz. Let $E_1$ be the event that $\|\boldsymbol{W}\| \leq 3\sqrt{\ell}$. By Corollary 5.35 in Vershynin [2012], for $\boldsymbol{A} \in \mathbb{R}^{\ell\times n}$ with rows of $\boldsymbol{A}$ following $\mathcal{N}(\boldsymbol{0},\boldsymbol{I}_n)$, we have $\mathbb{P}(\|\boldsymbol{A}\| \leq 3\sqrt{\ell}) \geq 1 - 2e^{-\ell/2}$, if $\ell \geq n$. As rows of $\boldsymbol{W}$ are truncated, we have $\mathbb{P}(E_1) \geq 1 - 2e^{-\ell/2}$, if $\ell \geq n$ as well. On $E_1$, we have

$$\|H_{-\epsilon}(\boldsymbol{x},\boldsymbol{y}) - H_{-\epsilon}(\tilde{\boldsymbol{x}},\tilde{\boldsymbol{y}})\|$$
$$\leq \frac{27\ell\sqrt{n}}{\epsilon} [\|\boldsymbol{x}-\tilde{\boldsymbol{y}}\| + \|\boldsymbol{y}-\tilde{\boldsymbol{y}}\|] \tag{51}$$

for all $\boldsymbol{x},\boldsymbol{y},\tilde{\boldsymbol{x}},\tilde{\boldsymbol{y}} \in \mathcal{S}^{n-1}$.

Finally, we complete the proof by a covering argument. Let $\mathcal{N}_\delta$ be a $\delta$-net on $\mathcal{S}^{n-1}$ such that $|\mathcal{N}_\delta| \leq (3/\delta)^n$. Take $\delta = \frac{\epsilon^2}{54\sqrt{n}}$. Combining (49) and (54), we have

$$\forall (\boldsymbol{x},\boldsymbol{y}), \in \mathcal{N}_\delta, \quad H_{-\epsilon}(\boldsymbol{x},\boldsymbol{y}) \preceq \mathbb{E}H_{-\epsilon}(\boldsymbol{x},\boldsymbol{y}) + \ell\epsilon I_n$$
$$\preceq \alpha\ell\boldsymbol{Q}_{\boldsymbol{x},\boldsymbol{y}} + 2\ell\epsilon I_n.$$

with probability at least

$$1 - 2|\mathcal{N}_\delta|e^{-\gamma_K\epsilon^2\ell/4} \geq 1 - 2\left(\frac{3}{\delta}\right)^n e^{-\gamma_K\epsilon^2\ell/4} \geq 1 - 2e^{-\gamma_K\epsilon^2\ell/4+n\log(3\cdot54\sqrt{n}/\epsilon^2)}.$$

If $\ell \geq \tilde{c}n\log(n)$ for some $\tilde{c} = \Omega(\epsilon^2\log\epsilon)$, then this probability is at least $1 - 2e^{-\tilde{\gamma}\ell}$ for some $\tilde{\gamma} = O(\epsilon^2)$. For $\boldsymbol{x},\boldsymbol{y} \in \mathcal{S}^{n-1}$, let $\tilde{\boldsymbol{x}},\tilde{\boldsymbol{y}} \in \mathcal{N}_\delta$ be such that $\|\boldsymbol{x}-\tilde{\boldsymbol{x}}\|_2 \leq \delta$, and $\|\boldsymbol{y}-\tilde{\boldsymbol{y}}\|_2 \leq \delta$. By (51), we have that

$$\forall \boldsymbol{x},\boldsymbol{y} \neq \boldsymbol{0}, \quad H_{-\epsilon}(\boldsymbol{x},\boldsymbol{y})$$
$$\preceq H_{-\epsilon}(\tilde{\boldsymbol{x}},\tilde{\boldsymbol{y}}) + \frac{27\ell\sqrt{n}}{\epsilon}2\delta\boldsymbol{I}_n$$
$$\preceq \alpha\ell\boldsymbol{Q}_{\boldsymbol{x},\boldsymbol{y}} + 3\ell\epsilon\boldsymbol{I}_n.$$

In conclusion, the result of this lemma holds if $\ell > (2c_K/\epsilon)^2 n$ and $\ell \geq \tilde{c}(n)\log n$, with probability at least $1 - 2e^{-\gamma_K\epsilon^2\ell/4} - 2e^{-\ell/2} - 2e^{-\tilde{\gamma}\ell} > 1 - 6e^{-\gamma\ell}$ for some $\gamma = O(\epsilon^2)$ and $\tilde{c} = \Omega(\epsilon^2\log\epsilon)$. $\square$

Next, we now provide an upper bound on the singular values of $G_\epsilon(\boldsymbol{h},\boldsymbol{x},\boldsymbol{m},\boldsymbol{y})$.

**Lemma 7.** *Fix $0 < \epsilon < 1$. Let $\boldsymbol{W} \in \mathbb{R}^{\ell\times n}$ where $i$th row of $\boldsymbol{W}$ satisfy $\boldsymbol{w}_i^\mathsf{T} = \boldsymbol{w}^\mathsf{T} \cdot \mathbf{1}_{\|\boldsymbol{w}\|_2 \leq 3\sqrt{n}}$ and $\boldsymbol{w} \sim \mathcal{N}(\boldsymbol{0},\boldsymbol{I}_n)$. If $\ell > cn\log n$, then with probability at least $1 - 4ne^{-\gamma n}$,*

$$\forall (\boldsymbol{x},\boldsymbol{y}) \neq (\boldsymbol{0},\boldsymbol{0}), \quad H_\epsilon(\boldsymbol{x},\boldsymbol{y}) \succeq \alpha\ell\boldsymbol{Q}_{\boldsymbol{x},\boldsymbol{y}} - 3\ell\epsilon I_n.$$

*Here, $c$ and $\gamma^{-1}$ are constants that depend only polynomially on $\epsilon^{-1}$ and $\alpha$ is*

$$\alpha = \frac{\Gamma\left(\frac{n+2}{2}\right) - \Gamma\left(\frac{n+2}{2},\frac{9n}{2}\right)}{\Gamma\left(\frac{n+2}{2}\right)}, \tag{52}$$

*where $\Gamma$ is the Gamma function.*

*Proof.* First we bound $\mathbb{E}[H_{-\epsilon}(\boldsymbol{x},\boldsymbol{y})]$ for fixed $\boldsymbol{x},\boldsymbol{y} \in \mathcal{S}^{n-1}$. Noting that $t_\epsilon(z) \geq \mathbf{1}_{z>0}(z) - \mathbf{1}_{-\epsilon\leq z\leq 0}(z)$ for all $z$, we have

$$\mathbb{E}[H_\epsilon(\boldsymbol{x},\boldsymbol{y})]$$

$$\succeq \ell \mathbb{E}\left[\left(\mathbf{1}_{\boldsymbol{w}_i^\intercal \boldsymbol{x} \geq 0}\mathbf{1}_{\boldsymbol{w}_i^\intercal \boldsymbol{y} \geq 0}\boldsymbol{w}_i\boldsymbol{w}_i^\intercal\right)\right] - 2\ell \mathbb{E}\left[\left(\mathbf{1}_{-\epsilon \leq \boldsymbol{w}_i^\intercal \boldsymbol{x} \leq 0}\boldsymbol{w}_i\boldsymbol{w}_i^\intercal\right)\right].$$

We first note that $\mathbb{E}\left[\mathbf{1}_{\boldsymbol{w}_i^\intercal \boldsymbol{x} \geq 0}\mathbf{1}_{\boldsymbol{w}_i^\intercal \boldsymbol{y} \geq 0}\boldsymbol{b}_i\boldsymbol{b}_i^\intercal\right] = \alpha \boldsymbol{Q}_{\boldsymbol{x},\boldsymbol{y}}$ where $\alpha$ satisfies $0.97 < \alpha < 1$. Also, we have $\mathbb{E}\left[\mathbf{1}_{-\epsilon \leq \boldsymbol{w}_i^\intercal \boldsymbol{x} \leq 0}\boldsymbol{w}_i\boldsymbol{w}_i^\intercal\right] \preceq \frac{\epsilon\alpha}{2}\boldsymbol{I}_n$. Thus,

$$\begin{aligned}\mathbb{E}\left[H_{-\epsilon}(\boldsymbol{x},\boldsymbol{y})\right] &\succeq \alpha\ell \cdot \boldsymbol{m}^\intercal \boldsymbol{Q}_{\boldsymbol{x},\boldsymbol{y}}\boldsymbol{y} - \epsilon\alpha\ell \boldsymbol{I}_n \\ &\succeq \alpha\ell \cdot \boldsymbol{m}^\intercal \boldsymbol{Q}_{\boldsymbol{x},\boldsymbol{y}} - \epsilon\ell \boldsymbol{I}_n\end{aligned} \tag{53}$$

Second, the same argument as in Lemma 6 provides that for fixed $\boldsymbol{x}, \boldsymbol{y} \in \mathcal{S}^{n-1}$, if $\ell > (2c_K/\epsilon)^2 n$, then we have with probability at least $1 - 2e^{-\gamma_K \frac{\epsilon^2 \ell}{4}}$,

$$\|H_{-\epsilon}(\boldsymbol{x},\boldsymbol{y}) - \mathbb{E}\left[H_{-\epsilon}(\boldsymbol{x},\boldsymbol{y})\right]\| \leq \epsilon\ell \tag{54}$$

Third, same argument as in Lemma 6 provides on the event $E_1$, we have

$$\|H_{-\epsilon}(\boldsymbol{x},\boldsymbol{y}) - H_{-\epsilon}(\tilde{\boldsymbol{x}},\tilde{\boldsymbol{y}})\| \leq \frac{27\ell\sqrt{n}}{\epsilon}\left[\|\boldsymbol{x} - \tilde{\boldsymbol{y}}\| + \|\boldsymbol{y} - \tilde{\boldsymbol{y}}\|\right]$$

for all $\boldsymbol{x}, \boldsymbol{y}, \tilde{\boldsymbol{x}}, \tilde{\boldsymbol{y}} \in \mathcal{S}^{n-1}$.

Finally, we complete the proof by an identical covering argument as in Lemma 6. We have if $\ell \geq c_0 n \log n$ then with probability at least $1 - 6e^{-\gamma\ell}$,

$$\forall \boldsymbol{x}, \boldsymbol{y} \neq \boldsymbol{0}, \quad H_\epsilon(\boldsymbol{x},\boldsymbol{y}) \succeq \alpha\ell \boldsymbol{Q}_{\boldsymbol{x},\boldsymbol{y}} - 3\ell\epsilon \boldsymbol{I}_n.$$

$\square$

## 5.5 Proof of joint-WDC condition

We now state a result that states random gaussian matrices with truncated rows satisfy joint-WDC.

**Lemma 8.** *Fix $0 < \epsilon < 1$. Let $\boldsymbol{B} \in \mathbb{R}^{\ell \times n}$ where ith row of $\boldsymbol{B}$ satisfy $\boldsymbol{b}_i^\intercal = \boldsymbol{b}^\intercal \cdot \mathbf{1}_{\|\boldsymbol{b}\|_2 \leq 3\sqrt{n/\ell}}$ and $\boldsymbol{b} \sim \mathcal{N}(\boldsymbol{0}, \boldsymbol{I}_n/\ell)$. Similarly, let $\boldsymbol{C} \in \mathbb{R}^{\ell \times p}$ where ith row of $\boldsymbol{C}$ satisfy $\boldsymbol{c}_i^\intercal = \boldsymbol{c}^\intercal \cdot \mathbf{1}_{\|\boldsymbol{c}\|_2 \leq 3\sqrt{p/\ell}}$ and $\boldsymbol{c} \sim \mathcal{N}(\boldsymbol{0}, \boldsymbol{I}_p/\ell)$. If $\ell > c((n \log n)^2 + (p \log p)^2)$, then with probability at least $1 - 8e^{-\gamma\ell/(n \log n)}$, $\boldsymbol{B}$ and $\boldsymbol{C}$ satisfy joint-WDC with constants $\epsilon$ and $\alpha = \alpha_1 \cdot \alpha_2$. Here, $c$ and $\gamma^{-1}$ are constants that depend only polynomially on $\epsilon^{-1}$ and*

$$\alpha_1 = \frac{\Gamma\left(\frac{n+2}{2}\right) - \Gamma\left(\frac{n+2}{2}, \frac{9n}{2}\right)}{\Gamma\left(\frac{n+2}{2}\right)} \text{ and } \alpha_2 = \frac{\Gamma\left(\frac{p+2}{2}\right) - \Gamma\left(\frac{p+2}{2}, \frac{9p}{2}\right)}{\Gamma\left(\frac{p+2}{2}\right)}, \tag{55}$$

*where $\Gamma$ is the Gamma function.*

The proof of Lemma 8 follows directly from Lemmas 9 and 10. Using Corollary 1, we provide a concentration result of $\boldsymbol{B}_{+,\boldsymbol{h}}^\intercal \mathrm{diag}(\boldsymbol{C}_{+,\boldsymbol{m}}\boldsymbol{m})\mathrm{diag}(\boldsymbol{C}_{+,\boldsymbol{y}}\boldsymbol{y})\boldsymbol{B}_{+,\boldsymbol{x}}$, which is part of the joint-WDC condition. We note that

$$\boldsymbol{B}_{+,\boldsymbol{h}}^\intercal \mathrm{diag}(\boldsymbol{C}_{+,\boldsymbol{m}}\boldsymbol{m})\mathrm{diag}(\boldsymbol{C}_{+,\boldsymbol{y}}\boldsymbol{y})\boldsymbol{B}_{+,\boldsymbol{x}} = \sum_{i=1}^{\ell} \mathbf{1}_{\boldsymbol{b}_i^\intercal \boldsymbol{h} > 0}\mathbf{1}_{\boldsymbol{b}_i^\intercal \boldsymbol{x} > 0}(\boldsymbol{c}_i^\intercal \boldsymbol{m})_+(\boldsymbol{c}_i^\intercal \boldsymbol{y})_+\boldsymbol{b}_i\boldsymbol{b}_i^\intercal$$

and it is not continuous in $\boldsymbol{h}$ and $\boldsymbol{x}$. So, we consider an arbitrarily good continuous approximation of $\boldsymbol{B}_{+,\boldsymbol{h}}^\intercal \mathrm{diag}(\boldsymbol{C}_{+,\boldsymbol{m}}\boldsymbol{m})\mathrm{diag}(\boldsymbol{C}_{+,\boldsymbol{y}}\boldsymbol{y})\boldsymbol{B}_{+,\boldsymbol{x}}$. Let

$$t_{-\epsilon}(z) = \begin{cases} 0 & z \leq -\epsilon, \\ 1 + \frac{z}{\epsilon} & -\epsilon \leq z \leq 0, \\ 1 & z \leq 0, \end{cases} \quad \text{and} \quad t_\epsilon(z) = \begin{cases} 0 & z \leq 0, \\ \frac{z}{\epsilon} & 0 \leq z \leq \epsilon, \\ 1 & z \geq \epsilon. \end{cases}$$

and define

$$G_{-\epsilon}(\boldsymbol{h}, \boldsymbol{x}, \boldsymbol{m}, \boldsymbol{y}) := \sum_{i=1}^{\ell} t_{-\epsilon}(\boldsymbol{b}_i^\intercal \boldsymbol{h})t_{-\epsilon}(\boldsymbol{b}_i^\intercal \boldsymbol{x})(\boldsymbol{c}_i^\intercal \boldsymbol{m})_+(\boldsymbol{c}_i^\intercal \boldsymbol{y})_+\boldsymbol{b}_i\boldsymbol{b}_i^\intercal,$$

$$G_\epsilon(\boldsymbol{h}, \boldsymbol{x}, \boldsymbol{m}, \boldsymbol{y}) := \sum_{i=1}^{\ell} t_\epsilon(\boldsymbol{b}_i^\intercal \boldsymbol{h})t_\epsilon(\boldsymbol{b}_i^\intercal \boldsymbol{x})(\boldsymbol{c}_i^\intercal \boldsymbol{m})_+(\boldsymbol{c}_i^\intercal \boldsymbol{y})_+\boldsymbol{b}_i\boldsymbol{b}_i^\intercal.$$

We now provide an upper bound on the singular values of $G_{-\epsilon}(\boldsymbol{h}, \boldsymbol{x}, \boldsymbol{m}, \boldsymbol{y})$.

**Lemma 9.** *Fix $0 < \epsilon < 1$. Let $\boldsymbol{B} \in \mathbb{R}^{\ell \times n}$ where ith row of $\boldsymbol{B}$ satisfy $\boldsymbol{b}_i^\mathsf{T} = \boldsymbol{b}^\mathsf{T} \cdot \mathbf{1}_{\|\boldsymbol{b}\|_2 \leq 3\sqrt{n}}$ and $\boldsymbol{b} \sim \mathcal{N}(\boldsymbol{0}, \boldsymbol{I}_n)$. Similarly, let $\boldsymbol{C} \in \mathbb{R}^{\ell \times p}$ where ith row of $\boldsymbol{C}$ satisfy $\boldsymbol{c}_i^\mathsf{T} = \boldsymbol{c}^\mathsf{T} \cdot \mathbf{1}_{\|\boldsymbol{c}\|_2 \leq 3\sqrt{p}}$ and $\boldsymbol{c} \sim \mathcal{N}(\boldsymbol{0}, \boldsymbol{I}_p)$. If $\ell > c((n \log n)^2 + (p \log p)^2)$, then with probability at least $1 - 4e^{-\gamma \ell / (n \log n)}$,*

$$\forall (\boldsymbol{h}, \boldsymbol{x}) \neq (\boldsymbol{0}, \boldsymbol{0}) \ and \ \boldsymbol{m}, \boldsymbol{y} \in \mathcal{S}^{p-1},$$
$$G_{-\epsilon}(\boldsymbol{h}, \boldsymbol{x}, \boldsymbol{m}, \boldsymbol{y}) \preceq \alpha_1 \alpha_2 \ell \boldsymbol{Q}_{\boldsymbol{h},\boldsymbol{x}} \boldsymbol{m}^\mathsf{T} \boldsymbol{Q}_{\boldsymbol{m},\boldsymbol{y}} \boldsymbol{y} + 4\ell\epsilon I_n.$$

*Here, $c$ and $\gamma^{-1}$ are constants that depend only polynomially on $\epsilon^{-1}$ and $\alpha_1$ and $\alpha_2$ is as in (55).*

*Proof.* First we bound $\mathbb{E}[G_{-\epsilon}(\boldsymbol{h}, \boldsymbol{x}, \boldsymbol{m}, \boldsymbol{y})]$ for fixed $\boldsymbol{h}, \boldsymbol{x} \in \mathcal{S}^{n-1}$ and $\boldsymbol{m}, \boldsymbol{y} \in \mathcal{S}^{p-1}$. Noting that $t_{-\epsilon}(z) \leq \mathbf{1}_{z \geq -\epsilon}(z) = \mathbf{1}_{z>0}(z) + \mathbf{1}_{-\epsilon \leq z \leq 0}(z)$, we have

$$\mathbb{E}\left[G_{-\epsilon}(\boldsymbol{h}, \boldsymbol{x}, \boldsymbol{m}, \boldsymbol{y})\right]$$

$$\preceq \mathbb{E}\left[\sum_{i=1}^{\ell} \mathbf{1}_{\boldsymbol{b}_i^\mathsf{T} \boldsymbol{h} \geq -\epsilon} \mathbf{1}_{\boldsymbol{b}_i^\mathsf{T} \boldsymbol{x} \geq -\epsilon} (\boldsymbol{c}_i^\mathsf{T} \boldsymbol{m})_+ (\boldsymbol{c}_i^\mathsf{T} \boldsymbol{y})_+ \boldsymbol{b}_i \boldsymbol{b}_i^\mathsf{T}\right]$$

$$= \ell \mathbb{E}\left[\mathbf{1}_{\boldsymbol{b}_i^\mathsf{T} \boldsymbol{h} \geq -\epsilon} \mathbf{1}_{\boldsymbol{b}_i^\mathsf{T} \boldsymbol{x} \geq -\epsilon} (\boldsymbol{c}_i^\mathsf{T} \boldsymbol{m})_+ (\boldsymbol{c}_i^\mathsf{T} \boldsymbol{y})_+ \boldsymbol{b}_i \boldsymbol{b}_i^\mathsf{T}\right]$$

$$\preceq \ell \mathbb{E}\left[\left(\mathbf{1}_{\boldsymbol{b}_i^\mathsf{T} \boldsymbol{h} \geq 0} \mathbf{1}_{\boldsymbol{b}_i^\mathsf{T} \boldsymbol{x} \geq 0} (\boldsymbol{c}_i^\mathsf{T} \boldsymbol{m})_+ (\boldsymbol{c}_i^\mathsf{T} \boldsymbol{y})_+ + \left(\mathbf{1}_{-\epsilon \leq \boldsymbol{b}_i^\mathsf{T} \boldsymbol{h} \leq 0} + \mathbf{1}_{-\epsilon \leq \boldsymbol{b}_i^\mathsf{T} \boldsymbol{x} \leq 0}\right)(\boldsymbol{c}_i^\mathsf{T} \boldsymbol{m})_+ (\boldsymbol{c}_i^\mathsf{T} \boldsymbol{y})_+\right) \boldsymbol{b}_i \boldsymbol{b}_i^\mathsf{T}\right]$$

$$= \ell \mathbb{E}\left[\left(\mathbf{1}_{\boldsymbol{b}_i^\mathsf{T} \boldsymbol{h} \geq 0} \mathbf{1}_{\boldsymbol{b}_i^\mathsf{T} \boldsymbol{x} \geq 0} \boldsymbol{b}_i \boldsymbol{b}_i^\mathsf{T}\right)(\boldsymbol{c}_i^\mathsf{T} \boldsymbol{m})_+ (\boldsymbol{c}_i^\mathsf{T} \boldsymbol{y})_+\right] + 2\ell \mathbb{E}\left[\left(\mathbf{1}_{-\epsilon \leq \boldsymbol{b}_i^\mathsf{T} \boldsymbol{h} \leq 0} \boldsymbol{b}_i \boldsymbol{b}_i^\mathsf{T}\right)(\boldsymbol{c}_i^\mathsf{T} \boldsymbol{m})_+ (\boldsymbol{c}_i^\mathsf{T} \boldsymbol{y})_+\right].$$

We first note that $\mathbb{E}\left[\mathbf{1}_{\boldsymbol{b}_i^\mathsf{T} \boldsymbol{h} \geq 0} \mathbf{1}_{\boldsymbol{b}_i^\mathsf{T} \boldsymbol{x} \geq 0} \boldsymbol{b}_i \boldsymbol{b}_i^\mathsf{T}\right] = \alpha_1 \boldsymbol{Q}_{\boldsymbol{h},\boldsymbol{x}}$ and $\mathbb{E}[(\boldsymbol{c}_i^\mathsf{T} \boldsymbol{m})_+ (\boldsymbol{c}_i^\mathsf{T} \boldsymbol{y})_+] = \alpha_2 \boldsymbol{m}^\mathsf{T} \boldsymbol{Q}_{\boldsymbol{m},\boldsymbol{y}} \boldsymbol{y}$ where $\alpha_i$ satisfies $0.97 < \alpha_i < 1$. Also, we have $\left|\boldsymbol{m}^\mathsf{T} \boldsymbol{Q}_{\boldsymbol{m},\boldsymbol{y}} \boldsymbol{y}\right| \leq \frac{1}{2}$ and $\mathbb{E}\left[\mathbf{1}_{-\epsilon \leq \boldsymbol{b}_i^\mathsf{T} \boldsymbol{h} \leq 0} \boldsymbol{b}_i \boldsymbol{b}_i^\mathsf{T}\right] \preceq \frac{\epsilon \alpha_1}{2} \boldsymbol{I}_n$. Thus,

$$\mathbb{E}\left[G_{-\epsilon}(\boldsymbol{h}, \boldsymbol{x}, \boldsymbol{m}, \boldsymbol{y})\right] \preceq \alpha_1 \alpha_2 \ell \cdot \boldsymbol{m}^\mathsf{T} \boldsymbol{Q}_{\boldsymbol{m},\boldsymbol{y}} \boldsymbol{y} \cdot \boldsymbol{Q}_{\boldsymbol{h},\boldsymbol{x}} + 2\ell \cdot \mathbb{E}\left[\mathbf{1}_{-\epsilon \leq \boldsymbol{b}_i^\mathsf{T} \boldsymbol{h} \leq 0} \boldsymbol{b}_i \boldsymbol{b}_i^\mathsf{T}\right] \cdot \alpha_2 \boldsymbol{m}^\mathsf{T} \boldsymbol{Q}_{\boldsymbol{m},\boldsymbol{y}} \boldsymbol{y}$$

$$\preceq \alpha_1 \alpha_2 \ell \cdot \boldsymbol{m}^\mathsf{T} \boldsymbol{Q}_{\boldsymbol{m},\boldsymbol{y}} \boldsymbol{y} \cdot \boldsymbol{Q}_{\boldsymbol{h},\boldsymbol{x}} + \frac{\epsilon \alpha_1 \alpha_2 \ell}{2} \boldsymbol{I}_n$$

$$\preceq \alpha_1 \alpha_2 \ell \cdot \boldsymbol{m}^\mathsf{T} \boldsymbol{Q}_{\boldsymbol{m},\boldsymbol{y}} \boldsymbol{y} \cdot \boldsymbol{Q}_{\boldsymbol{h},\boldsymbol{x}} + \frac{\epsilon \ell}{2} \boldsymbol{I}_n \quad (56)$$

Second, we show concentration of $G_{-\epsilon}(\boldsymbol{h}, \boldsymbol{x}, \boldsymbol{m}, \boldsymbol{y})$ for fixed $\boldsymbol{h}, \boldsymbol{x} \in \mathcal{S}^{n-1}$ and $\boldsymbol{m}, \boldsymbol{y} \in \mathcal{S}^{p-1}$. Let $\boldsymbol{\xi}_i = \sqrt{t_{-\epsilon}(\boldsymbol{b}_i^\mathsf{T} \boldsymbol{h}) t_{-\epsilon}(\boldsymbol{b}_i^\mathsf{T} \boldsymbol{x})(\boldsymbol{c}_i^\mathsf{T} \boldsymbol{m})_+ (\boldsymbol{c}_i^\mathsf{T} \boldsymbol{y})_+} \boldsymbol{b}_i$. We have

$$G_{-\epsilon}(\boldsymbol{h}, \boldsymbol{x}, \boldsymbol{m}, \boldsymbol{y}) - \mathbb{E}\left[G_{-\epsilon}(\boldsymbol{h}, \boldsymbol{x}, \boldsymbol{m}, \boldsymbol{y})\right]$$

$$= \sum_{i=1}^{\ell} \left(t_{-\epsilon}(\boldsymbol{b}_i^\mathsf{T} \boldsymbol{h}) t_{-\epsilon}(\boldsymbol{b}_i^\mathsf{T} \boldsymbol{x})(\boldsymbol{c}_i^\mathsf{T} \boldsymbol{m})_+ (\boldsymbol{c}_i^\mathsf{T} \boldsymbol{y})_+ \boldsymbol{b}_i \boldsymbol{b}_i^\mathsf{T} - \mathbb{E}\left[t_{-\epsilon}(\boldsymbol{b}_i^\mathsf{T} \boldsymbol{h}) t_{-\epsilon}(\boldsymbol{b}_i^\mathsf{T} \boldsymbol{x})(\boldsymbol{c}_i^\mathsf{T} \boldsymbol{m})_+ (\boldsymbol{c}_i^\mathsf{T} \boldsymbol{y})_+ \boldsymbol{b}_i \boldsymbol{b}_i^\mathsf{T}\right]\right)$$

$$= \sum_{i=1}^{\ell} \left(\boldsymbol{\xi}_i \boldsymbol{\xi}_i^\mathsf{T} - \mathbb{E}\left[\boldsymbol{\xi}_i \boldsymbol{\xi}_i^\mathsf{T}\right]\right).$$

Note that $\boldsymbol{\xi}_i$ is sub-Gaussian for all $i$ and that the sub-Gaussian norm of $\boldsymbol{\xi}_i$ is bounded from above by $K = \tilde{K}\sqrt{n}$, where $\tilde{K}$ is an absolute constant. By Corollary 1, there exists a $c = \bar{c}\sqrt{n} \log n$ and $\gamma = \frac{\bar{\gamma}}{n \log n}$ such that for all $t \geq 0$, with probability at least $1 - 2e^{-\gamma t^2}$,

$$\|G_{-\epsilon}(\boldsymbol{h}, \boldsymbol{x}, \boldsymbol{m}, \boldsymbol{y}) - \mathbb{E}\left[G_{-\epsilon}(\boldsymbol{h}, \boldsymbol{x}, \boldsymbol{m}, \boldsymbol{y})\right]\| \leq \max(\delta, \delta^2)\ell, \quad \text{where } \delta = c\sqrt{\frac{n}{\ell}} + \frac{t}{\sqrt{\ell}}.$$

Here, $\bar{c}$ and $\bar{\gamma}$ are absolute constants. If $\ell > (2\bar{c}/\epsilon)^2 n^2 (\log n)^2$, $t = \epsilon\sqrt{\ell}/2$, and $\epsilon < 1$, we have

$$\|G_{-\epsilon}(\boldsymbol{h}, \boldsymbol{x}, \boldsymbol{m}, \boldsymbol{y}) - \mathbb{E}\left[G_{-\epsilon}(\boldsymbol{h}, \boldsymbol{x}, \boldsymbol{m}, \boldsymbol{y})\right]\| \leq \epsilon\ell \quad (57)$$

with probability at least $1 - 2e^{-\bar{\gamma} \frac{\epsilon^2}{4} \frac{\ell}{n \log n}}$.

Third, we bound the Lipschitz constant of $G_{-\epsilon}$. For $\tilde{h}, \tilde{x} \in \mathbb{R}^n$ and $\tilde{m}, \tilde{y} \in \mathcal{S}^{p-1}$ we have

$$G_{-\epsilon}(h, x, m, y) - G_{-\epsilon}(\tilde{h}, \tilde{x}, \tilde{m}, \tilde{y})$$

$$= \sum_{i=1}^{\ell} \left[ t_{-\epsilon}(b_i^\mathsf{T} h) t_{-\epsilon}(b_i^\mathsf{T} x)(c_i^\mathsf{T} m)_+ (c_i^\mathsf{T} y)_+ - t_{-\epsilon}(b_i^\mathsf{T} \tilde{h}) t_{-\epsilon}(b_i^\mathsf{T} \tilde{x})(c_i^\mathsf{T} \tilde{m})_+ (c_i^\mathsf{T} \tilde{y})_+ \right] b_i b_i^\mathsf{T}$$

$$= \sum_{i=1}^{\ell} \left[ t_{-\epsilon}(b_i^\mathsf{T} h) t_{-\epsilon}(b_i^\mathsf{T} x)(c_i^\mathsf{T} m)_+ ((c_i^\mathsf{T} y)_+ - (c_i^\mathsf{T} \tilde{y})_+) \right.$$
$$+ t_{-\epsilon}(b_i^\mathsf{T} h) t_{-\epsilon}(b_i^\mathsf{T} x)(c_i^\mathsf{T} \tilde{y})_+ ((c_i^\mathsf{T} m)_+ - (c_i^\mathsf{T} \tilde{m})_+)$$
$$+ t_{-\epsilon}(b_i^\mathsf{T} h)(c_i^\mathsf{T} \tilde{m})_+ (c_i^\mathsf{T} \tilde{y})_+ (t_{-\epsilon}(b_i^\mathsf{T} x) - t_{-\epsilon}(b_i^\mathsf{T} \tilde{x}))$$
$$\left. + t_{-\epsilon}(b_i^\mathsf{T} \tilde{x})(c_i^\mathsf{T} \tilde{m})_+ (c_i^\mathsf{T} \tilde{y})_+ \left( t_{-\epsilon}(b_i^\mathsf{T} h) - t_{-\epsilon}(b_i^\mathsf{T} \tilde{h}) \right) \right] b_i b_i^\mathsf{T}$$

$$= B^\mathsf{T} \left[ \mathrm{diag}\left( t_{-\epsilon}(Bh) \odot t_{-\epsilon}(Bx) \odot (Cm)_+ \right) \mathrm{diag}((Cy)_+ - (C\tilde{y})_+) \right.$$
$$+ \mathrm{diag}\left( t_{-\epsilon}(Bh) \odot t_{-\epsilon}(Bx) \odot (C\tilde{y})_+ \right) \mathrm{diag}((Cm)_+ - (C\tilde{m})_+)$$
$$+ \mathrm{diag}\left( t_{-\epsilon}(Bh) \odot (C\tilde{m})_+ \odot (C\tilde{y})_+ \right) \mathrm{diag}(t_{-\epsilon}(Bx) - t_{-\epsilon}(B\tilde{x}))$$
$$\left. + \mathrm{diag}\left( t_{-\epsilon}(B\tilde{x}) \odot (C\tilde{m})_+ \odot (C\tilde{y})_+ \right) \mathrm{diag}\left( t_{-\epsilon}(Bh) - t_{-\epsilon}(B\tilde{h}) \right) \right] B$$

Thus,

$$\|G_{-\epsilon}(h, x, m, y) - G_{-\epsilon}(\tilde{h}, \tilde{x}, \tilde{m}, \tilde{y})\|$$

$$\leq \|B\|^2 \left[ \|Cm\|_\infty \|(Cy)_+ - (C\tilde{y})_+\|_\infty + \|C\tilde{y}\|_\infty \|(Cm)_+ - (C\tilde{m})_+\|_\infty \right.$$
$$\left. + \|C\tilde{m}\|_\infty \|C\tilde{y}\|_\infty \|t_{-\epsilon}(Bx) - t_{-\epsilon}(B\tilde{x})\|_\infty + \|C\tilde{m}\|_\infty \|C\tilde{y}\|_\infty \|t_{-\epsilon}(Bh) - t_{-\epsilon}(B\tilde{h})\|_\infty \right]$$

$$\leq \|B\|^2 \left[ \max_{i \in [\ell]} \|c_i\|_2 \max_{i \in [\ell]} |(c_i^\mathsf{T} y)_+ - (c_i^\mathsf{T} \tilde{y})_+| + \max_{i \in [\ell]} \|c_i\|_2 \max_{i \in [\ell]} |(c_i^\mathsf{T} m)_+ - (c_i^\mathsf{T} \tilde{m})_+| \right.$$
$$\left. + \left( \max_{i \in [\ell]} \|c_i\|_2 \right)^2 \max_{i \in [\ell]} |t_{-\epsilon}(b_i^\mathsf{T} x) - t_{-\epsilon}(b_i^\mathsf{T} \tilde{x})| + \left( \max_{i \in [\ell]} \|c_i\|_2 \right)^2 \max_{i \in [\ell]} \left| t_{-\epsilon}(b_i^\mathsf{T} h) - t_{-\epsilon}(b_i^\mathsf{T} \tilde{h}) \right| \right]$$

$$\leq \|B\|^2 \left[ \max_{i \in [\ell]} \|c_i\|_2 \max_{i \in [\ell]} |c_i^\mathsf{T}(y - \tilde{y})| + \max_{i \in [\ell]} \|c_i\|_2 \max_{i \in [\ell]} |c_i^\mathsf{T}(m - \tilde{m})| \right.$$
$$\left. + \left( \max_{i \in [\ell]} \|c_i\|_2 \right)^2 \max_{i \in [\ell]} \frac{1}{\epsilon} |b_i^\mathsf{T}(x - \tilde{x})| + \left( \max_{i \in [\ell]} \|c_i\|_2 \right)^2 \max_{i \in [\ell]} \frac{1}{\epsilon} \left| b_i^\mathsf{T}(h - \tilde{h}) \right| \right]$$

$$\leq \|B\|^2 \left[ \left( \max_{i \in [\ell]} \|c_i\|_2 \right)^2 \|y - \tilde{y}\| + \left( \max_{i \in [\ell]} \|c_i\|_2 \right)^2 \|m - \tilde{m}\| \right.$$
$$\left. + \frac{1}{\epsilon} \left( \max_{i \in [\ell]} \|c_i\|_2 \right)^2 \max_{i \in [\ell]} \|b_i\|_2 \|x - \tilde{x}\| + \frac{1}{\epsilon} \left( \max_{i \in [\ell]} \|c_i\|_2 \right)^2 \max_{i \in [\ell]} \|b_i\|_2 \left\| h - \tilde{h} \right\| \right]$$

$$\leq \|B\|^2 \left[ 9p \|y - \tilde{y}\| + 9p \|m - \tilde{m}\| + \frac{27}{\epsilon} \sqrt{np} \|x - \tilde{x}\| + \frac{27}{\epsilon} \sqrt{np} \left\| h - \tilde{h} \right\| \right]$$

where the first inequality follows because $|t_{-\epsilon}(z)| \leq 1$ for all $z$, and the third inequality follows because $t_{-\epsilon}(z)$ is $1/\epsilon$-Lipschitz and $(z)_+$ is 1-Lipschitz. Let $E_1$ be the event that $\|B\| \leq 3\sqrt{\ell}$. By Corollary 5.35 in Vershynin [2012], for $A \in \mathbb{R}^{\ell \times n}$ with rows of $A$ following $\mathcal{N}(0, I_n)$, we have $\mathbb{P}(\|A\| \leq 3\sqrt{\ell}) \geq 1 - 2e^{-\ell/2}$, if $\ell \geq n$. As rows of $B$ are truncated, we have $\mathbb{P}(E_1) \geq 1 - 2e^{-\ell/2}$, if $\ell \geq n$ as well. On $E_1$, we have

$$\|G_{-\epsilon}(h, x, m, y) - G_{-\epsilon}(\tilde{h}, \tilde{x}, \tilde{m}, \tilde{y})\|$$
$$\leq \frac{729\ell\sqrt{np}}{\epsilon} \left[ \|y - \tilde{y}\| + \|m - \tilde{m}\| + \|x - \tilde{x}\| + \left\| h - \tilde{h} \right\| \right] \tag{58}$$

for all $\tilde{h}, \tilde{x} \in \mathcal{S}^{n-1}$ and $\tilde{m}, \tilde{y} \in \mathcal{S}^{p-1}$.

Finally, we complete the proof by a covering argument. Let $\mathcal{N}_\delta$ be a $\delta$-net on $\mathcal{S}^{n-1} \times \mathcal{S}^{p-1}$ such that $|\mathcal{N}_\delta| \leq (3/\delta)^{n+p}$. Take $\delta = \frac{\epsilon^2}{2916\sqrt{np}}$. Combining (56) and (57), we have

$$\forall (\boldsymbol{h}, \boldsymbol{m}), (\boldsymbol{x}, \boldsymbol{y}) \in \mathcal{N}_\delta, \quad G_{-\epsilon}(\boldsymbol{h}, \boldsymbol{x}, \boldsymbol{m}, \boldsymbol{y}) \preceq \mathbb{E} G_{-\epsilon}(\boldsymbol{h}, \boldsymbol{x}, \boldsymbol{m}, \boldsymbol{y}) + \ell\epsilon I_n$$
$$\preceq \alpha^2 \ell \boldsymbol{Q}_{\boldsymbol{h},\boldsymbol{x}} \boldsymbol{m}^\intercal \boldsymbol{Q}_{\boldsymbol{m},\boldsymbol{y}} \boldsymbol{y} + 3\ell\epsilon I_n.$$

with probability at least

$$1 - 2|\mathcal{N}_\delta| e^{-\gamma_K \epsilon^2 \frac{\ell}{4n\log n}} \geq 1 - 2\left(\frac{3}{\delta}\right)^{n+p} e^{-\gamma_K \epsilon^2 \frac{\ell}{4n\log n}} \geq 1 - 2 e^{-\gamma_K \epsilon^2 \frac{\ell}{4n\log n} + (n+p)\log(3 \cdot 2916\sqrt{np}/\epsilon^2)}.$$

If $\ell \geq \tilde{c} n(n+p)\log(n)\log(np)$ for some $\tilde{c} = \Omega(\epsilon^2 \log \epsilon^{-1})$, then this probability is at least $1 - 2 e^{-\tilde{\gamma}\ell/(n\log(n))}$ for some $\tilde{\gamma} = O(\epsilon^2)$. For $(\boldsymbol{h}, \boldsymbol{m}), (\boldsymbol{x}, \boldsymbol{y}) \in \mathcal{S}^{n-1} \times \mathcal{S}^{p-1}$, let $(\tilde{\boldsymbol{h}}, \tilde{\boldsymbol{m}}), (\tilde{\boldsymbol{x}}, \tilde{\boldsymbol{y}}) \in \mathcal{N}_\delta$ be such that $\|\boldsymbol{h} - \tilde{\boldsymbol{h}}\|_2 \leq \delta$, $\|\boldsymbol{x} - \tilde{\boldsymbol{x}}\|_2 \leq \delta$, $\|\boldsymbol{m} - \tilde{\boldsymbol{m}}\|_2 \leq \delta$ and $\|\boldsymbol{y} - \tilde{\boldsymbol{y}}\|_2 \leq \delta$. By (58), we have that

$$\forall (\boldsymbol{h}, \boldsymbol{x}) \neq (\boldsymbol{0}, \boldsymbol{0}) \text{ and } \boldsymbol{m}, \boldsymbol{y} \in \mathcal{S}^{p-1}, \quad G_{-\epsilon}(\boldsymbol{h}, \boldsymbol{x}, \boldsymbol{m}, \boldsymbol{y})$$
$$\preceq G_{-\epsilon}(\tilde{\boldsymbol{h}}, \tilde{\boldsymbol{x}}, \tilde{\boldsymbol{m}}, \tilde{\boldsymbol{y}}) + \frac{729\ell\sqrt{np}}{\epsilon} 4\delta \boldsymbol{I}_n$$
$$\preceq \alpha_1\alpha_2 \ell \boldsymbol{Q}_{\boldsymbol{h},\boldsymbol{x}} \boldsymbol{m}^\intercal \boldsymbol{Q}_{\boldsymbol{m},\boldsymbol{y}} \boldsymbol{y} + 4\ell\epsilon \boldsymbol{I}_n.$$

In conclusion, the result of this lemma holds if $\ell > (2\bar{c}/\epsilon)^2 n^2 (\log n)^2$ and $\ell \geq \tilde{c} n(n+p)\log(n)\log(np)$, with probability at least $1 - 2 e^{-\ell/2} - 2 e^{-\tilde{\gamma}\ell/(n\log(n))} > 1 - 4 e^{-\gamma\ell/(n\log(n))}$ for some $\gamma = O(\epsilon^2)$ and $\tilde{c} = \Omega(\epsilon^2 \log \epsilon)$. □

Next, we now provide an upper bound on the singular values of $G_\epsilon(\boldsymbol{h}, \boldsymbol{x}, \boldsymbol{m}, \boldsymbol{y})$.

**Lemma 10.** *Fix $0 < \epsilon < 1$. Let $\boldsymbol{B} \in \mathbb{R}^{\ell \times n}$ where ith row of $\boldsymbol{B}$ satisfy $\boldsymbol{b}_i^\intercal = \boldsymbol{b}^\intercal \cdot \boldsymbol{1}_{\|\boldsymbol{b}\|_2 \leq 3\sqrt{n}}$ and $\boldsymbol{b} \sim \mathcal{N}(\boldsymbol{0}, \boldsymbol{I}_n)$. Similarly, let $\boldsymbol{C} \in \mathbb{R}^{\ell \times p}$ where ith row of $\boldsymbol{C}$ satisfy $\boldsymbol{c}_i^\intercal = \boldsymbol{c}^\intercal \cdot \boldsymbol{1}_{\|\boldsymbol{c}\|_2 \leq 3\sqrt{p}}$ and $\boldsymbol{c} \sim \mathcal{N}(\boldsymbol{0}, \boldsymbol{I}_p)$. If $\ell > c((n\log n)^2 + (p\log p)^2)$, then with probability at least $1 - 4 e^{-\gamma\ell/(n\log n)}$,*

$$\forall (\boldsymbol{h}, \boldsymbol{x}) \neq (\boldsymbol{0}, \boldsymbol{0}) \text{ and } \boldsymbol{m}, \boldsymbol{y} \in \mathcal{S}^{p-1},$$
$$G_\epsilon(\boldsymbol{h}, \boldsymbol{x}, \boldsymbol{m}, \boldsymbol{y}) \succeq \alpha_1\alpha_2 \ell \boldsymbol{Q}_{\boldsymbol{h},\boldsymbol{x}} \boldsymbol{m}^\intercal \boldsymbol{Q}_{\boldsymbol{m},\boldsymbol{y}} \boldsymbol{y} - 4\ell\epsilon I_n.$$

*Here, $c$ and $\gamma^{-1}$ are constants that depend only polynomially on $\epsilon^{-1}$ and $\alpha_1$ and $\alpha_2$ as in (55).*

*Proof.* First we bound $\mathbb{E}\left[G_\epsilon(\boldsymbol{h}, \boldsymbol{x}, \boldsymbol{m}, \boldsymbol{y})\right]$ for fixed $\boldsymbol{h}, \boldsymbol{x} \in \mathcal{S}^{n-1}$ and $\boldsymbol{m}, \boldsymbol{y} \in \mathcal{S}^{p-1}$. Noting that $t_\epsilon(z) \geq = \boldsymbol{1}_{z>0}(z) - \boldsymbol{1}_{0 \leq z \leq \epsilon}(z)$, we have

$$\mathbb{E}\left[G_\epsilon(\boldsymbol{h}, \boldsymbol{x}, \boldsymbol{m}, \boldsymbol{y})\right]$$
$$\succeq \ell\mathbb{E}\left[\left(\boldsymbol{1}_{\boldsymbol{b}_i^\intercal \boldsymbol{h} \geq 0}\boldsymbol{1}_{\boldsymbol{b}_i^\intercal \boldsymbol{x} \geq 0}(\boldsymbol{c}_i^\intercal \boldsymbol{m})_+(\boldsymbol{c}_i^\intercal \boldsymbol{y})_+ - \left(\boldsymbol{1}_{0 \leq \boldsymbol{b}_i^\intercal \boldsymbol{h} \leq \epsilon} + \boldsymbol{1}_{0 \leq \boldsymbol{b}_i^\intercal \boldsymbol{x} \leq \epsilon}\right)(\boldsymbol{c}_i^\intercal \boldsymbol{m})_+(\boldsymbol{c}_i^\intercal \boldsymbol{y})_+\right)\boldsymbol{b}_i\boldsymbol{b}_i^\intercal\right]$$
$$= \ell\mathbb{E}\left[\left(\boldsymbol{1}_{\boldsymbol{b}_i^\intercal \boldsymbol{h} \geq 0}\boldsymbol{1}_{\boldsymbol{b}_i^\intercal \boldsymbol{x} \geq 0}\boldsymbol{b}_i\boldsymbol{b}_i^\intercal\right)(\boldsymbol{c}_i^\intercal \boldsymbol{m})_+(\boldsymbol{c}_i^\intercal \boldsymbol{y})_+\right] - 2\ell\mathbb{E}\left[\left(\boldsymbol{1}_{0 \leq \boldsymbol{b}_i^\intercal \boldsymbol{h} \leq \epsilon}\boldsymbol{b}_i\boldsymbol{b}_i^\intercal\right)(\boldsymbol{c}_i^\intercal \boldsymbol{m})_+(\boldsymbol{c}_i^\intercal \boldsymbol{y})_+\right].$$

We first note that $\mathbb{E}\left[\boldsymbol{1}_{\boldsymbol{b}_i^\intercal \boldsymbol{h} \geq 0}\boldsymbol{1}_{\boldsymbol{b}_i^\intercal \boldsymbol{x} \geq 0}\boldsymbol{b}_i\boldsymbol{b}_i^\intercal\right] = \alpha_1\boldsymbol{Q}_{\boldsymbol{h},\boldsymbol{x}}$ and $\mathbb{E}\left[(\boldsymbol{c}_i^\intercal \boldsymbol{m})_+(\boldsymbol{c}_i^\intercal \boldsymbol{y})_+\right] = \alpha_2\boldsymbol{m}^\intercal \boldsymbol{Q}_{\boldsymbol{m},\boldsymbol{y}}\boldsymbol{y}$ where $\alpha_i$ satisfies $0.97 < \alpha_i < 1$. Also, we have $\left|\boldsymbol{m}^\intercal \boldsymbol{Q}_{\boldsymbol{m},\boldsymbol{y}}\boldsymbol{y}\right| \leq \frac{1}{2}$ and $\mathbb{E}\left[\boldsymbol{1}_{0 \leq \boldsymbol{b}_i^\intercal \boldsymbol{h} \leq \epsilon}\boldsymbol{b}_i\boldsymbol{b}_i^\intercal\right] \preceq \frac{\epsilon\alpha_1}{2}\boldsymbol{I}_n$. Thus,

$$\mathbb{E}\left[G_\epsilon(\boldsymbol{h}, \boldsymbol{x}, \boldsymbol{m}, \boldsymbol{y})\right] \succeq \alpha_1\alpha_2\ell \cdot \boldsymbol{m}^\intercal \boldsymbol{Q}_{\boldsymbol{m},\boldsymbol{y}}\boldsymbol{y} \cdot \boldsymbol{Q}_{\boldsymbol{h},\boldsymbol{x}} - 2\ell \cdot \mathbb{E}\left[\boldsymbol{1}_{0 \leq \boldsymbol{b}_i^\intercal \boldsymbol{h} \leq \epsilon}\boldsymbol{b}_i\boldsymbol{b}_i^\intercal\right] \cdot \alpha_2\boldsymbol{m}^\intercal \boldsymbol{Q}_{\boldsymbol{m},\boldsymbol{y}}\boldsymbol{y}$$
$$\succeq \alpha_1\alpha_2\ell \cdot \boldsymbol{m}^\intercal \boldsymbol{Q}_{\boldsymbol{m},\boldsymbol{y}}\boldsymbol{y} \cdot \boldsymbol{Q}_{\boldsymbol{h},\boldsymbol{x}} - \frac{\epsilon\alpha_1\alpha_2\ell}{2}\boldsymbol{I}_n$$
$$\succeq \alpha_1\alpha_2\ell \cdot \boldsymbol{m}^\intercal \boldsymbol{Q}_{\boldsymbol{m},\boldsymbol{y}}\boldsymbol{y} \cdot \boldsymbol{Q}_{\boldsymbol{h},\boldsymbol{x}} - \epsilon\ell \boldsymbol{I}_n \tag{59}$$

Second, we show concentration of $G_\epsilon(\boldsymbol{h}, \boldsymbol{x}, \boldsymbol{m}, \boldsymbol{y})$ for fixed $\boldsymbol{h}, \boldsymbol{x} \in \mathcal{S}^{n-1}$ and $\boldsymbol{m}, \boldsymbol{y} \in \mathcal{S}^{p-1}$ and is similar to the steps shown in proof of Lemma 9. Let $\boldsymbol{\xi}_i = \sqrt{t_\epsilon(\boldsymbol{b}_i^\intercal \boldsymbol{h})t_\epsilon(\boldsymbol{b}_i^\intercal \boldsymbol{x})(\boldsymbol{c}_i^\intercal \boldsymbol{m})_+(\boldsymbol{c}_i^\intercal \boldsymbol{y})_+}\boldsymbol{b}_i$. If $\ell > (2\bar{c}/\epsilon)^2 n^2 (\log n)^2$, we have

$$\|G_\epsilon(\boldsymbol{h}, \boldsymbol{x}, \boldsymbol{m}, \boldsymbol{y}) - \mathbb{E}\left[G_\epsilon(\boldsymbol{h}, \boldsymbol{x}, \boldsymbol{m}, \boldsymbol{y})\right]\| \leq \epsilon n \tag{60}$$

with probability at least $1 - 2e^{-\bar{\gamma}\frac{\epsilon^2}{4}\frac{\ell}{n\log n}}$. Here, $\bar{c}$ and $\bar{\gamma}$ are absolute constants.

Third, we bound the Lipschitz constant of $G_\epsilon$, and is again similar to the steps shown in proof of Lemma 9. If $\ell \geq n$ then we have

$$\|G_\epsilon(\boldsymbol{h}, \boldsymbol{x}, \boldsymbol{m}, \boldsymbol{y}) - G_\epsilon(\tilde{\boldsymbol{h}}, \tilde{\boldsymbol{x}}, \tilde{\boldsymbol{m}}, \tilde{\boldsymbol{y}})\|$$
$$\leq \frac{729\ell\sqrt{np}}{\epsilon} \left[ \|\boldsymbol{y} - \tilde{\boldsymbol{y}}\| + \|\boldsymbol{m} - \tilde{\boldsymbol{m}}\| + \|\boldsymbol{x} - \tilde{\boldsymbol{x}}\| + \left\|\boldsymbol{h} - \tilde{\boldsymbol{h}}\right\| \right] \tag{61}$$

for all $\tilde{\boldsymbol{h}}, \tilde{\boldsymbol{x}} \in \mathcal{S}^{n-1}$ and $\tilde{\boldsymbol{m}}, \tilde{\boldsymbol{y}} \in \mathcal{S}^{p-1}$ with probability at least $1 - 2e^{-\ell/2}$.

Finally, we complete the proof by a covering argument. Let $\mathcal{N}_\delta$ be a $\delta$-net on $\mathcal{S}^{n-1} \times \mathcal{S}^{p-1}$ such that $|\mathcal{N}_\delta| \leq (3/\delta)^{n+p}$. Take $\delta = \frac{\epsilon^2}{2916\sqrt{np}}$. Combining (59) and (60), we have

$$\forall (\boldsymbol{h}, \boldsymbol{m}), (\boldsymbol{x}, \boldsymbol{y}) \in \mathcal{N}_\delta, \quad G_\epsilon(\boldsymbol{h}, \boldsymbol{x}, \boldsymbol{m}, \boldsymbol{y}) \succeq \mathbb{E}G_\epsilon(\boldsymbol{h}, \boldsymbol{x}, \boldsymbol{m}, \boldsymbol{y}) - \ell\epsilon I_n$$
$$\succeq \alpha^2 \ell \boldsymbol{Q}_{\boldsymbol{h},\boldsymbol{x}} \boldsymbol{m}^\mathsf{T} \boldsymbol{Q}_{\boldsymbol{m},\boldsymbol{y}} \boldsymbol{y} - 3\ell\epsilon I_n$$

with probability at least

$$1 - 2|\mathcal{N}_\delta| e^{-\gamma_K \epsilon^2 \frac{\ell}{4n\log n}} \geq 1 - 2\left(\frac{3}{\delta}\right)^{n+p} e^{-\gamma_K \epsilon^2 \frac{\ell}{4n\log n}} \geq 1 - 2e^{-\gamma_K \epsilon^2 \frac{\ell}{4n\log n} + (n+p)\log(108\sqrt{np}/\epsilon^2)}.$$

If $\ell \geq \tilde{c}n(n+p)\log(n)\log(np)$ for some $\tilde{c} = \Omega(\epsilon^2 \log \epsilon)$, then this probability is at least $1 - 2e^{-\tilde{\gamma}\ell/(n\log n)}$ for some $\tilde{\gamma} = O(\epsilon^2)$. For $(\boldsymbol{h}, \boldsymbol{m}), (\boldsymbol{x}, \boldsymbol{y}) \in \mathcal{S}^{n-1} \times \mathcal{S}^{p-1}$, let $(\tilde{\boldsymbol{h}}, \tilde{\boldsymbol{m}}), (\tilde{\boldsymbol{x}}, \tilde{\boldsymbol{y}}) \in \mathcal{N}_\delta$ be such that $\|\boldsymbol{h} - \tilde{\boldsymbol{h}}\|_2 \leq \delta$, $\|\boldsymbol{x} - \tilde{\boldsymbol{x}}\|_2 \leq \delta$, $\|\boldsymbol{m} - \tilde{\boldsymbol{m}}\|_2 \leq \delta$ and $\|\boldsymbol{y} - \tilde{\boldsymbol{y}}\|_2 \leq \delta$. By (61), we have that

$$\forall (\boldsymbol{h}, \boldsymbol{x}) \neq (\boldsymbol{0}, \boldsymbol{0}) \text{ and } \boldsymbol{m}, \boldsymbol{y} \in \mathcal{S}^{p-1}, \quad G_\epsilon(\boldsymbol{h}, \boldsymbol{x}, \boldsymbol{m}, \boldsymbol{y})$$
$$\succeq \alpha_1 \alpha_2 \ell \boldsymbol{Q}_{\boldsymbol{h},\boldsymbol{x}} \boldsymbol{m}^\mathsf{T} \boldsymbol{Q}_{\boldsymbol{m},\boldsymbol{y}} \boldsymbol{y} - 4\ell\epsilon \boldsymbol{I}_n.$$

In conclusion, the result of this lemma holds if $\ell > (2\bar{c}/\epsilon)^2 n^2 (\log n)^2$ and $\ell \geq \tilde{c}n(n + p)\log(n)\log(np)$, with probability at least $1 - 2e^{-\ell/2} - 2e^{-\tilde{\gamma}\ell/(n\log n)} > 1 - 4e^{-\gamma\ell/(n\log n)}$ for some $\gamma = O(\epsilon^2)$ and $\tilde{c} = \Omega(\epsilon^2 \log \epsilon)$. □

### 5.6 Concentration of matrices with sub-gaussian rows

The proof of Lemmas 6 and 7 require results from concentration of sub-exponential random variables that has a better dependence on the sub-exponential parameters. To this end, we use the following Bernstein inequality and refer the readers to Jeong et al. [2019] for a proof of the theorem.

**Theorem 5.** *Let $\boldsymbol{a} = (a_1, \ldots, a_n)$ be a fixed non-zero vector and let $y_1, \ldots, y_m$ be independent, mean zero sub-exponential random variables satisfying $\mathbb{E}|y_i| \leq 2$ and $\|y_i\|_{\psi_1} \leq K_i^2$ ($K_i \geq 2$). Then for every $u \geq 0$, we have*

$$\mathbb{P}\left(\left|\sum_{i=1}^m a_i y_i\right| \geq u\right) \leq 2\exp\left[-c\min\left(\frac{u^2}{\sum_{i=1}^m a_i^2 K_i^2 \log K_i}, \frac{u}{\|\boldsymbol{a}\|_\infty K^2 \log K}\right)\right],$$

*where $K = \max_i K_i$ and $c$ is an absolute constant.*

We now state a theorem that controls the singular values of a random matrix $\boldsymbol{A}$. The Theorem is exactly the same as Theorem 5.39 in Vershynin [2012] with the notable difference in the dependence of the constants to the sub-gaussian parameters. We use Theorem 5 to get this improved dependence.

**Theorem 6.** *Let $\boldsymbol{A}$ be a $N \times n$ matrix whose rows $\boldsymbol{a}_i$ are independent sub-gaussian isotropic random vectors in $\mathbb{R}^n$. Then for every $t \geq 0$, with probability at least $1 - 2\exp(-ct^2)$ one has*

$$\sqrt{N} - C\sqrt{n} - t \leq s_{\min}(\boldsymbol{A}) \leq s_{\max}(\boldsymbol{A}) \leq \sqrt{N} + C\sqrt{n} + t. \tag{62}$$

*Here $C = C_K = K\sqrt{\log K}\sqrt{\frac{\log 9}{c_1}}$, $c = c_K = \frac{c_1}{K^2 \log K} > 0$ with $c_1$ is an absolute constant and $K = \max_i \|\boldsymbol{a}_i\|_{\psi_2}$.*

The proof structure of Theorem 6 is exactly the same as the proof of Theorem 5.39 in Vershynin [2012], and so we provide the proof presented in Vershynin [2012] below.

*Proof.* The proof is a basic version of a covering argunemt, and it has three steps. We need to control $\|\boldsymbol{A}\boldsymbol{x}\|_2$ for all vectors on the unit sphere. To this end, we discretize the sphere using a $\mathcal{N}$ (the approximation step), establish a tight control of $\|\boldsymbol{A}\boldsymbol{x}\|_2$ for every fixed vector $\boldsymbol{x} \in \mathcal{N}$ with high probability (the concentration step), and finish off by taking a union bound over all $\boldsymbol{x}$ in the net.

**Step 1: Approximation**. Using Lemma 5.36 in Vershynin [2012] for the matrix $\boldsymbol{B} = \boldsymbol{A}/\sqrt{N}$ we see that the conclusion of the theorem is equivalent to

$$\|\tfrac{1}{N}\boldsymbol{A}^\intercal\boldsymbol{A} - \boldsymbol{I}\| \leq \max(\delta, \delta^2) =: \epsilon \text{ where } \delta = C\sqrt{\frac{n}{N}} + \frac{t}{\sqrt{N}}. \tag{63}$$

Using Lemma 5.34 in Vershynin [2012], we can evaluate the operator norm in (63) on a $\frac{1}{4}$-net $\mathcal{N}$ pf unit sphere $\mathcal{S}^{n-1}$:

$$\|\frac{1}{N}\boldsymbol{A}^\intercal\boldsymbol{A} - \boldsymbol{I}\| \leq 2\max_{\boldsymbol{x}\in\mathcal{N}}\left|\left\langle(\frac{1}{N}\boldsymbol{A}^\intercal\boldsymbol{A} - \boldsymbol{I})\boldsymbol{x}, \boldsymbol{x}\right\rangle\right| = 2\max_{\boldsymbol{x}\in\mathcal{N}}|\frac{1}{N}\|\boldsymbol{A}\boldsymbol{x}\|_2^2 - 1|.$$

So to complete the proof it suffices to show that, which high probability,

$$\max_{\boldsymbol{x}\in\mathcal{N}}|\frac{1}{N}\|\boldsymbol{A}\|_2^2 - 1| \leq \frac{\epsilon}{2}. \tag{64}$$

By Lemma 5.2 in Vershynin [2012], we can choose the net $\mathcal{N}$ so that it has cardinality $|\mathcal{N}| \leq 9^n$.

**Step 2: Concentration** Let us fix any vector $\boldsymbol{x} \in \mathcal{S}^{n-1}$. We can express $\|\boldsymbol{A}\boldsymbol{x}\|_2^2$ as a sum of independent random variablies

$$\|\boldsymbol{A}\boldsymbol{x}\|_2^2 = \sum_{i=1}^{N}\langle\boldsymbol{a}_i, \boldsymbol{x}\rangle =: \sum_{i=1}^{N} z_i^2 \tag{65}$$

where $\boldsymbol{a}_i$ denote the rows of the matrix $\boldsymbol{A}$. By assumption, $z_i = \langle\boldsymbol{a}_i, \boldsymbol{x}\rangle$ are independent sub-gaussian random variables with $\mathbb{E}z_i^2 = 1$ and $\|z_i\|_{\psi_2} \leq K$. Therefore, by Remark 5.18 and Lemma 5.14 in Vershynin [2012], $z_i^2 - 1$ are independent centered sub-exponential random variables with $\|z_i^2 - 1\|_{\psi_1} \leq 2\|z_i^2\|_{\psi_1} \leq 4\|z_i\|_{\psi_2}^2 = 4K^2$.

We can therefore use an exponential deviation inequality, Theorem 5, to control the sum (65).

$$\begin{aligned}
\mathbb{P}\left\{|\frac{1}{N}\|\boldsymbol{A}\|_2^2 - 1| \geq \frac{\epsilon}{2}\right\} &= \mathbb{P}\left\{|\frac{1}{N}\sum_{i=1}^{N} z_i^2 - 1| \geq \frac{\epsilon}{2}\right\} \\
&\leq 2\exp\left[-\tilde{c}_1\min\left(\frac{\epsilon^2 N^2/4}{\sum_{i=1}^{N} 4K_i^2\log 2K_i}, \frac{\epsilon N/2}{4K\log 2K}\right)\right] \\
&\leq 2\exp\left[-\frac{\tilde{c}_1}{4K^2\log 2K}\min(\epsilon^2, \epsilon)N\right] \\
&= 2\exp\left[-\frac{\tilde{c}_1}{4K^2\log 2K}\delta N\right] \\
&\leq 2\exp\left[-\frac{c_1}{K^2\log K}(C^2 n + t^2)\right],
\end{aligned}$$

where the last inequality follows by the definition of $\delta$ and using the inequality $(a+b)^2 \geq a^2 + b^2$ for $a, b \geq 0$.

**Step 3: Union bound.** Taking the union bound over all vectors $\boldsymbol{x}$ in the net $\mathcal{N}$ of cardinality $|\mathcal{N}| \leq 9^n$, we obtain

$$\mathbb{P}\left\{\max_{\boldsymbol{x}\in\mathcal{N}}|\frac{1}{N}\|\boldsymbol{A}\boldsymbol{x}\|_2^2 - 1| \geq \frac{\epsilon}{2}\right\} \leq 9^n \cdot 2\exp[-\frac{c_1}{K^2\log K}(C^2 n + t^2)] \leq 2\exp[-\frac{c_1}{K^2\log K}t^2],$$

where the second inequality follows for $C = C_K$ sufficiently large, e.g. $C = K\sqrt{\log K}\sqrt{\frac{\log 9}{c_1}}$. $\quad\square$

We now state a corollary of Theorem 6 that applies to general, non-isotropic sub-gaussian distribution.

**Corollary 1.** *Let $A$ be a $N \times n$ matrix whose rows $a_i$ are independent sub-gaussian random vectors in $\mathbb{R}^n$ with second moment matrix $\Sigma$. Then for every $t \geq 0$, with probability at least $1 - 2\exp(-ct^2)$ one has*

$$\|\tfrac{1}{N}A^\mathsf{T}A - \Sigma\| \leq \max(\delta, \delta^2) \text{ where } \delta = C\sqrt{\frac{n}{N}} + \frac{t}{\sqrt{N}}. \tag{66}$$

*Here $C = C_K = K\sqrt{\log K}\sqrt{\frac{\log 9}{c_1}}$, $c = c_K = \frac{c_1}{K^2 \log K} > 0$ with $c_1$ is an absolute constant and $K = \max_i \|a_i\|_{\psi_2}$.*