[Reviews · NeurIPS 2019]

Reviewer 1



The authors solve the problem of blind demodulation, prevalent in several inverse imaging applications. The main contribution of this paper is to identify the conditions under which the directional derivative of the loss function is strictly negative, with the minimum occurring at the ground truth values of the forward mapping. Therefore a simple gradient descent based approach is sufficient to ensure global convergence. The authors combine two conditions Weight Distribution Condition, WDC, a variant of which first appears in [Hand, Voroninski 2017] and a new joint-WDC developed in the context of the demodulation problem. As long as these two conditions are satisfied, this paper shows that the global convergence criterion can be met. Additionally they show that Gaussian and truncated Gaussian weight matrices meet these conditions. Bulk of the paper rests on the theoretical exposition. The paper provides a good description of prior work. I did no go through the theoretical proofs from the supplementary material in detail, however the proof outline from the main paper provides sufficient intuition. Experimental results are limited and look promising, however it is unclear if the settings of the main theorem are validated or not (eg. WDC for the weights of the trained generator). Moreover, in lines [64-70] authors claim that using generative priors can correspond to lower measurement/dimensionality requirements as compared to sparsity priors. However I do not see any experimental validation, in terms of comparisons, for this claim. I think this is a drawback, because the whole motivation behind using generative priors rests on the fact that it may correspond to lower sample requirements, and without useful comparisons, it is unclear what the benefit of using generative priors is in comparison to conventional methods. Post author feedback: I have read the author's comments and factored in other reviewers' comments as well. The theoretical analysis in this paper is mostly convincing and assumptions for analysis have either been properly stated or clarified in the author feedback. I still believe that empirical validation is missing; however that does not negate other technical contributions of this paper, which is reflected in my original score. My assessment of the quality of this paper stays the same; I believe this is a good submission and should be accepted for publication.

Reviewer 2



The general spirit of the paper points in a fruitful direction: understanding the implication of deep generative models for solving inverse problems. As the paper notes, these models are increasingly widely used, and often lead to better practical performance compared to more traditional structural signal models (sparsity, smoothness, etc.). The paper performs a geometric analysis of this problem, arguing that for random, expansive deep generative models, the landscape of optimization is essentially dictated by the scale ambiguity in the BIP (1). This is, in some sense expected, because under the stated conditions, the deep generative models are near-isometric, but showing this through a rigorous analysis is a nontrivial technical problem. There is some question regarding the relevance of the setup considered here to practical bilinear inverse problems. Models for physical phenomena usually have additional symmetries (e.g., most reasonable models for images are shift invariant), leading to a more complicated landscape of optimization. The paper can be viewed as a first step in this direction: to my knowledge these are the first theoretical results on bilinear problems with deep generative models. The introduction (e.g., lines 58-70) compares the paper’s results to the literature on e.g., sparse phase retrieval. It is not clear whether a general comparison of this kind makes sense: whether a sparse model or a deep generative model exhibits better dimension-accuracy tradeoffs depends on the signal being modeled. It is also unclear whether the $n^2 + p^2$ scaling is necessary in this setting. The setting shares some commonalities with deconvolution under random subspace models (ala Romberg). In that problem a quadratic dependence is unnecessary. A2 and A3 are conditions of the theorem, but are not stated with sufficient precision (what does it mean for “n_i = \Omega(n_{i-1}) up to a log factor?”). The theorems are not stated as cleanly as they could be. E.g., the theorem statement involves a truncation of the rows of the last layer which on an event of large probability has no effect. It would be cleaner simply to assume all the weights are gaussian and then prove that the norms are bounded by $3\sqrt{n_{d-1}/\ell}$ with high probability. The paper does not explicitly state a result showing that the proposed algorithm converges to (a neighborhood of) a global minimizer. This seems reasonable on intuitive grounds, and perhaps a simple proof (in function values?) can be given based on the work that is already in the paper. Comments on algorithmic implications would be helpful. The paper asserts that the correct hyperbolic curve has the lowest objective value. I believe this is true, although it does not show up in theorem statement. Has this been proved? EDIT: After considering the other reviews and the authors' response, I still rate the submission a 7. The authors will address some clarity issues in the final statement if accepted. The response was not completely convincing on the algorithmic implications of the geometric analysis, but overall there is enough technical contributions to recommend acceptance anyway.

Reviewer 3



This paper studies blind demodulation problem, in which we hope to recover two unknown vectors from their entrywise multiplication. This paper assumes that these two unknown vectors are generated from two expansive ReLU neural networks, where the output dimensions are at least quadratic in the input dimensions. The authors study the optimization landscape of the square loss function and show that there is no stationary points outside of the neighborhoods around four hyperbolic curves and the set of local maximizers. Motivated by this landscape property, the authors further design a gradient descent scheme, which tries to find the hyperbolic curve with the lowest objective. In the experiments, the authors show that this algorithm can effectively remove distortions in the MNIST dataset. The paper is well written. In the proof, the authors first show that the result holds as long as the weight matrices satisfy the weight distributed condition and the joint weight distributed condition, then prove random Gaussian matrices satisfy these properties with high probability. Here are my major comments: 1. This paper assumes that the two generative models (neural networks) have random Gaussian matrices. In practice, it’s hard to imagine that any meaningful network has only random Gaussian weight matrices. Thus, I was wondering to what extent this result can be generalized to other weight matrices. 2. This paper also assumes the two generative models as prior. This assumption feels very strong to me. Is it possible to only assume the architecture of the two networks instead of the value of its weights? 3. In the paper, the proposed algorithm is only tested in the experiments, though without any theoretical guarantee. Since the algorithm is inspired by the landscape properties, I would expect some theoretical analysis for the algorithm. It would be good to discuss the challenges for analyzing this algorithm in the paper. 4. In many places, the authors claim set K is the set of local maximizers. However, in Theorem 2, the authors only show that the partial derivative along any direction is non-negative. I don’t think this is enough for proving that points in K are local maximizers. It’s possible that a point has zero gradient and a positive definite Hessian, in which case it’s a local minimizer. Feel free to correct me if my understanding is wrong. ------------------------------------------------------------------- I have read the authors' response and will not change my score. I am not an expert in this field and am not sure whether it's reasonable to assume the generative network has random Gaussian weights. The authors can explain more on the practicality of this model in the revision. In the rebuttal, the authors mentioned that the result still holds as long as the weight matrix is close to random Gaussian. It would be good to formally formulate this set of approximate random Gaussian matrices, which might make the result stronger.

[Author Response · NeurIPS 2019]

## Reviewer 1

- **Sparsity level vs latent code dimension of same signal:** We can expect the latent code dimensionality of a GAN to be smaller than the sparsity level of the corresponding image with respect to a wavelet basis. For example, consider a set of images that correspond to a single train going down a single track. This set of images form a one dimensional submanifold of the manifold of natural images. If properly parameterized by a generative model, then it would have a latent dimensionality of approximately 1, whereas the number of wavelet coefficients needed to describe any of those images is much greater. The work of Bora et al. shows that compressed sensing can be done with 5-10x fewer measurements than sparsity models. This provides evidence for the more economical representation of generative models than of sparsity models. Additionally, it is more natural to view the natural signal manifold as a low-dimensional manifold, as opposed to being the combinatorially-many union of low dimensional spaces. Performance gains are provided by the fact that the natural signal manifold can be directly exploited, whereas the union of subspaces can only be indirectly exploited via convex relaxations. We will add exposition to this effect to the paper.

## Reviewer 2

- **Sparsity level vs latent code dimension of same signal:** Please see response to Reviewer 1 regarding sparsity level and latent code dimension of same the signal.

- **A2 and A3 conditions in the theorem statements:** Thank you for your note. By $n_i = \Omega(n_{i-1})$, up to a log factor, we mean that there exists an absolute constant $c > 0$ such that $n_i \geq cn_{i-1} \log(n_{i-1})$. We will make this clear in the paper.

- **Truncation of the rows of the last layers:** Thank you for the insightful observation. We agree that on an event of high probability the truncation of the rows have no effect. This will make the theorem shorter and easier to understand. We will change the assumptions of the theorem and its proof to reflect this observation. It will result in a 1-2 line addition to the proof.

- **Convergence of the algorithm:** The focus of the paper was to show that the empirical risk objective function, under certain conditions, has a descent direction for every point outside a neighborhood of four hyperbolic curves, one of which contains the set of global minimizers. Based on this landscape of the objective function, we propose a gradient descent scheme, which will converge to a point in one of the four neighborhoods. In principle, a convergence result to the global minimizer by gradient descent is possible, though it would be considerably challenging because it would require showing a convexity-like property around the hyperbola. We believe that such an exposition would significantly increase the technicality of the paper, and we leave this for possible future work. We will clarify this in the paper.

- **Hyperbolic curve with lowest objective value:** The correct hyperbolic curve has the lowest objective because the objective is non-negative and the objective value at the correct hyperbolic curve is zero. This is the reason by which the negation-checking tweak of the gradient descent algorithm works.

## Reviewer 4

- **Random gaussian matrices:** In the paper we provide two deterministic conditions that are sufficient to characterize the landscape of the objective function, and show that Gaussian matrices satisfy these conditions. In essence, we only require approximate Gaussian matrices. We also note that the state-of-the-art literature for provable convergence of the training of neural networks (for regression and classification) admit proofs only in the case that the final trained weights are close to their random initialized. Thus, our neural network assumptions are consistent with the best known cases for which networks can be provably trained. We will add a few sentences to this effect in the paper.

- **Generative model assumptions:** The benefit of using a generative prior is that it is a reusable prior and can be used in multiple inverse problems. For example, a generative prior can be trained once and used in in-painting, image-to-image translation, compressed sensing, blind deconvolution, etc.

- **Convergence of the algorithm:** Please see response to reviewer 2 regarding convergence of a gradient descent scheme to a global minimizer.

- **The set $\mathcal{K}$ as the set of local maximizers:** Thank you for the observation. We agree that the directional derivative at a point along any direction being non-negative does not rule out the possibility of that point being a local minimizer. In the proof of Theorem 2, we show that for every point in the set $\mathcal{K}$, the directional derivative along a set of directions is zero and for every other direction, the directional derivative is strictly negative. Thus, the points in the set $\mathcal{K}$ are local maximizers. We will modify the Theorem statement and the proof to reflect this clearly in the final manuscript.

[Meta-Review · NeurIPS 2019]

This paper gives geometric analysis of blind demodulation with generative priors, and designed an alternating gradient method. The reviewers found the technical contributions to be solid.